# DeepPrim: a Physics-Driven 3D Short-term Weather Forecaster via Primitive Equation Learning

**Jiawei Chen**[1,2,3]***Weiqi Chen**[2,3]*, **Rong Hu,**[1,]*, **Peiyuan Liu**[2,3,4], **Haifan Zhang**[2,3], **Liang Sun**[2,3†]
[1]Zhejiang University [2] DAMO Academy, Alibaba Group
[3] Hupan Laboratory [4] Tsinghua University

## Abstract

Solving primitive equations is essential for accurate weather forecasting. However, traditional numerical weather prediction (NWP) methods often incorporate various simplifications that limit their effectiveness in parameterizing unresolved physical processes. Meanwhile, existing deep learning-based models mostly focus on pure data-driven paradigms, overlooking the fundamental physical principles that govern atmospheric dynamics. To address these challenges, we present **DeepPrim**, a novel 3D deep weather forecaster designed to learn primitive equations of the Earth's atmosphere. Specifically, DeepPrim aims at accurately modeling 3D atmospheric motion through Navier-Stokes equation in pressure coordinates and effectively capturing the interactions between the solved advection and key weather variables (e.g., temperature and water vapor) through corresponding equations. By seamlessly integrating fundamental atmospheric physics with advanced data-driven techniques, our model effectively approximates complicated physical processes without relying on empirical simplifications. Experimentally, DeepPrim achieves impressive performance in short-term global and regional weather forecasting tasks and exhibits the superior capacity to capture 3D atmospheric dynamics. It is now deployed as part of the Baguan weather forecasting system, especially specializing in short-term forecasting. The code is available at https://github.com/DAMO-DI-ML/DeepPrim.

## 1 Introduction

Weather forecasting is an essential tool for planning, decision-making, and risk mitigation in many industries (Coiffier, 2011). Traditionally, atmospheric researchers developed numerical weather prediction (NWP) systems by solving primitive equations and other equations related to the Earth atmosphere, such as the water vapor equation (Hurrell et al., 2013; Bouallègue et al., 2024). As a widespread and prevalent paradigm in recent decades, NWP-based forecasting systems offer notable interpretability and accuracy, but may suffer from parameterization errors of unresolved physical processes and high complexity in solving partial differential equations (PDEs) (Kochkov et al., 2024). To address these limitations, deep learning-based weather forecasting methods have emerged, leveraging deep neural networks such as Vision Transformers (ViTs) (Nguyen et al., 2023; Bi et al., 2023), Graph Neural Networks (GNNs) (Lam et al., 2023; Keisler, 2022), or neural ordinary differential equation (ODEs) (Verma et al., 2024) in a data-driven fashion based on comprehensive datasets such as ECMWF Reanalysis v5 (ERA5) (Hersbach et al., 2020).

Despite demonstrating promising performance, existing deep weather forecasters are still facing three key challenges: ❶ *Pressure-level-aware 3D atmospheric dynamics*. As illustrated in Fig. 1, the weather state evolutions are influenced by the longitude-latitude-pressure level 3D atmospheric dynamics, which can often be described by Navier-Stokes equations (Bauer et al., 2015). While current deep forecasters achieve strong performance by modeling atmospheric variables across pressure levels as spatial-temporal videos or 2D fluid fields, these methods may not explicitly model verti-

---

*Equal contribution. This work was done when Jiawei Chen was a research intern at DAMO Academy.
† Corresponding author.

cally coupled dynamics governed by physical principles (e.g., pressure-aware advection, inter-level energy transfer). Thus, the exploration of 3D interaction modeling could serve as a complementary inductive bias to further enhance the learning of weather representation. ❷ *Critical physics priors of the Earth's atmosphere*. While recent advances in data-driven weather forecasting demonstrate certain capabilities in capturing atmospheric physics (Hakim & Masanam, 2024) with end-to-end training, most deep forecasters predominantly rely on statistical correlations from historical data without explicit incorporation of fundamental physical constraints such as mass, momentum, and energy conservation. ❸ *Short-term forecasting for real-time decision-making*. Short-term forecasting, especially sub-24-hour prediction, is crucial for real-time actionable planning in agriculture, disaster mitigation, and energy management (Weckwerth et al., 2005; Klein et al., 2015). More importantly, physical law adherence (e.g., abrupt convective processes, localized storm evolution) (Verma et al., 2024) and 3D atmospheric dynamic modeling (e.g., convective updrafts and inter-level energy transfer) are effective and crucial in modeling short-term weather evolutions.

In this paper, we reconcile the intrinsic tension between these challenges and address them by *incorporating physical priors to model 3D atmospheric dynamics* in a coherent framework, **DeepPrim**, which integrates pressure-level-aware 3D neural networks with primitive equations of atmospheric physics for continuous-time *short-term weather forecasting* in a Neural ODE system. DeepPrim first models atmospheric motion by learning force terms in the Navier-Stokes equation with a 3-dimensional bicomponent Vision Transformer (3D-BiViT), which can explicitly model the atmospheric dy-

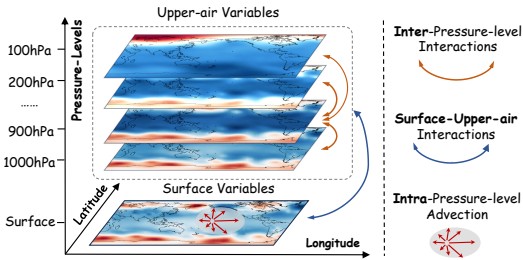

Figure 1: **3D atmospheric dynamics** in weather evolutions with the coordinates of longitude-latitude-pressure levels ($x$-$y$-$p$).

namics in upper-air and at surface differentially yet collaboratively. Followingly, the model uses the learned advection to model key weather variables through corresponding equations, with a source-sink network to characterize the resolved and unresolved physical processes. In summary, our contributions include:

- We introduce DeepPrim for short-term weather forecasting, pioneering the explicit learning of primitive equations using deep learning and enhancing deep weather forecasting by integrating critical physical principles of the Earth's atmospheric dynamics.

- We propose 3D-BiViT to learn the force terms in the Navier-Stokes equation, effectively modeling the 3D atmospheric dynamics by capturing both inter- and intra-pressure-level interactions for upper-air and surface variables differentially yet collaboratively.

- DeepPrim outperforms most recent counterpart methods, reducing RMSE by 38.7% and 35.8% in short-term global and regional weather forecasting tasks, respectively. It also demonstrates superior abilities to capture pressure-level-aware 3D atmospheric dynamics, reliably characterizing continuous-time weather evolutions. DeepPrim has been deployed as part of the Baguan weather forecasting system (Niu et al., 2025), specializing in short-term forecasting.

## 2 RELATED WORK

**Numerical Weather Prediction** As a predominant paradigm, NWP systems mostly formulate the atmospheric physical rules via PDEs and numerical simulations. Typical methods include Earth System Models (ESM) (Hurrell et al., 2013) and the operational Integrated Forecast (IFS) of ECMWF (Bouallègue et al., 2024). By integrating physics laws, NWP approaches have achieved remarkable success, demonstrating high accuracy, robust stability, and strong interpretability. On the other hand, it suffers from sensitivity to initial conditions, errors of parameterization, and high computational cost (Kochkov et al., 2024).

**Deep Learning for Weather Forecasting** Recently, deep learning methods have emerged as powerful weather forecasters due to their strong expressive ability. Built upon vision backbones, e.g., ViT and its variants, ClimaX (Nguyen et al., 2023), FourCastNet (Pathak et al., 2022), Pangu (Bi et al., 2023), FengWu (Chen et al., 2023a), and FuXi (Chen et al., 2023b) treat regional grids as patch tokens to forecast future weather states via certain pre/post training strategies. Alternatively,

Table 1: **Overview and comparison of typical data-driven weather forecasters**. $x$-$y$-$p$ denote longitude-latitude-pressure levels. *More detailed analysis can be found in Table 9 and Section B.5.*

| Method | Continuous-time | Physics-informed | $x$-$y$-$p$ 3D dynamics | Key techniques for 3D atmospheric modeling |
|---|---|---|---|---|
| GraphCast (2023) | ✗ | ✗ | ✗ | N/A |
| ClimaX (2023) | ✗ | ✗ | ✗ | N/A |
| FuXi (2023) | ✗ | ✗ | ✗ | N/A |
| WeatherGFT (2024) | ✓ | ✓ | ✗ | N/A |
| ClimODE (2024) | ✓ | ✓ | ✗ | N/A |
| Pangu (2023) | ✗ | ✗ | ✓ | Unified attention blocks for $x$-$y$-$p$ coordinates by treating pressure levels as geometric heights. |
| NeuralGCM (2024) | ✓ | ✓ | ✓ | Local neural networks in space on individual vertical columns of the atmosphere that output tendencies. |
| **DeepPrim** (Ours) | ✓ | ✓ | ✓ | **Physics-informed** dual-attention for explicit and customized **intra- and inter- pressure level** modeling. |

GraphCast (Lam et al., 2023) and Artificial Intelligence Forecasting System (AIFS) (Lang et al., 2024) employ GNNs and transform earth systems as graph representations for weather predictions. Compared with NWP methods, deep learning-based models present higher inference efficiency and scalability, but may overlook the fundamental physical laws and often focus on forecasting at pre-defined fixed lead time.

**Physics-informed Deep Weather Forecasting** To integrate both physics priors and deep learning's expressive capacity, physics-informed deep weather forecasting (Luo et al., 2025) has been attracted increasing attention recently. For example, DeepPhysiNet (Li et al., 2024), PINNs (Cai et al., 2021), and PINO (Li et al., 2021) incorporate physics law-embedded PDEs into loss functions to guide the deep forecasters. ClimODE (Verma et al., 2024), as a pioneering work, first introduces continuity-equation–inspired neural advection within a Neural ODE framework for weather forecasting. NeuralGCM (Kochkov et al., 2024) combines general circulation models with machine learning for weather and climate. WeatherGFT (Xu et al., 2024) incorporates physical PDEs into ViTs in a parallel manner and leverages the latter as the correction module. Recently, (Karlbauer et al., 2024) provides a systematic comparison of deep learning backbones for Navier-Stokes and atmospheric dynamics, offering a useful guideline for backbone choices.

**Deep Learning for PDEs** Beyond weather-specific applications, there is a rapidly growing literature on deep learning methods for partial differential equations (PDEs) that explicitly incorporate physical constraints. Hard-constrained approaches learn differentiable solvers that enforce constraints during training and inference (Négiar et al., 2022), and have recently been scaled using mixture-of-experts designs (Chalapathi et al.) and conservation-law-respecting architectures (Hansen et al., 2023). Other works leverage uncertainty quantification to improve out-of-domain generalization for PDEs (Mouli et al.) or formulate end-to-end probabilistic frameworks for learning with hard constraints (Utkarsh et al., 2025).

**Research Gap** In general, existing weather forecasters mostly focus on spatial-temporal variations by treating weather states as videos or 2D fluids. The pressure-level information and interactions, despite being fundamental and critical, are seldom investigated and not explicitly exploited, as shown in Table 1. Pangu (Bi et al., 2023) uses unified attention blocks and positional embedding to encode height information, which may not explicitly distinguish heterogeneous horizontal-vertical interactions. NeuralGCM (Kochkov et al., 2024) employs individual networks to model physical tendencies in the corresponding single vertical column, which may omit the inference of intra-pressure-level influences. Inspired by the primitive equations, our DeepPrim bridges the gap between pressure-level-aware explicit and flexible 3D dynamics modeling with physics-informed continuous-time weather forecasting in the neural ODE system, offering a more realistic and reliable short-term weather forecasting paradigm.

## 3 LEARNING PRIMITIVE EQUATIONS VIA DEEPPRIM

We model the evolution of weather states as a spatial-temporal process $\mathbf{u}(\mathbf{c}, t) = (u_1(\mathbf{c}, t), ..., u_K(\mathbf{c}, t)) \in \mathbb{R}^K$ of $K$ variables $u_k(\mathbf{c}, t) \in \mathbb{R}$ over continuous time $t \in \mathbb{R}$ and latitude-longitude coordinates $\mathbf{c} = (x, y) \in \Omega = [-90°, 90°] \times [-180°, 180°] \in \mathbb{R}^2$. The total number of variables is represented as $K = D_S + P \times D_U$, where $D_S$ denotes the surface variables, include $v_x$ (latitude-direction wind), $v_y$ (longitude-direction wind), and $T$ (temperature). Additionally, $D_U$ refers to the upper-air variables measured at $P$ pressure levels, with added $\phi$ (geopotential) and $q$

(humidity). The weather forecasting problem can be formulated as:

$$\mathbf{u}(t_{-2} : t_0) \xrightarrow{f(\cdot)} \hat{\mathbf{u}}(t_1 : t_N), \tag{1}$$

where $f(\cdot)$ denotes the (learned) weather dynamics. The model takes as input three weather states, which correspond to the current time, $t_0$, and predicts weather states at the next $N$ time steps. The complete notations and explanations are summarized in Table 5 in Section B.1.

## 3.1 PHYSICAL MODEL

Solving the primitive equations (along with the water vapor equation) of the Earth's atmosphere, which comprise a set of equations related to wind, temperature, pressure, geopotential, and water vapor, provides the foundation for NWP (Kalnay, 2003). Although various variables are interconnected, the momentum equation that describes atmospheric motion serves as the cornerstone (Bauer et al., 2015), as the variations of other weather variables (such as temperature and water vapor) are primarily driven by horizontal advection within the atmospheric circulation scale (Emanuel, 1994). Therefore, accurate modeling of atmospheric motion is crucial for forecasting these prognostic variables. To achieve this, we first consider the Navier-Stokes equation in pressure coordinates ($x$-$y$-$p$), which enables us to account for detailed viscous friction as

$$\frac{d\mathbf{v}}{dt} + \underbrace{f\mathbf{k} \times \mathbf{v}}_{\text{Coriolis force}} = \underbrace{-\nabla_p \phi}_{\text{pressure gradient force}} \underbrace{+ \nu \nabla_p^2 \mathbf{v} + A \frac{\partial^2 \mathbf{v}}{\partial p^2}}_{\text{viscous friction}}, \tag{2}$$

where $\mathbf{v} = (v_x, v_y)$ is the horizontal wind, $\phi$ is the geopotential, $f = 7.29e\text{-}5$, $\nabla_p = \left[\frac{\partial}{\partial x}; \frac{\partial}{\partial y}\right]$ denotes horizontal gradients, $\nu$ and $A$ are the viscosity coefficient and turbulent exchange coefficient, respectively. The expansion of $\frac{d(\cdot)}{dt}$ is as follows:

$$\frac{d(\cdot)}{dt} = \frac{\partial(\cdot)}{\partial t} + \mathbf{v} \cdot \nabla_p(\cdot) + \omega \frac{\partial(\cdot)}{\partial p}, \tag{3}$$

where $\omega$ is the vertical wind, which dynamically evolves over time and can be derived from the continuity equation:

$$\omega = -\int \left(\frac{\partial v_x}{\partial x} + \frac{\partial v_y}{\partial y}\right) dp. \tag{4}$$

The coefficients $\nu$ and $A$ are typically estimated using empirical formulas in practice, which may lead to a certain degree of imprecision. Moreover, other physical processes are modeled under moderate simplifications when solving primitive equations. For example, in most general circulation models, temperature is constrained by a pseudo-adiabatic process considering moist processes such as condensation and evaporation, while other processes, including heat originating from friction, radiative heating, and cooling, remain not considered (Randall et al., 2007; Stocker, 2014).

To describe these processes, parameterization methods are introduced to approximate the interactions between resolved and unresolved processes. It is important to note that these parameterization methods are not unique and can be customized or adjusted within different numerical schemes based on experiments and observations. However, since experiments and observations data are geographically limited, this can result in considerable variations in the calculated outcomes of parameterization schemes for the same process. Thus, the selection of specific parameterization schemes and configurations can lead to significantly different simulation results (Banks et al., 2016). Conversely, deep neural networks offer a way to bypass these simplifications by directly learning resolved and unresolved processes through a data-driven approach.

***Takeaways.*** The aforementioned mathematical models provide ***three key physical insights*** guiding the design of our learning framework[1]: ❶. As estimating the horizontal advection is the key in solving the primitive equations, the modeling of horizontal motion should be carefully designed. ❷. The viscous friction term in the Navier-Stokes equations highlights the appropriateness of incorporating higher-order gradients and 3D interactions to accurately model atmospheric motion. The pressure gradient force term suggests that additional variables should also be incorporated. ❸. The use of

---

[1]We provide detailed formulations of primitive equations and analyze their relation to our model designs in Section C.

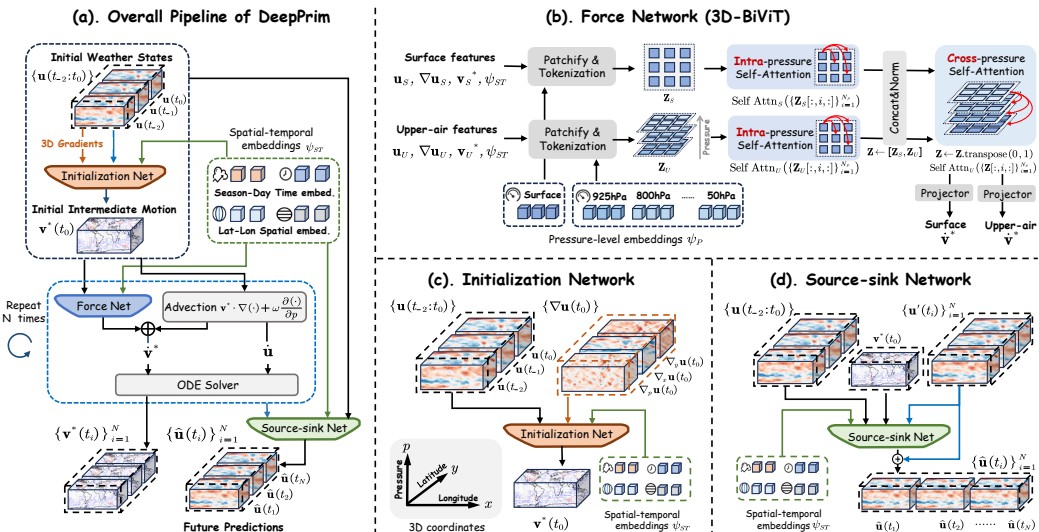

Figure 2: **DeepPrim Framework** and key modules, including (a) overall pipeline and architectures of (b) force network based on 3D-BiViT (Sec. 3.2.2), (c) initialization network (Sec. 3.2.1), and (d) source-sink network (Sec. 3.2.3). *More figure details can be found in Appendix B.2 and B.4.*

parameterization methods encourages us to learn a source-sink term in the advection equation to simulate both resolved and unresolved processes. By doing so, we can replace the parameterization schemes with a learnable source-sink network that minimizes the discrepancy between simulations and observations.

## 3.2 DeepPrim as Physics-Driven Weather Forecaster

**Model Overview** Building upon the insights derived from the physics discussed in Section 3.1, we model the force term and the source-sink term using specially designed neural networks. Treating these networks as universal approximators, the derived equations are expressed as follows:

$$
\begin{aligned}
\dot{\mathbf{v}}^* &:= \frac{\partial \mathbf{v}^*}{\partial t} = -\left( \mathbf{v}^* \cdot \nabla_p \mathbf{v}^* + \omega \frac{\partial \mathbf{v}^*}{\partial p} \right) + \text{Force}(\mathbf{u}, \mathbf{v}^*) \\
\dot{\mathbf{u}} &:= \frac{\partial \mathbf{u}}{\partial t} = -\left( \mathbf{v}^* \cdot \nabla_p \mathbf{u} + \omega \frac{\partial \mathbf{u}}{\partial p} \right) + \text{Source-Sink}(\mathbf{u}, \mathbf{v}^*),
\end{aligned}
\tag{5}
$$

where $\omega$ can be calculated by Eq. (4). The force model $\text{Force}(\cdot)$ encapsulates external forces including pressure gradients, Coriolis forces, and frictional effects. Meanwhile, the source-sink model $\text{Source-Sink}(\cdot)$ embodies the contributions from both resolved and unresolved processes, which may include phenomena such as turbulent mixing and energy exchanges that are not explicitly modeled. It is important to note that in the formula for the primitive equations, $\mathbf{v}$ denotes horizontal wind, and $\mathbf{u}$ typically represents other prognostic variables (i.e., $T$, $q$, $\phi$). To enhance the model's capability and flexibility, our learning framework treats $\mathbf{u}$ by including all prognostic variables, encompassing horizontal wind, with $\mathbf{v}^*$ denoting the intermediate atmospheric motion. Consequently, applying the source-sink network to the wind component can be viewed as a correction mechanism.

Given the initial values $\mathbf{u}(t_0)$ and $\mathbf{v}^*(t_0)$, the numerical solution can be accurately approximated with ODE solvers such as Euler (Biswas et al., 2013) and Runge-Kutta (Runge, 1895) by discretizing the system into multiple discrete time steps as in Eq. (6) as follows:

$$
\begin{aligned}
\begin{bmatrix} \mathbf{u}(t_i) \\ \mathbf{v}^*(t_i) \end{bmatrix} &= \begin{bmatrix} \mathbf{u}(t_0) \\ \mathbf{v}^*(t_0) \end{bmatrix} + \int_{t_0}^{t_i} \begin{bmatrix} \dot{\mathbf{u}}(\tau) \\ \dot{\mathbf{v}}^*(\tau) \end{bmatrix} d\tau, \\
\mathbf{v}^*(t_0) &= [v_x, v_y](t_0) + \text{Initialization}(\mathbf{u}(t_0)).
\end{aligned}
\tag{6}
$$

Backpropagation of ODEs is compatible with standard autodiff, while also admitting tractable adjoint form (Chen et al., 2018). Since the introduced intermediate motion $\mathbf{v}^*$ is not entirely equivalent to the wind $\mathbf{v}$, we utilize a residual initialization network to estimate the initial values of $\mathbf{v}^*$.

To sum up, we propose DeepPrim with its overall pipeline depicted in Fig. 2 (a). It consists of three key modules motivated by Eqs. (5, 6): 1) ***Initialization network*** that estimates the initial values of the

intermediate motion fields considering the temporal variations and 3-dimensional spatial gradients of weather states; 2) **Force network** that leverages 3-dimensional bicomponent Vision Transformer (3D-BiViT) to model the time deviation of intermediate motion for upper-air and surface separately inspired by the force term of the Navier-Stokes equation; 3) **Source-sink network** that captures the value gains or losses of each variable caused by resolved and unsolved processes for final predictions. Note that we do not simulate the source and sink rate at each discrete ODE step in Eq. (5), but directly model the total amount of the source and sink after integration from start time to forecast time, which can ensure the stability of ODE integration and avoid overfitting. Details of each module are elaborated below.

### 3.2.1 INITIALIZATION NETWORK

Estimating initial values is crucial for ensuring the precision and stability of ODE solutions. In the numerical solutions of primitive equations in NWP, the initial values are typically derived from assimilated weather states. As we incorporate the intermediate motion into our learning framework, we introduce an initialization network that estimates the initial values of this intermediate motion more coherently within the ODE learning system.

Specifically, we model the initial intermediate motion by modulating the horizontal wind using convolutional networks. This approach accounts for both temporal variations and 3D spatial gradients of the initial weather conditions, represented mathematically as:

$$\mathbf{v}^*(t_0) = [v_x, v_y]_{t_0} + \mathrm{Conv}(\underbrace{\mathbf{u}(t_{-2}:t_0), \dot{\mathbf{u}}(t_0),}_{\text{temporal variations}} \underbrace{\nabla \mathbf{u}(t_0)}_{\text{spatial gradients}}, \psi_{ST}), \tag{7}$$

where $\nabla = \left[\frac{\partial}{\partial x}; \frac{\partial}{\partial y}; \frac{\partial}{\partial p}\right]$ denotes 3D spatial gradients. The initialization network integrates various additional features to capture essential information relevant to atmospheric conditions. These features include spatial-temporal derivatives of various weather variables, land-sea mask, incoming solar radiation, orography, and spatial-temporal embeddings (denoted as $\psi_{ST}$) that convey both temporal information (such as day and season) and spatial location information (longitude and latitude). Detailed implementation of these embeddings can be found in Section B.3.

### 3.2.2 FORCE NETWORK

The force network is introduced to learn the force term in the Navier-Stokes equation for modeling atmospheric dynamics. As previously discussed, the structure of viscous friction implies 3D interactions within the neural network. Thus, we design a 3-dimensional bicomponent Vision Transformer (3D-BiViT) to learn the force term for upper-air and surface, differentially yet collaboratively. This network utilizes features consistent with the initialization network as input along with $\mathbf{v}^*(t)$, encompassing three core modules.

**Tokenization and Embedding** Technically, we adopt ViT (Dosovitskiy et al., 2021) as the backbone. As depicted in Fig.2 (b), the input channels are first tokenized into a sequence of $N = (H/p) \times (W/p) = h \times w$ patches, where $p \times p$ is the patch size and $N$ is the number of patches. These input tokens are then embedded into vectors of dimension embed_dim with convolution networks. Moreover, due to the distinct characteristics of surface and upper-air dynamics, we set two distinct embedding layers for upper-air and atmospheric varibles.

**Pressure-Level Embeddings** Subsequently, these token embeddings are added with adaptive pressure-level embeddings. The pressure-level embeddings are designed to inform the model of pressure information. We implemented the pressure-level embeddings $\psi_P$ as learnable parameters with shape $[1, 1, P+1, \mathrm{embed\_dim}]$, where $P+1$ is the total pressure levels (with upper-air variables of $P$ and surface variables of 1). The pressure-level embeddings are added to the token embeddings with shape $[B, N, P+1, \mathrm{embed\_dim}]$.

**Intra- and Cross-Pressure Level Attentions** To characterize 3D atmospheric dynamics, we propose a two-stage self-attention mechanism for 3D dynamics learning. For the concatenated representations of surface and upper-air variables $[\mathbf{Z}_S, \mathbf{Z}_U]$ with shape $(B \times (P+1) \times N \times \mathrm{embed\_dim})$, the intra-pressure self-attention operates on the tokens within each pressure level (i.e., $N$) to model the inter-pressure-level dynamics. Afterwards, the cross-pressure level self-attention layer operates on the tokens across different pressure levels (i.e., $P+1$) to model the vertical (intra-pressure-level)

atmospheric dynamics. Similar with embedding layers, projections of query, key, and value, as well as the feed-forward networks in the intra-pressure attention blocks, are learned separately for upper-air and surface tokens.

### 3.2.3 SOURCE-SINK NETWORK

The weather state evolutions may undergo value loss or gain processes due to the energy or matter exchange within the atmosphere and between the atmosphere and the external system, such as the radiative heating and cooling, day-night cycle, as well as condensation and evaporation. To model such impacts, we introduce a source model to further account for these resolved and unresolved processes. Following the aforementioned discussion, we do not model the source and sink rate in Eq. (5), but model the total amount of source and sink from $t_0$ to $\{t_i\}_{i=1}^N$. Specifically, the source-sink network receives the initial weather states $\mathbf{u}(t_0)$ and initial intermediate motion $\mathbf{v}^*(t_0)$, and ODE solutions $\{\mathbf{u}'(t_i)\}_{i=1}^N$ along with spatiotemporal embeddings $\psi_{ST}$ as input to correct the output of ODE solutions, which is formulated as

$$\{\hat{\mathbf{u}}(t_i)\}_{i=1}^N = \text{Conv}\left(\{\mathbf{u}(t_0)\}, \mathbf{v}^*(t_0), \{\mathbf{u}(t_i)\}_{i=1}^N, \psi\right) + \{\mathbf{u}'(t_i)\}_{i=1}^N, \tag{8}$$

where $\{\hat{\mathbf{u}}(t_i)\}_{i=1}^N$ is the final prediction of the model.

### 3.3 OPTIMIZATION OBJECTIVE

We provide the overall implementations of DeepPrim in Algorithm 1 in Section B.2. In the training stage, we minimize the latitude-weighted MSE loss between the predicted values $\hat{\mathbf{u}}$ and the ground truth $\mathbf{u}$ at the target lead time $t_N$ [2]:

$$\mathcal{L} = \frac{1}{KHW} \sum_{k=1}^K \sum_{h=1}^H \sum_{w=1}^W \alpha(h)\left(\hat{\mathbf{u}}_{k,h,w}(t_N) - \mathbf{u}_{k,h,w}(t_N)\right)^2, \tag{9}$$

where $\alpha(h)$ is the latitude weighting factor that accounts for the varying grid cell areas on a spherical Earth, and more details on the weighting factor can be found in Section E.

## 4 EXPERIMENTS

**Dataset** We use the ERA5 dataset (Rasp et al., 2020) with $5.625°$ ($64 \times 32$ latitude-longitude grids), $1.40625°$ resolutions ($256 \times 128$) and $0.25°$ resolutions ($1440 \times 720$). Following ClimaX (Nguyen et al., 2023), we use the data from 1979 to 2015 as the training set, 2016 as the validation set, and 2017 to 2018 as the test set. The performance is evaluated on five key variables: geopotential at 500 hPa ($z500$), temperature at 850 hPa ($t850$), temperature at 2 meters ($t2m$), and wind speeds at 10 meters ($u10$ and $v10$). More details of the dataset and experimental settings can be found in Section D.

**Baselines** We extensively compare DeepPrims with several counterparts on ***Resolution-Matched "Apples-to-Apples"*** comparison scenarios: FourCastNet (FCN) (2022), ClimaX (2023), Pangu (2023), GraphCast (2023), ClimODE (2024), NeuralGCM (2024) and WeatherGFT (2024). We also add Integrated Forecasting System (IFS) (2023) as a gold-standard baseline, which is recognized as amongest the most skillful NWP in the world. Also, we evaluate the performance using *latitude-weighted root mean squared error* (RMSE). More details of baselines and the performance results can be found in Section F and Table 16. More details of metrics can be found in Section E.

**Implementations** The backbones of the initialization and the source-sink models are ResNet (He et al., 2016), and the backbone of the force network is based on ViT (Dosovitskiy et al., 2021). We report the results of two variants of DeepPrim: 1) "DeepPrim" takes all surface and upper-air variables as output variables for training; 2) "DeepPrim[†]" only outputs five key variables, in line with ClimODE and ClimaX. More implementation and hyperparameter details are in Section B.

**Tasks** We consider the tasks of global weather forecasting (Section 4.1) and regional weather forecasting (Section A of Appendix). More detailed task settings are provided in Section D.3.

---

[2]Alternatively, one can also add supervision signals to the sequence of weather states $\{\hat{\mathbf{u}}(t_i)\}_{i=1}^N$. In our experiments, we chose the form in Eq. (9) for computational efficiency.

Table 2: **Global weather forecasting results on 5.625° and 1.40625° ERA5 dataset.** The best results are **bolded** and the second-best results are underlined, except for IFS. The RMSE curves along lead time are shown in Fig. 9. *The result sources are summarized in Table 16*.

| Variable | Lead-Time $\Delta t$ (h) | Latitude-weighted RMSE (5.625° ERA5) | | | | | | | | Latitude-weighted RMSE (1.40625° ERA5) | | | | |
|---|---|---|---|---|---|---|---|---|---|---|---|---|---|---|
| | | IFS (2023) | NODE (2018) | FCN (2022) | ClimaX* (2023) | ClimaX (2023) | ClimODE (2024) | DeepPrim Ours | DeepPrim† Ours | IFS (2023) | NeuralGCM (2024) | ClimaX (2023) | WeatherGFT (2024) | DeepPrim Ours |
| z500 | 6 | 26.9 | 300.6 | 149.4 | 247.5 | 62.7 | 102.9 | **50.1** | 53.6 | 26.96 | 28.43 | 49.67 | **22.10** | 27.44 |
| | 12 | (N/A) | 460.2 | 217.8 | 265.3 | 81.9 | 134.8 | **71.0** | 77.4 | (N/A) | **35.01** | (N/A) | 36.71 | 40.15 |
| | 18 | (N/A) | 627.6 | 275.0 | 319.8 | **88.9** | 162.7 | 94.1 | 102.2 | (N/A) | **40.52** | (N/A) | 41.41 | 46.82 |
| | 24 | 51.0 | 877.8 | 333.0 | 364.9 | **96.2** | 193.4 | 121.0 | 126.7 | 50.96 | **48.21** | 72.76 | 54.13 | 59.72 |
| t850 | 6 | 0.69 | 1.82 | 1.18 | 1.64 | 0.88 | 1.16 | 0.76 | **0.72** | 0.69 | 0.52 | 0.84 | 0.46 | **0.44** |
| | 12 | (N/A) | 2.32 | 1.47 | 1.77 | 1.09 | 1.32 | 0.93 | **0.87** | (N/A) | 0.58 | (N/A) | 0.59 | **0.53** |
| | 18 | (N/A) | 2.93 | 1.65 | 1.93 | 1.10 | 1.47 | 1.04 | **0.98** | (N/A) | 0.62 | (N/A) | 0.63 | **0.55** |
| | 24 | 0.87 | 3.35 | 1.83 | 2.17 | 1.11 | 1.55 | 1.13 | **1.06** | 0.87 | 0.76 | 1.02 | 0.71 | **0.69** |
| t2m | 6 | 0.97 | 2.72 | 1.28 | 2.02 | 0.95 | 1.21 | 0.94 | **0.73** | 0.97 | (N/A) | 1.11 | 0.67 | **0.63** |
| | 12 | (N/A) | 3.16 | 1.48 | 2.26 | 1.24 | 1.45 | 1.12 | **0.88** | (N/A) | (N/A) | (N/A) | 0.72 | **0.70** |
| | 18 | (N/A) | 3.45 | 1.61 | 2.45 | 1.19 | 1.43 | 1.17 | **0.97** | (N/A) | (N/A) | (N/A) | 0.76 | **0.73** |
| | 24 | 1.02 | 3.86 | 1.68 | 2.37 | 1.10 | 1.40 | 1.19 | **0.99** | 1.02 | (N/A) | 1.19 | 0.81 | **0.76** |
| u10 | 6 | 0.80 | 2.30 | 1.47 | 1.58 | 1.08 | 1.41 | 0.89 | **0.82** | 0.79 | (N/A) | 1.04 | **0.54** | 0.62 |
| | 12 | (N/A) | 3.13 | 1.89 | 1.96 | 1.23 | 1.81 | 1.07 | **1.00** | (N/A) | (N/A) | (N/A) | **0.71** | 0.71 |
| | 18 | (N/A) | 3.41 | 2.05 | 2.24 | 1.27 | 1.97 | 1.24 | **1.15** | (N/A) | (N/A) | (N/A) | **0.77** | 0.79 |
| | 24 | 1.11 | 4.10 | 2.33 | 2.49 | 1.41 | 2.01 | 1.39 | **1.25** | 1.11 | (N/A) | 1.31 | **0.90** | 0.96 |
| v10 | 6 | 0.94 | 2.58 | 1.54 | 1.60 | (N/A) | 1.53 | 0.92 | **0.85** | (N/A) | (N/A) | (N/A) | (N/A) | (N/A) |
| | 12 | (N/A) | 3.19 | 1.81 | 1.97 | (N/A) | 1.81 | 1.10 | **1.03** | (N/A) | (N/A) | (N/A) | (N/A) | (N/A) |
| | 18 | (N/A) | 3.58 | 2.11 | 2.26 | (N/A) | 1.96 | 1.27 | **1.19** | (N/A) | (N/A) | (N/A) | (N/A) | (N/A) |
| | 24 | 1.33 | 4.07 | 2.39 | 2.48 | (N/A) | 2.04 | 1.43 | **1.24** | (N/A) | (N/A) | (N/A) | (N/A) | (N/A) |

**(1).** "DeepPrim" takes all surface and upper-air variables as output variables for training, whose full results are summarized in Table 19; "DeepPrim†" only outputs five key variables; **(2).** "ClimaX*" denotes ClimaX that is not pre-trained on CMIP6 (Eyring et al., 2016); **(3).** "NeuralGCM" denotes the official 1.4° deterministic NeuralGCM model; **(4).** The results of IFS are cited from ClimaX (2023) and are consistent with those in ClimODE and WeatherGFT. *We provide more discussion of IFS and WeatherBench2 in Section F.2.*

## 4.1 GLOBAL WEATHER FORECASTING

We summarize the results of DeepPrim and baselines in Table 2 (5.625° and 1.40625° ERA5 datasets) and Table 3 (0.25° ERA5 datasets) at $\Delta t = \{6, 12, 18, 24\}$ hours. We also present and discuss the forecasting at longer lead time, i.e., $\Delta t = \{36, 72, 144\}$ hours, in Section G.2.

❶ **Results on 5.625°** Our DeepPrim/DeepPrim† generally achieves the best RMSE, outperforming ClimODE by 38.6%, and suppressing the pre-trained ClimaX by 10.7% on average. DeepPrim is comparable with IFS and outperforms IFS in predicting $t2m$ and $v10$. Also, the **five**-output-variable DeepPrim† generally outperforms the **full**-output-variable Deep-Prim, as full-variable optimization may require balance and compromise for all variables.

❷ **Results on 1.40625°** Our full-output-variable DeepPrim surpasses IFS in $t850$, $t2m$, and $u10$ predictions and outperforms the pre-trained ClimaX by 36.11% on average. With only **1/20** of the parameters of WeatherGFT as shown in Table 10, Deep-Prim achieves comparable performance while excelling in forecasting $t850$ and $t2m$. DeepPrim also achieves competitive performance with Neural-GCM.

Table 3: **Global forecast results on 0.25°.**

| Variable | Lead-Time $\Delta t$ (hours) | IFS (2023) | Pangu (2023) | GraphCast (2023) | DeepPrim Ours |
|---|---|---|---|---|---|
| z500 | 6 | 27.6 | 16.9 | **15.9** | 24.3 |
| | 12 | 38.6 | 30.2 | **28.5** | 31.8 |
| | 18 | 39.7 | 35.6 | **31.2** | 36.3 |
| | 24 | 50.8 | 45.2 | **41.0** | 44.8 |
| t850 | 6 | 0.70 | 0.42 | 0.27 | **0.24** |
| | 12 | 0.77 | 0.58 | 0.45 | **0.44** |
| | 18 | 0.78 | 0.64 | **0.49** | 0.50 |
| | 24 | 0.86 | 0.72 | 0.59 | **0.58** |
| t2m | 6 | 1.05 | 0.62 | **0.51** | 0.53 |
| | 12 | 1.10 | 0.68 | **0.54** | 0.56 |
| | 18 | 1.07 | 0.75 | 0.58 | **0.56** |
| | 24 | 1.14 | 0.72 | 0.63 | **0.62** |
| u10 | 6 | 0.95 | 0.47 | **0.38** | 0.34 |
| | 12 | 1.08 | 0.70 | 0.61 | **0.58** |
| | 18 | 1.11 | 0.79 | **0.68** | 0.67 |
| | 24 | 1.23 | 0.91 | 0.81 | **0.76** |
| v10 | 6 | 0.98 | 0.48 | 0.38 | **0.36** |
| | 12 | 1.1 | 0.72 | 0.63 | **0.58** |
| | 18 | 1.1 | 0.82 | 0.70 | **0.66** |
| | 24 | 1.26 | 0.94 | 0.84 | **0.82** |

❸ **Results on 0.25°** In comparison with IFS, Pangu(0.25°), and GraphCast(0.25°), DeepPrim has achieved the best overall performance, especially in the prediction of U &V wind speed, as Deep-Prim comprehensively models the force term and horizontal advection in a physically-informed manner. GraphCast excels at z500, as it leverages a graph-based structure that explicitly encodes both local and global interactions for modeling large-scale global circulation patterns.

## 4.2 REGIONAL WEATHER FORECASTING

In practice, making regional weather forecasting based on accessible local data is also critical. Traditional NWP methods require the given boundary conditions for regional prediction. In contrast,

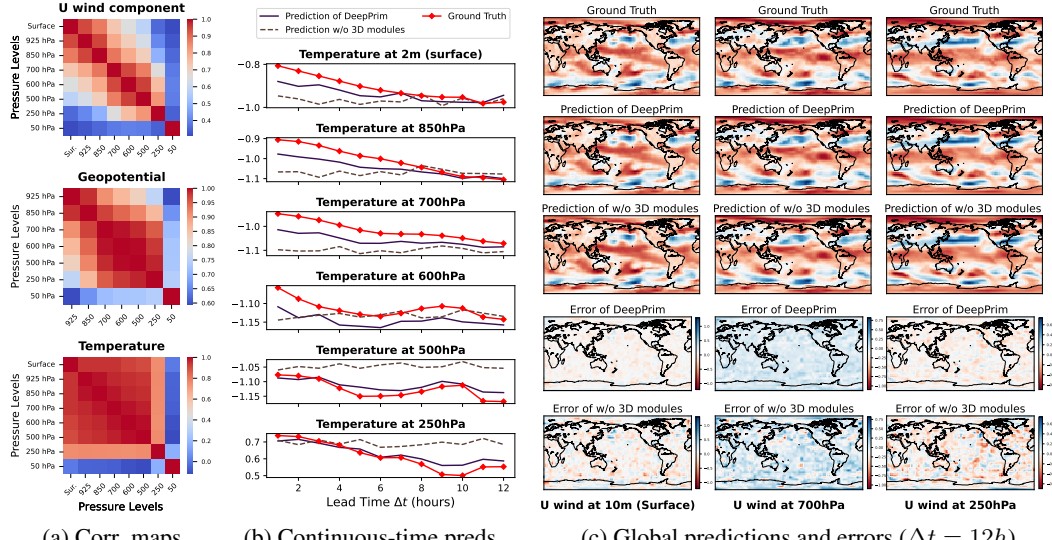

Figure 3: **Ablation studies of DeepPrim** based on the global weather forecasting task. The full results can be found in Table 20.

(a) Corr. maps     (b) Continuous-time preds.     (c) Global predictions and errors ($\Delta t = 12h$)

Figure 4: **Necessity and Effect of 3D atomphric dynamic modeling**. (a) Correlation maps of typical weather variables at different pressure levels based on ERA5 dataset; (b) Continuous-time predictions for temperature at different pressure levels of a specific location based on DeepPrim and its `w/o` 3D module version; (3) Global weather forecasting for U component of wind at different pressure levels.

DeepPrim can adaptively learn the boundary conditions in a data-driven manner and is more flexible. Following previous work (Verma et al., 2024), we focus on regional forecasting in three typical areas: North America, South America, and Australia. We summarize the comparison in Table 4 on Appendix A. It can be observed that DeepPrim consistently achieves the best forecasting performance for five variables across three regions, suppressing the second-best baseline ClimODE or ClimaX by $35.8\%$ on average in different tasks, which validates the superiority of DeepPrim in effectively capturing regional weather dynamics via 3D pressure-level aware modeling.

## 4.3 INVESTIGATION OF 3D ATMOSPHERIC DYNAMIC MODELING

As presented in Fig. 4 (a), some weather variables (e.g., wind speed, temperature) at different pressure levels show typical correlation patterns, with variables at more adjacent pressure levels showing higher correlations. As mentioned in Section 3.1, the correlation maps further emphasize the necessity of 3D atmospheric dynamic modeling.

**Ablation evidence** We conduct ablation studies (denoted as w/o 3D modules) by removing the 3D-related modules of DeepPrim, i.e., cross-pressure self-attention and the pressure level derivatives $\nabla_p \mathbf{u}$, and equally model the surface and the upper variables in the 2D coordinates. As shown in Fig. 3, removing the 3D modules would bring an average performance degradation of $12.3\%$.

**Visualization showcase** Fig. 4 (b) presents the continuous time predictions based on DeepPrim and its w/o 3D module version for a local grid with a lead time $\Delta t = 12h$. Remarkably, the temperature at different pressure levels presents typical correlated variations and generally shows downward trends due to the day-night cycle. Due to pressure-level-aware 3D modeling designs, *our DeepPrim successfully captures these correlations and models the variation trends of weather evolutions*

*more precisely* in comparison with its w/o 3D module version. Furthermore, Fig. 4 (c) highlights the improvement of predictions and errors on a global scale led by the 3D modeling strategy.

## 4.4 ABLATION STUDIES

**Investigation of initialization network** We adopt both the temporal variations $\dot{\mathbf{u}}$ and 3D spatial derivations $\nabla \mathbf{u}$ of initial conditions to estimate the initial intermediate motion fields $\mathbf{v}^*(t_0)$. According to Fig. 3, both the two terms can facilitate the estimation and bring performance improvements, and the inclusion of $\nabla \mathbf{u}$ enjoys better improvement.

**Effect of source-sink network** The results in Fig. 3 also emphasize the importance of the source-sink network, as it is vital to characterize the value loss or gain processes of weather states, e.g., day-night cycles, and it can also alleviate the error accumulation problem of the ODE systems.

## 5 CONCLUSION

In this paper, we introduce the 3D atmospheric modeling in a physics-driven deep-learning paradigm, and propose DeepPrim for short-term weather forecasting. Through learning primitive equations, DeepPrim effectively models atmospheric motion by portraying the force term in Navier-Stokes equation with 3D-BiViT and captures weather state evolution by leveraging the learned advection and source-sink model. In the future, we plan to extend DeepPrim with customized rolling training and inference strategies to improve long-term predictions and explore more applications by incorporating additional physical priors with rigorous stability and conservation analysis and advanced time-stepping methods.

## 6 ACKNOWLEDGEMENT

This work was supported by DAMO Academy through DAMO Academy Research Intern Program.

## 7 ETHICS STATEMENT

Our work focuses solely on scientific problems and does not involve human subjects, animals, or environmentally sensitive materials. We foresee no ethical risks or conflicts of interest.

## 8 REPRODUCIBILITY STATEMENT

We have rigorously formalized the model architecture through equations and illustrations in the main text. We involve the reproducibility details in Appendix, including algorithm implementations and hyperparameters (Appendix B.2), dataset descriptions (Appendix D), and evaluation metrics (Appendix E). We provide our source code in: `https://github.com/DAMO-DI-ML/DeepPrim`.

## 9 USE OF LLMS

The authors used LLM solely as a general-purpose assistive tool for grammar refinement and minor formatting suggestions. LLM did not contribute to research ideation, experimental design, data analysis, or interpretation, and all content was reviewed and is the full responsibility of the authors.

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

CONTENTS

## A    EXPERIMENT RESULTS ON REGIONAL WEATHER FORECASTING

Due to the spece limitation, here we summarize the experimental results of regional forecasting in Table 4.

Table 4: **Regional weather forecasting results** based on $5.625°$ ERA5 dataset with Latitude-weighted RMSE. "$\Delta$" denotes the relative error reduction of DeepPrim in comparison with the best baseline. The RMSE curve plots along lead time are presented in Fig. 11, Fig. 12, and Fig. 13.

| Variable | Lead-Time $\Delta t$ (h) | North-America | | | | | South-America | | | | | Australia | | | | |
|---|---|---|---|---|---|---|---|---|---|---|---|---|---|---|---|---|
| | | NODE (2018) | ClimaX (2023) | ClimODE (2024) | DeepPrim$^\dagger$ Ours | $\Delta$RMSE | NODE (2018) | ClimaX (2023) | ClimODE (2024) | DeepPrim$^\dagger$ Ours | $\Delta$RMSE | NODE (2018) | ClimaX (2023) | ClimODE (2024) | DeepPrim$^\dagger$ Ours | $\Delta$RMSE |
| z500 | 6 | 232.8 | 273.4 | 134.5 | **69.1** | ↓48.6% | 225.6 | 205.4 | 107.7 | **59.9** | ↓44.4% | 251.4 | 190.2 | 103.8 | **53.4** | ↓48.6% |
| | 12 | 469.2 | 329.5 | 225.0 | **121.5** | ↓46.0% | 365.6 | 220.2 | 169.4 | **85.6** | ↓49.5% | 344.8 | 184.7 | 170.7 | **70.3** | ↓58.8% |
| | 18 | 667.2 | 543 | 307.7 | **194.5** | ↓36.8% | 551.9 | 269.2 | 237.8 | **122.6** | ↓48.4% | 539.9 | 222.2 | 211.2 | **94.9** | ↓55.1% |
| | 24 | 893.7 | 494.8 | 390.1 | **277.5** | ↓28.9% | 660.3 | 301.8 | 292.0 | **166.3** | ↓43.0% | 632.7 | 324.9 | 308.2 | **117.3** | ↓61.9% |
| t850 | 6 | 1.96 | 1.62 | 1.28 | **0.80** | ↓37.5% | 1.58 | 1.38 | 0.97 | **0.72** | ↓25.8% | 1.37 | 1.19 | 1.05 | **0.60** | ↓42.9% |
| | 12 | 3.34 | 1.86 | 1.81 | **1.12** | ↓38.1% | 2.18 | 1.62 | 1.25 | **0.87** | ↓30.4% | 2.18 | 1.30 | 1.20 | **0.72** | ↓40.0% |
| | 18 | 4.21 | 2.75 | 2.03 | **1.22** | ↓39.9% | 2.74 | 1.79 | 1.43 | **1.02** | ↓28.7% | 2.68 | 1.39 | 1.33 | **0.85** | ↓36.1% |
| | 24 | 5.39 | 2.27 | 2.23 | **1.45** | ↓35.0% | 3.41 | 1.97 | 1.65 | **1.19** | ↓27.9% | 3.32 | 1.92 | 1.63 | **0.93** | ↓42.9% |
| t2m | 6 | 2.65 | 1.75 | 1.61 | **0.53** | ↓67.1% | 2.12 | 1.85 | 1.33 | **0.80** | ↓39.8% | 1.88 | 1.57 | 0.80 | **0.65** | ↓18.8% |
| | 12 | 3.43 | 1.87 | 2.13 | **0.69** | ↓63.1% | 2.42 | 2.08 | 1.04 | **0.92** | ↓11.5% | 2.02 | 1.57 | 1.10 | **0.76** | ↓30.9% |
| | 18 | 3.53 | 2.27 | 1.96 | **0.78** | ↓60.2% | 2.60 | 2.15 | **0.98** | 1.02 | ↑4.1 % | 3.51 | 1.72 | 1.23 | **0.82** | ↓33.3% |
| | 24 | 3.39 | 1.93 | 2.15 | **0.85** | ↓56.0% | 2.56 | 2.23 | 1.17 | **1.12** | ↓4.3% | 2.46 | 2.15 | 1.25 | **0.89** | ↓28.8% |
| u10 | 6 | 1.96 | 1.74 | 1.54 | **0.94** | ↓39.0% | 1.94 | 1.27 | 1.25 | **0.81** | ↓35.2% | 1.91 | 1.40 | 1.35 | **0.95** | ↓29.6% |
| | 12 | 2.91 | 2.24 | 2.01 | **1.43** | ↓28.9% | 2.74 | 1.57 | 1.49 | **0.97** | ↓34.9% | 2.86 | 1.77 | 1.78 | **1.16** | ↓34.5% |
| | 18 | 3.4 | 3.24 | 2.17 | **1.66** | ↓23.5% | 3.24 | 1.83 | 1.81 | **1.13** | ↓37.6% | 3.44 | 2.03 | 1.96 | **1.33** | ↓32.1% |
| | 24 | 3.96 | 3.14 | 2.34 | **2.08** | ↓11.1% | 3.77 | 2.04 | 2.08 | **1.33** | ↓34.8% | 3.91 | 2.64 | 2.33 | **1.52** | ↓34.8% |
| v10 | 6 | 2.16 | 1.83 | 1.67 | **1.00** | ↓40.1% | 2.29 | 1.31 | 1.30 | **0.86** | ↓33.8% | 2.38 | 1.47 | 1.44 | **1.00** | ↓30.6% |
| | 12 | 3.2 | 2.43 | 2.03 | **1.54** | ↓24.1% | 3.42 | 1.64 | 1.71 | **1.03** | ↓37.2% | 3.60 | 1.79 | 1.87 | **1.20** | ↓33.0% |
| | 18 | 3.96 | 3.52 | 2.31 | **1.91** | ↓17.3% | 4.16 | 1.90 | 2.07 | **1.20** | ↓36.8% | 4.31 | 2.33 | 2.23 | **1.35** | ↓39.5% |
| | 24 | 4.57 | 3.39 | 2.50 | **2.44** | ↓2.4% | 4.76 | 2.14 | 2.43 | **1.42** | ↓33.6% | 4.88 | 2.58 | 2.53 | **1.53** | ↓39.5% |

## B    IMPLEMENTATION DETAILS OF DEEPPRIM

### B.1    NOTATIONS

To clearly describe the task of weather forecasting and introduce the proposed approach, we first introduce some necessary notations in Table 5. Specifically, we use subscript $_S$ to denote surface variables, use subscript $_U$ to denote upper-air variables, and use subscript $_{sta}$ to denote static variables.

### B.2    ALGORITHM IMPLEMENTATIONS OF DEEPPRIM

We summarize the model feed-forward procedures of DeepPrim in Algorithm 1, where $K = D_S + PD_U$, $\dot{\mathbf{u}} = \frac{d\mathbf{u}}{dt}$ denotes time derivatives, $C$ denotes the channels of spatiotemporal embeddings. Also, Algorithm 2 summarizes the implementations of the force network based on 3D-BiViT, which is used for estimating the time derivatives of intermediate motion fields $\dot{\mathbf{v}}^*$.

For the input initial weather states $\mathbf{u}(t_{-2} : t_0)$ with time window length of 3, we first compute the 3D spatial derivatives $\nabla\mathbf{u}(t_0)$ along the longitude-latitude-presspre level coordinates. The initial states $\mathbf{u}(t_{-2} : t_0)$ and both their time derivatives $\dot{\mathbf{u}}(t_{-2} : t_0)$ and spatial derivatives $\nabla\mathbf{u}(t_0)$ along with the spatialtemporal embeddings $\psi_{ST}$ are fed into the initialization network to estimate the initial values of intermediate motion fields $\mathbf{v}^*(t_0)$. Then, both $\mathbf{u}(t_0)$ and $\mathbf{v}^*(t_0)$ are fed into the force-network-based ODE system, where $\dot{\mathbf{u}}(t_0)$ is calculated via advection equation and $\dot{\mathbf{v}}^*(t_0)$ is calculated based on 3D-BiViT. By solving the ODE system with numerical solvers, we can obtain the future weather predictions $\{\mathbf{u}'(t_i)\}_{i=1}^N$. Furthermore, the source model takes the inputs of initial state $\mathbf{u}(t_{-2} : t_0)$, initial intermediate motion fields $\mathbf{v}(t_0)$, ODE predictions $\{\mathbf{u}'(t_i)\}_{i=1}^N$ and spatiotemporal embeddings $\psi$ to characterize the external source-sink effects and correct the ODE predictions, output the final predictions $\{\hat{\mathbf{u}}(t_i)\}_{i=1}^N$.

### B.3    SPATIAL-TEMPORAL EMBEDDINGS

The spatial-temporal embeddings $\psi_{ST} \in \mathbb{R}^{C \times H \times W}$ are designed to inform the model of both longitude-latitude spatial location information and day-season time information. The spatial-

---

**Algorithm 1** DeepPrim as weather forecaster (Kindly note: for notational convenience and to be consistent with prior work, we flatten the variables across all pressure levels into a single dimension in our notation. Actually, $K = 1 \times D_S + P \times D_U$)

---

**Input:** Initial weather states $\mathbf{u}(t_{-2} : t_0) \in \mathbb{R}^{3 \times (K+3) \times H \times W}$ (include surface variables $\mathbf{u}_S(t_{-2} : t_0) \in \mathbb{R}^{3 \times 1 \times D_S \times H \times W}$, upper-air variables $\mathbf{u}_U(t_{-2} : t_0) \in \mathbb{R}^{3 \times P \times D_U \times H \times W}$, and static variables $\mathbf{u}_{sta} = \{\text{lsm,oro,lat}\} \in \mathbb{R}^{3 \times H \times W}$), discrete time steps $N$, the length of time step $\Delta t$, spatialtemporal embeddings $\psi_{ST}$.

**Output:** Future weather states $\{\hat{\mathbf{u}}(t_i)\}_{i=1}^{N} \in \mathbb{R}^{N \times K \times H \times W}$.

1: # Intermediate Motion Field Estimation

2: Obtain 3-dimensional spatial derivatives $\nabla \mathbf{u}(t_0) \in \mathbb{R}^{3 \times K \times H \times W}$

3: Estimate the initial intermediate motion fields:

4: $\mathbf{v}^*(t_0) = (v_x, v_y)_{t_0} + \text{Conv}(\underbrace{\mathbf{u}(t_{-2} : t_0), \dot{\mathbf{u}}(t_0)}_{\text{temporal variations}}, \underbrace{\nabla \mathbf{u}(t_0)}_{\text{spatial gradients}}, \psi_{ST}) \in \mathbb{R}^{2 \times K \times H \times W}$

5: # Force Network and ODE System

6: **for** $i$ in range($N$) time steps **do**

7: $\begin{bmatrix} \dot{\mathbf{u}}(t_i) \\ \dot{\mathbf{v}}^*(t_i) \end{bmatrix} = \begin{bmatrix} -\left(\mathbf{v}^*(t_i) \cdot \nabla_p \mathbf{u}(t_i) + w\frac{\partial \mathbf{u}(t_i)}{\partial p}\right) \\ -\left(\mathbf{v}^*(t_i) \cdot \nabla_p \mathbf{v}^*(t_i) + w\frac{\partial \mathbf{v}^*(t_i)}{\partial p}\right) + \text{Force}(\mathbf{u}, \mathbf{v}^*) \end{bmatrix}$

8: $\begin{bmatrix} \mathbf{u}(t_{i+1}) \\ \mathbf{v}^*(t_{i+1}) \end{bmatrix} = \begin{bmatrix} \mathbf{u}(t_i) + \Delta t \dot{\mathbf{u}}(t_i) \\ \mathbf{v}^*(t_i) + \Delta t \dot{\mathbf{v}}^*(t_i) \end{bmatrix}$

9: **end for**

10: Solve the ODE system and obtain the predictions $\{\mathbf{u}'(t_i)\}_{i=1}^{N} \in \mathbb{R}^{N \times K \times H \times W}$

11: # Source-sink Estimation and Correction

12: Obtain the source terms $\{\mathbf{u}_{\text{source-sink}}(t_i)\}_{i=1}^{N} = \text{Conv}\left(\{\mathbf{u}(t_0)\}, \mathbf{v}^*(t_0), \{\mathbf{u}'(t_i)\}_{i=1}^{N}, \psi_{ST}\right)$

13: Correct and obtain the final predictions $\{\hat{\mathbf{u}}(t_i)\}_{i=1}^{N} = \{\mathbf{u}_{\text{source-sink}}(t_i)\}_{i=1}^{N} + \{\mathbf{u}'(t_i)\}_{i=1}^{N}$

---

---

**Algorithm 2** Force Network for learning time derivative of intermediate motion field via 3D-BiViT

---

**Input:** Current weather states and their spatial derivatives, current intermediate motion fields, spatiotemporal embeddings and pressure-level embeddings $\Omega = \{\mathbf{u}\,(t_i)\,, \nabla \mathbf{u}\,(t_i)\,, \mathbf{v}^*(t_i), \psi_{ST}, \psi_P\}$.

**Output:** Time derivatives of intermediate motion fields $\dot{\mathbf{v}}^*(t_i)$.

1: # Patch Embedding
2: Get the inputs of surface variables $\Omega_S = [\mathbf{u}_S\,(t_i)\,, \mathbf{u}_{sta}, \nabla \mathbf{u}_S\,(t_i), \mathbf{v}_S^*(t_i), \psi_{ST}] \in \mathbb{R}^{(D_S+3+3D_S+2D_S+C)\times 1\times H \times W}$
3: Get the inputs of upper-air variables $\Omega_U = [\mathbf{u}_U\,(t_i)\,, \mathbf{u}_{sta}, \nabla \mathbf{u}_U\,(t_i), \mathbf{v}_U^*(t_i), \psi_{ST}] \in \mathbb{R}^{(D_U+3+3D_U+2D_U+C)\times P \times H \times W}$
4: Patch-wise token embedding of surface variables $\mathbf{Z}_S \leftarrow \text{Embed}_S\,(\Omega_S) \in \mathbb{R}^{1\times N_P \times d}$
5: Patch-wise token embedding of upper-air variables $\mathbf{Z}_U \leftarrow \text{Embed}_U\,(\Omega_U) \in \mathbb{R}^{P\times N_P \times d}$
6: Add pressure-level embeddings $\mathbf{Z}_S \leftarrow \mathbf{Z}_S + \psi_P[0:1,:,:], \mathbf{Z}_U \leftarrow \mathbf{Z}_U + \psi_P[1:P,:,:]$
7: # Intra-pressure-level Self-Attention
8: $\mathbf{Z}_S \leftarrow \text{Self-Attn}_S\left(\{\mathbf{Z}_S[:,i,:]\}_{i=1}^{N_P}\right)$
9: $\mathbf{Z}_U \leftarrow \text{Self-Attn}_U\left(\{\mathbf{Z}_U[:,i,:]\}_{i=1}^{N_P}\right)$
10: # Cross-pressure-level Self-Attention
11: Concat the embeddings $\mathbf{Z} \leftarrow [\mathbf{Z}_S, \mathbf{Z}_U] \in \mathbb{R}^{(P+1)\times N_P \times d}$
12: $\mathbf{Z} \leftarrow \mathbf{Z}.\texttt{transpose(0,1)} \in \mathbb{R}^{N_P \times (P+1)\times d}$
13: $\mathbf{Z} \leftarrow \text{Self-Attn}\left(\{\mathbf{Z}[:,i,:]\}_{i=1}^{P+1}\right)$
14: # Projection Head
15: Obtain the representations of surface variables $\mathbf{Z}_S \in \mathbb{R}^{1\times N_P \times d}$
16: Obtain the representations of upper-air variables $\mathbf{Z}_U \in \mathbb{R}^{P\times N_P \times d}$
17: Compute the intermediate motion fields' time derivatives of surface variables and upper-air variables.
18: $\dot{\mathbf{v}}_{\mathbf{S}}(t_i) = \text{Projector}_S(\mathbf{Z}_S), \dot{\mathbf{v}}_{\mathbf{U}}(t_i) = \text{Projector}_U(\mathbf{Z}_U))$.

---

Table 5: Notations and their meanings.

| Notations | Meanings |
|---|---|
| T | temperature |
| $v_x$ | latitude-direction wind |
| $v_y$ | longitude-direction wind |
| $\phi$ | geopotential |
| $q$ | humidity |
| $w$ | vertical wind speeed |
| $\nu$ | viscosity coefficient |
| $A$ | turbulent exchange coefficient |
| $f$ | Coriolis force coefficient |
| $x$-$y$-$p$ | longitude-latitude-pressure level coordinates |
| $H$ | number of latitude grids |
| $W$ | number of longitude grids |
| $D_S$ | number of surface variables |
| $D_U$ | number of upper-air variables at each pressure levels |
| $P$ | number of pressure levels |
| $K$ | number of total variables at all pressure levels $K = D_S + PD_U$ |
| $\mathbf{u}(t)$ | weather states at time $t$ |
| $\mathbf{v}^*(t)$ | intermediate motion fields at time $t$ |
| $\dot{\mathbf{u}}$ | time derivatives of weather states $\frac{d\mathbf{u}}{dt}$ |
| $\dot{\mathbf{v}}^*$ | time derivatives of intermediate motion fields $\frac{d\mathbf{v}^*}{dt}$ |
| $\nabla_P \mathbf{u}$ | horizontal gradients of weather states |
| $\nabla \mathbf{u}$ | 3D (horizontal and vertical/pressure-level) spatial gradients of weather states |
| $\nabla \cdot \mathbf{v}^*$ | divergence of intermediate motion fields |
| $N$ | number of discrete time steps in the ODE system |
| $\Delta t$ | the length of time interval/step in the ODE system |
| $\psi_{ST}$ | spatialtemporal embeddings |
| $\psi_P$ | Pressure-level embeddings |
| $C$ | the number of spatiotemporal embeddings' channels |
| $N_P$ | number of patches |
| $d$/Embed_dim | embedded dimensions |

temporal embeddings $\psi_{ST}$ are encoded as additional input channels to the model and consist of the following parts:

**Spatial Embedding**   Consider the spherical properties of the Earth, we encode latitude $h$ and longitude $w$ values with trigonometric and spherical coordinates as

$$\psi_s = [\sin(h), \cos(h), \sin(w), \cos(w), \sin(h)\cos(w), \sin(h)\sin(w)].$$

**Temporal Embedding**   Consider that daily and seasonal periodic patterns (e.g., day-night, winter-summer) are important for weather forecasting, we encode daily and seasonal cycles are using trigonometric functions as:

$$\psi_t = \left[\sin(2\pi t), \cos(2\pi t), \sin\left(\frac{2\pi t}{365}\right), \cos\left(\frac{2\pi t}{365}\right)\right],$$

where $t$ denotes the global time of the corresponding samples.

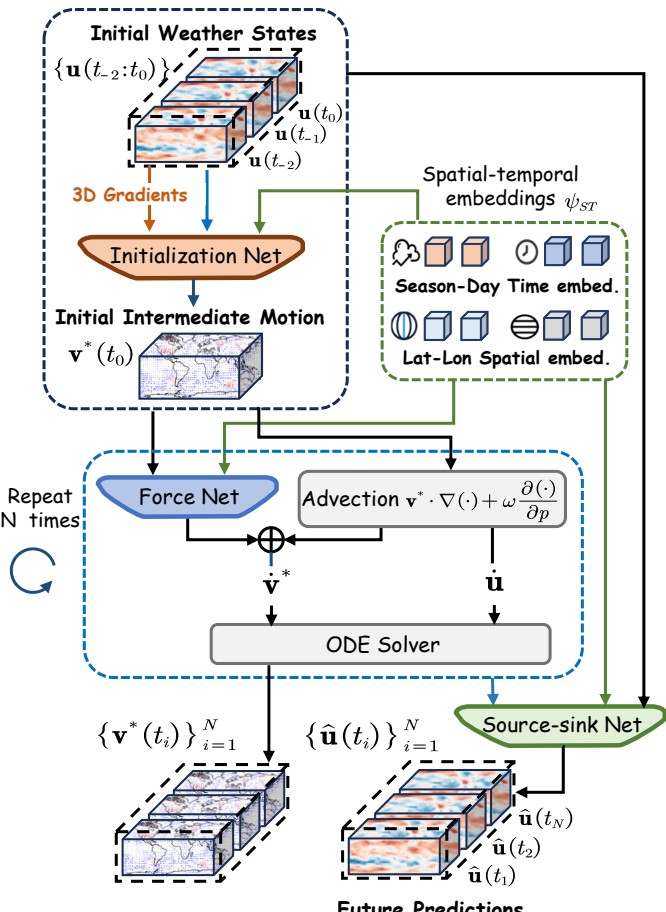

Figure 5: **Overall pipeline of DeepPrim Framework**

**Spatial-temporal Joint Embeddings**  To capture the cyclical spatial location and temporal periodic patterns jointly, we design spatial-temporal joint embeddings as

$$\psi_{st} = \psi_s \circ \psi_t,$$

where $\circ$ denote the Hadamard product.

Finally, the overall spatial-temporal embeddings integrates all the aforementioned terms as:

$$\psi_{ST} = [\psi_s, \psi_t, \psi_{st}].$$

### B.4 MODEL ARCHITECTURES AND HYPER-PARAMETERS

In our DeepPrim, the backbones of the initialization network and the source-sink network are based on ResNet (He et al., 2016), and the backbone of the force network is based on ViT (Dosovitskiy et al., 2021). The hyperparameter configurations of the initialization model, the force network, and the source-sink model are summarized in Table 6, Table 7, and Table 8, respectively.

### B.5 DIFFERENCE BETWEEN PANGU, NEURALGCM, AND DEEPPRIM IN 3D ATMOSPHERIC DYNAMICS MODELING

As pioneering methods, both Pangu and NeuralGCM emphasize the importance of 3D atmospheric modeling with customized techniques. In this section, we'll discuss the difference between Pangu, NeuralGCM, and DeepPrim in 3D atmospheric dynamics modeling as summarized in Table 9.

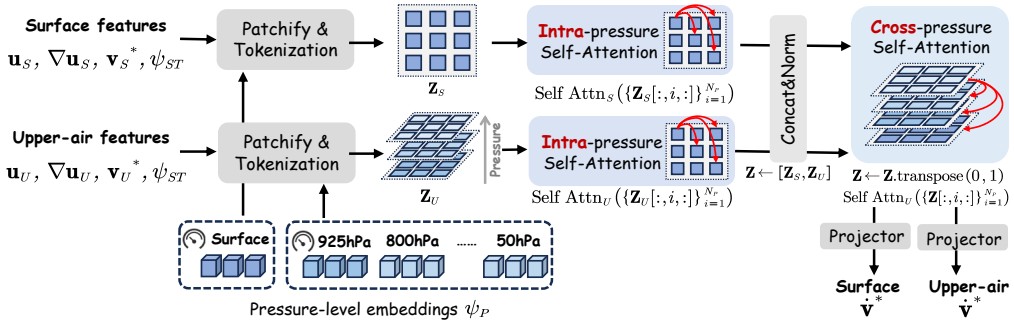

Figure 6: **Architecture of Force Network** that composes of 3D-BiViT. The force network encapsulates external forces (including pressure gradients, Coriolis forces, and frictional effects) by modeling the time deviation of intermediate motion for upper-air and surface inspired by the force term of the Navier-Stokes equation. The feature forward steps involve (1). tokenization and embedding, (2). pressure-Level embedding, and (3). Intra– and cross-pressure level attentions.

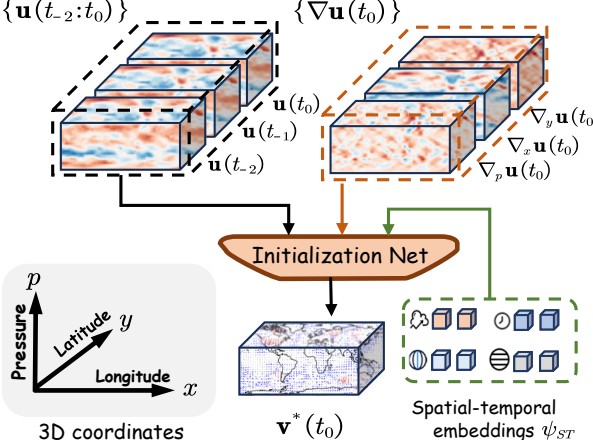

Figure 7: **Architecture of Initilization Network Model**. The initialization network integrates various additional features for initial intermediate motion estimation. The input features include spatial-temporal derivatives of various weather variables, land-sea mask, incoming solar radiation, orography, and spatial-temporal embeddings $\psi_{ST}$.

Specifically, Pangu leverages Unified attention blocks and positional embedding to model pressure information, which may not explicitly distinguish heterogeneous horizontal-vertical interactions. NeuralGCM employs individual neural networks to model physical tendencies in the corresponding single vertical column, which may omit the inference of intra-pressure-level influences. In comparison, our DeepPrim explicitly couples pressure-level dynamics with primitive equations and uses separate attentions for horizontal and cross-pressure level interactions rather than solely relying on

Table 6: Default hyperparameters of the initialization network.

| Hyperparameter | Description | Value |
|---|---|---|
| Kernel size | Size of each convolutional kernels | 3 |
| Padding size | Size of padding of each convolutional layer | 1 |
| Stride | Step size of each convolutional layer | 1 |
| Dropout | Dropout probability | 0.1 |
| Leakage Coefficient | Slope of LeakyReLU for negative inputs | 0.3 |
| ResBlock List | (Number of ResBlocks, Hidden dimensions) | $[(5, 256), (3, 64), (2, \text{out channels})]$ |

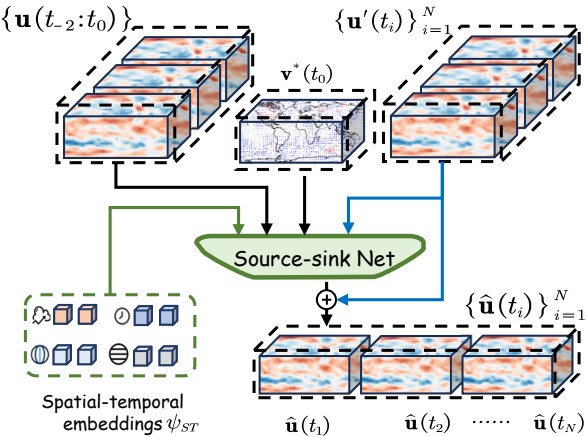

Figure 8: **Architecture of Source-Sink Network Model** that captures the value gains or losses of each variable caused by resolved and unsolved processes for final predictions. The source-sink network receives the initial weather states $\mathbf{u}(t_0)$ and initial intermediate motion $\mathbf{v}^*(t_0)$, and ODE solutions $\{\mathbf{u}'(t_i)\}_{i=1}^{N}$ along with spatiotemporal embeddings $\psi_{ST}$ as input to correct the output of ODE solutions.

Table 7: Default hyperparameters of the ViT in 3D force network. Kindly note the patch size $p_{\text{size}}$ is set to 2 for $5.625°$ model, is set to 4 for $1.40625°$ model, and is set to 24 for $0.25°$ model.

| Hyperparameter | Description | Value |
|---|---|---|
| $p_{\text{size}}$ | Patch size | 2/4/24 |
| $D$ | Dimension of hidden layers | 128 |
| $L_{\text{Surface\_inter}}$ | Number of Transformer blocks of inter-pressure level self-attentions for surface variables | 1 |
| $L_{\text{Upper\_inter}}$ | Number of Transformer blocks of inter-pressure level self-attentions for upper-air variables | 1 |
| $L_{\text{Intra}_p\text{ressure}}$ | Number of Transformer blocks of intra-pressure level self-attentions | 1 |
| Heads | Number of attention heads | 8 |
| MLP ratio | Expansion factor for MLP | 4 |
| Decoder Depth | Number of layers of the final prediction head | 2 |
| Drop path | Stochastic depth rate | 0.1 |
| Dropout | Dropout rate | 0.1 |

positional biases. Thus, DeepPrim enables more flexible and physically consistent 3D atmospheric dynamics modeling capacity, as we elaborately demonstrated in the Section 4.3.

## B.6 NUMBER OF MODEL PARAMETERS AND COMPARISON WITH BASELINES

We summarize the parameter number of DeepPrim and other advanced deep weather forecasting approaches in Table 10. It can be seen that DeepPrim generally maintains great lightweight designs with a relatively small number of parameters in comparison with most baselines, and also

Table 8: Default hyperparameters of the source-sink model.

| Hyperparameter | Description | Value |
|---|---|---|
| Kernel size | Size of each 3D convolutional kernels | 3 |
| Padding size | Size of padding of each 3D convolutional layer | 1 |
| Stride | Step size of each 3D convolutional layer | 1 |
| Dropout | Dropout probability | 0.1 |
| Leakage Coefficient | Slope of LeakyReLU for negative inputs | 0.3 |
| ResBlock List | (Number of ResBlocks, Hidden dimensions) | $[(5, 256), (3, 64), (2, \text{out channels})]$ |

Table 9: Comparison with Pangu, NeuralGCM, and DeepPrim in 3D atmospheric dynamics modeling in the perspectives of key ideas and technical designs.

| Aspect | Pangu (2023) | NeuralGCM (2024) | DeepPrim (Ours) |
|---|---|---|---|
| Key ideas | Treats pressure levels as geometric height equivalents in positional embeddings, may not explicitly distinguish heterogeneous interactions | Feeds data for individual columns of the atmosphere into a neural network used to produce corresponding physics tendencies | Leverages intra-pressure-level attention for horizontal interactions and cross-pressure-level attention for vertical coupling separately yet collaboratively |
| Technical Designs | Unified attention blocks for lat-lon-pressure coordinates (implicit horizontal/vertical interaction blending) | Neural networks that act locally in space, on individual vertical columns of the atmosphere. | Physics-informed modular design explicitly separates intra-pressure-layer horizontal dynamics and inter-pressure-layer vertical interactions |

presents superior performance in both global and regional weather forecasting tasks. Compared with ClimODE, our DeepPrim further exploits 3D atmospheric dynamics and integrates a parametrized initialization network for motion velocity estimation. The resulting performance gains and 3D atmospheric modeling capabilities over ClimODE require acceptable additional parameters.

.

Table 10: Parameter number of DeepPrim and advanced deep weather forecasting approaches. "Dataset Resolution" denotes the resolution of the ERA5 dataset used in experiments.

| | GraphCast (Lam et al., 2023) | Pangu-Weather (Bi et al., 2023) | ClimaX (Nguyen et al., 2023) | ClimaX (Nguyen et al., 2023) | ClimODE (Verma et al., 2024) | WeatherGFT (Xu et al., 2024) | NeuralGCM (Kochkov et al., 2024) | DeepPrim Ours | DeepPrim Ours | DeepPrim Ours |
|---|---|---|---|---|---|---|---|---|---|---|
| Dataset Resolution | 0.25° | 0.25° | 5.625° | 1.40625° | 5.625° | 1.40625° | 1.40625° | 5.625° | 1.40625° | 0.25° |
| Parameters (Million) | 37M | 256M | 108M | 109M | 2.8M | 470M | 18.3M | 22M | 23M | 45M |

## B.7 Optimization Configurations

We conduct all the experiments with PyTorch[3]. In the training stage, we choose the batch size of 8, and run experiments on 8 NVIDIA A800-SXM4-80GB GPUs. In the ODE system, we set time interval $\Delta t = 1h$. We use the AdamW optimizer with $\beta_1 = 0.9$, $\beta_2 = 0.999$. The learning rate is set to $1e\text{-}5$ for the ODE model components and $5e\text{-}4$ for the rest. A weight decay of $1e\text{-}5$ is applied to all parameters except for the positional embeddings. The learning rate follows a linear warmup schedule starting from $1e\text{-}8$ for the first $20,000$ steps (approximately 1 epoch), transitioning to a cosine-annealing schedule for the remaining $90,000$ steps (approximately 9 epochs), with a minimum value of $1e\text{-}8$. The maximum epoch is set as 50, and we use an early-stopping strategy with a patience value of 3.

## B.8 Training and Inference time

We summarize the Inference time of DeepPrim in Table 11 and the training time cost of DeepPrim in Table 12. In general, the training time is affordable. Also, the inference speed of DeepPrim is appreciable due to its lightweight design.

## C Primitive Equations of the Atmosphere

The primitive equations, which form the foundation of numerical weather prediction (NWP), consist of a set of nonlinear partial differential equations governing atmospheric dynamics and thermodynamics. These equations describe the conservation of momentum, mass, energy, and water vapor in the atmosphere. Below, we present the complete set of primitive equations in pressure coordinates ($x$-$y$-$p$), along with detailed explanations of each term.

---

[3]https://pytorch.org/

Table 11: **Inference time of DeepPrim** for each sample at different forecasting lead times based on a single NVIDIA A800-SXM4-80GB GPU. The evaluation metric is seconds (s).

| Resolution | Lead time (hours) | | | |
|---|---|---|---|---|
| | $\Delta t = 6$ | $\Delta t = 12$ | $\Delta t = 18$ | $\Delta t = 24$ |
| DeepPrim(5.625°) | 0.16s | 0.33s | 0.47s | 0.66s |
| DeepPrim(1.4°) | 0.64s | 1.23s | 1.81s | 2.31s |
| DeepPrim(0.25°) | 2.48s | 4.98s | 7.47s | 9.48s |

Table 12: **Training time of DeepPrim** at different forecasting lead times. The maximum epoch is set as 50, and we use an early-stopping strategy with a patience of 3. Please refer to Section B.7 and Section B.4 for more optimization configurations.

| Resolution | Hardware | Lead time (hours) | | | |
|---|---|---|---|---|---|
| | | $\Delta t = 6$ | $\Delta t = 12$ | $\Delta t = 18$ | $\Delta t = 24$ |
| DeepPrim(5.625°) | 4 NVIDIA A800-SXM4-80GB GPUs | 13hours | 1day2hours | 1day7hours | 1day19hours |
| DeepPrim(1.4°) | 4 NVIDIA A800-SXM4-80GB GPUs | 5days | 6days | 6days | 7days |
| DeepPrim(0.25°) | 8 NVIDIA A800-SXM4-80GB GPUs | 8days | - | - | 15days |

Kindly Note: Due to computational resource limitations, we did not directly train DeepPrim(0.25°) models with $\Delta t = 12$ and $\Delta t = 18$. Instead, we obtained the results by rolling the DeepPrim(0.25°) model of $\Delta t = 6$. For example, we obtained the results of $\Delta t = 12$ by rolling the DeepPrim(0.25°) model of $\Delta t = 6$ twice.

### C.1 MOMENTUM EQUATION

The horizontal momentum equation (Navier-Stokes equation) describes the evolution of the horizontal wind vector $\mathbf{v} = (v_x, v_y)$ under the influence of the Coriolis force, pressure gradient force, and horizontal force (we ):

$$\frac{d\mathbf{v}}{dt} + f\mathbf{k} \times \mathbf{v} = -\nabla_p \phi + \mathbf{F}_h,$$

$$\mathbf{F}_h = \nu \nabla_p^2 \mathbf{v} + A\frac{\partial^2 \mathbf{v}}{\partial p^2},$$

where:

- $\mathbf{v} = (v_x, v_y)$: Horizontal wind velocity vector ($v_x$: latitude-direction wind, $v_y$: lontitude-direction wind).
- $f = 7.29 \times 10^{-5}\,\mathrm{s}^{-1}$: Coriolis force coefficient.
- $\mathbf{k}$: Unit vector in the vertical direction.
- $\phi = gz$: Geopotential ($g$: gravitational acceleration, $z$: geometric height).
- $\nabla_p = \left(\frac{\partial}{\partial x}, \frac{\partial}{\partial y}\right)$: Horizontal gradient operator in pressure coordinates.
- $\mathrm{F}_h$: Horizontal Force.
- $\nu$: Horizontal viscosity coefficient.
- $A$: Vertical turbulent exchange coefficient.

The material derivative $\frac{d}{dt}$ is expanded as:

$$\frac{d}{dt} = \frac{\partial}{\partial t} + \mathbf{v} \cdot \nabla_p + \omega \frac{\partial}{\partial p},$$

where $\omega = \frac{\mathrm{d}p}{\mathrm{d}t}$ is the vertical velocity in pressure coordinates.

## C.2 CONTINUITY EQUATION

The continuity equation expresses the conservation of mass in the atmosphere:

$$\frac{\partial \omega}{\partial p} + \nabla_p \cdot \mathbf{v} = 0.$$

Integrating this equation vertically yields the vertical velocity $\omega$:

$$\omega = -\int_{p_{\text{top}}}^{p} \nabla_p \cdot \mathbf{v} \, \mathrm{d}p,$$

where $p_{\text{top}}$ is the pressure at the model top.

## C.3 THERMODYNAMIC EQUATION

The thermodynamic equation describes the evolution of temperature $T$ due to adiabatic processes and diabatic heating:

$$\frac{\mathrm{d}T}{\mathrm{d}t} - \frac{RT}{c_p p}\omega = \frac{Q}{c_p} + K_T \nabla_p^2 T + K_v \frac{\partial^2 T}{\partial p^2},$$

where:

- $T$: Temperature.
- $R = 287 \, \mathrm{J\,kg^{-1}K^{-1}}$: Gas constant for dry air.
- $c_p = 1004 \, \mathrm{J\,kg^{-1}K^{-1}}$: Specific heat capacity at constant pressure.
- $Q$: Diabatic heating rate (e.g., radiative heating, latent heat release).
- $K_T$: Horizontal thermal diffusivity.
- $K_v$: Vertical thermal diffusivity.

## C.4 WATER VAPOR EQUATION

The conservation of water vapor is governed by the following equation:

$$\frac{\mathrm{d}q}{\mathrm{d}t} = S_q + K_q \nabla_p^2 q + K_v \frac{\partial^2 q}{\partial p^2},$$

where:

- $q$: Specific humidity (water vapor mixing ratio).
- $S_q$: Sources and sinks of water vapor (e.g., evaporation, condensation). Under saturated pseudo-adiabatic conditions, $S_q$ can be computed as:
$$\begin{cases} S_q = \frac{\delta F}{RT} \frac{\mathrm{d}\phi}{\mathrm{d}t}, \\ \delta = \begin{cases} 0, \frac{\mathrm{d}\phi}{\mathrm{d}t} < 0 \text{ and } q \geq q_s \\ 1, \text{else} \end{cases} \\ F = q_s T \frac{LR - c_p R_v T}{c_p R_v T^2 + L^2 q_s} \\ e_s = 6.112 \times \exp\left(\frac{17.67 T'}{T' + 243.5}\right) \\ T' = T - 273.15 \\ q_s = \frac{0.622 e_s}{p - 0.378 e_s} \end{cases}$$
- $K_q$: Horizontal diffusivity for water vapor.
- $K_v$: Vertical diffusivity for water vapor.

## C.5 HYDROSTATIC BALANCE EQUATION

The hydrostatic balance equation relates the vertical gradient of geopotential $\phi$ to the temperature $T$:

$$\frac{\partial \phi}{\partial p} = -\frac{RT}{p}.$$

## C.6 IDEAL GAS LAW

The ideal gas law provides the relationship between pressure, density, and temperature:

$$p = \rho RT,$$

where $\rho$ is the air density.

## C.7 SUMMARY OF SYMBOLS

- **Coordinates**: $x$ (longitude), $y$ (latitude), $p$ (pressure).
- **Time**: $t$.
- **Wind**: $\mathbf{v} = (v_x, v_y)$ (horizontal), $\omega$ (vertical in pressure coordinates).
- **Thermodynamic Variables**: $T$ (temperature), $q$ (specific humidity), $\phi$ (geopotential).
- **Physical Constants**: $f$ (Coriolis parameter), $R$ (gas constant), $c_p$ (specific heat capacity).
- **Diffusivity Coefficients**: $\nu$ (horizontal viscosity), $A$ (vertical viscosity), $K_T$ (thermal diffusivity), $K_q$ (water vapor diffusivity).

## C.8 THE DERIVED EQUATIONS AND ANALYSIS

This set of equations forms the basis for atmospheric modeling and numerical weather prediction, as described in (Kalnay, 2003), (Bauer et al., 2015), (Vallis, 2017), and (Holton & Hakim, 2013). The resulting equations of $T$, $q$, and $\phi$ are:

$$\frac{\partial T}{\partial t} = \frac{-L\frac{\partial \phi}{\partial p}w - \frac{1}{\rho}w}{c_p} - v_x\frac{\partial T}{\partial x} - v_y\frac{\partial T}{\partial y} - w\frac{\partial T}{\partial p},$$

$$\frac{\partial q}{\partial t} = \frac{\delta F}{RT}\left(\frac{\partial \phi}{\partial t} + v_x\frac{\partial \phi}{\partial x} + v_y\frac{\partial \phi}{\partial y} + w\frac{\partial \phi}{\partial p}\right) - v_x\frac{\partial q}{\partial x} - v_y\frac{\partial q}{\partial y} - w\frac{\partial q}{\partial p},$$

$$\frac{\partial \phi}{\partial t} = -\int \frac{R}{p}\frac{\partial T}{\partial t}\,\mathrm{d}p.$$

Although these variables are interconnected, the momentum equation that describes atmospheric motion serves as the cornerstone (Bauer et al., 2015), as the variations of other weather variables (such as temperature and water vapor) are primarily driven by horizontal advection within the atmospheric circulation scale (i.e., the advection term has a larger magnitude than other terms in thermodynamic equation and water vapor equation) (Emanuel, 1994). Therefore, accurately modeling atmospheric motion is crucial for forecasting these prognostic variables. As a result, in Deep-Prim, we **first introduce a 3D force network together with an initialization network to simulate the Navier-Stokes Equation**. Moreover, in NWP, physical processes are modeled under moderate simplifications. For example, in most general circulation models, temperature is constrained by a pseudo-adiabatic system considering moist processes such as condensation and evaporation, while other processes, including heat originating from friction, radiative heating, and cooling, remain unsolved (Randall et al., 2007; Stocker, 2014). To describe these processes, parameterization methods are introduced to approximate the interactions between resolved and unresolved processes. It is important to note that these parameterization methods are not unique; they can be customized and adjusted within different numerical schemes based on human heuristics. The choice of parameter types and configurations can lead to significantly different simulation results. Conversely, deep neural networks offer a way to bypass these simplifications by directly learning unresolved processes

through a data-driven approach. Thus, in DeepPrim, we **additionally apply a source-sink network to parameterize both the resolved and unresolved processes**.

# D DATASET DETAILS AND EXPERIMENTAL SETTINGS

In our experiments, we use the preprocessed ERA5 data from WeatherBench (Rasp et al., 2020). EAR5 is a well-acknowledged weather forecasting benchmark dataset and it is widely used in data-driven weather forecasting methods.

WeatherBench regridded the raw ERA5 dataset[4] from its $0.25°$ resolution to $5.625°$, $2.8125°$, and $1.40625°$ resolutions. To comprehensively evaluate our model, **we choose the ERA datasets with $5.625°$ resolution, the datasets with $1.40625°$, and the datasets with $0.25°$ resolution for performance evaluation**. The processed dataset includes 8 atmospheric variables across 13 pressure levels, 6 surface variables, and 5 static variables. We normalize all the inputs via z-score normalization for each variable at each pressure level. Also, we apply the inverse normalization for the predictions of future states for performance evaluation.

## D.1 ERA5 DATASET WITH $5.625°$ RESOLUTION

In line with (Nguyen et al., 2023), we selected 6 atmospheric variables at **7 pressure levels**, 3 surface variables, and 3 static variables for the ERA5 dataset with $5.625°$ resolution, as detailed in Table 14. In our model training, we choose all variables as input variables, and all variables except three static variables as output variables that are used for loss calculation. For performance evaluation, we focus on five key target variables in line with previous work (Verma et al., 2024), including geopotential at 500 hPa ($z500$), temperature at 850 hPa ($t850$), temperature at 2 meters ($t2m$), and zonal wind speeds at 10 meters ($u10$ and $v10$).

Table 13: Summary of ECMWF variables utilized in the ERA5 dataset with $5.625°$ resolution. The variables $lsm$ and $oro$ are constant and invariant with time. "Abbrev." denotes abbreviation.

| Type | Variable Name | Abbrev. | Description | Pressure Levels |
|---|---|---|---|---|
| Static Variable | Land-sea mask | $lsm$ | Binary mask distinguishing land (1) from sea (0) | N/A |
| | Orography | $oro$ | Height of Earth's surface | N/A |
| | Latitude | $lat$ | Latitude of each grid point | N/A |
| Surface Variable | 2 metre temperature | $t2m$ | Temperature measured 2 meters above the surface | Single level |
| | 10 metre U wind component | $u10$ | East-west wind speed at 10 meters above the surface | Single level |
| | 10 metre V wind component | $v10$ | North-south wind speed at 10 meters above the surface | Single level |
| Upper-air Variable | Geopotential | $z$ | Height relative to a pressure level | $50, 250, 500, 600, 700, 850, 925$ hPa |
| | U wind component | $u$ | Wind speed in the east-west direction | $50, 250, 500, 600, 700, 850, 925$ hPa |
| | V wind component | $v$ | Wind speed in the north-south direction | $50, 250, 500, 600, 700, 850, 925$ hPa |
| | Temperature | $t$ | Atmospheric temperature | $50, 250, 500, 600, 700, 850, 925$ hPa |
| | Specific humidity | $q$ | Mixing ratio of water vapor to total air mass | $50, 250, 500, 600, 700, 850, 925$ hPa |
| | Relative humidity | $r$ | Humidity relative to saturation | $50, 250, 500, 600, 700, 850, 925$ hPa |

## D.2 ERA5 DATASET WITH $1.40625°$ AND $0.25°$ RESOLUTION

In line with (Xu et al., 2024), we selected 6 atmospheric variables at all **13 pressure levels**, 3 surface variables, and 3 static variables for the ERA5 dataset with $1.40625°$ resolution, as detailed in Table 14. In our model training, we choose all variables as input variables, and all variables except three static variables as output variables that are used for loss calculation. For performance evaluation, we focus on five key target variables in line with previous work (Verma et al., 2024), including geopotential at 500 hPa ($z500$), temperature at 850 hPa ($t850$), temperature at 2 meters ($t2m$), and zonal wind speeds at 10 meters ($u10$ and $v10$).

---

[4]More details of ERA5 data can be found in `https://confluence.ecmwf.int/display/CKB/ERA5%3A+data+documentation`.

Table 14: Summary of ECMWF variables utilized in the ERA5 dataset with $1.40625°$ and $0.25°$ resolution. The variables $lsm$ and $oro$ are constant and invariant with time.

| Type | Variable Name | Abbrev. | Description | Pressure Levels |
|---|---|---|---|---|
| Static Variable | Land-sea mask | $lsm$ | Binary mask distinguishing land (1) from sea (0) | N/A |
| | Orography | $oro$ | Height of Earth's surface | N/A |
| | Latitude | $lat$ | Latitude of each grid point | N/A |
| Surface Variable | 2 metre temperature | $t2m$ | Temperature measured 2 meters above the surface | Single level |
| | 10 metre U wind component | $u10$ | East-west wind speed at 10 meters above the surface | Single level |
| | 10 metre V wind component | $v10$ | North-south wind speed at 10 meters above the surface | Single level |
| Upper-air Variable | Geopotential | $z$ | Height relative to a pressure level | 50, 100,150, 200, 250,300, 400, 500, 600, 700, 850, 925,1000 hPa |
| | U wind component | $u$ | Wind speed in the east-west direction | 50, 100,150, 200, 250,300, 400, 500, 600, 700, 850, 925,1000 hPa |
| | V wind component | $v$ | Wind speed in the north-south direction | 50, 100,150, 200, 250,300, 400, 500, 600, 700, 850, 925,1000 hPa |
| | Temperature | $t$ | Atmospheric temperature | 50, 100,150, 200, 250,300, 400, 500, 600, 700, 850, 925,1000 hPa |
| | Specific humidity | $q$ | Mixing ratio of water vapor to total air mass | 50, 100,150, 200, 250,300, 400, 500, 600, 700, 850, 925,1000 hPa |
| | Relative humidity | $r$ | Humidity relative to saturation | 50, 100,150, 200, 250,300, 400, 500, 600, 700, 850, 925,1000 hPa |

Table 15: Latitudinal and longitudinal boundaries with grid size for each region based on the ERA5 dataset with $5.625$ resolution.

| Region | Latitude Range | Longitude Range | Grid Size (lat x lon) |
|---|---|---|---|
| North America | $(15, 65)$ | $(220, 300)$ | $8 \times 14$ |
| South America | $(-55, 20)$ | $(270, 330)$ | $14 \times 10$ |
| Australia | $(-50, 10)$ | $(100, 180)$ | $10 \times 14$ |
| Global | $(-90, 90)$ | $(0, 360)$ | $32 \times 64$ |

### D.3 EXPERIMENTAL SETTING OF REGIONAL FORECASTING

In practice, it is not always necessary or possible to make global weather forecasting, especially when we cannot have access to global data. Instead, making regional weather forecasting based on specific local regional data is also critical. In line with the experimental setting of ClimaX (Nguyen et al., 2023) and ClimODE (Verma et al., 2024), we consider regional weather forecasting for three typical regions, including North America, South America, and Australia, based on ERA datasets with $5.625°$ resolution. The detailed information is summarized in Table 15.

## E EVALUATION METRICS FOR WEATHER FORECASTING

This section provides detailed explanations of all the evaluation metrics for weather forecasting used in the main experiments. For each metric, $u$ and $\tilde{u}$ represent the predicted and ground truth values, respectively, both shaped as $K \times H \times W$, where $K$ is the number of total weather factors ($K = (P + 1) \times D$, where $P$ is the number of pressure levels, $D$ is the number of weather factors), and $H \times W$ is the spatial resolution of latitude ($H$) and longitude ($W$). To account for the non-uniform grid cell areas, the latitude weighting term $\alpha(\cdot)$ is introduced.

**Latitude-weighted Root Mean Square Error (RMSE)** assesses model accuracy while considering the Earth's curvature. The latitude weighting adjusts for the varying grid cell areas at different latitudes, ensuring that errors are appropriately measured. Lower RMSE values indicate better model performance.

$$\text{RMSE} = \frac{1}{K} \sum_{k=1}^{K} \sqrt{\frac{1}{HW} \sum_{h=1}^{H} \sum_{w=1}^{W} \alpha(h) \left(\tilde{u}_{k,h,w} - u_{k,h,w}\right)^2}, \ \alpha(h) = \frac{\cos(\text{lat}(h))}{\frac{1}{H} \sum_{h'=1}^{H} \cos\left(\text{lat}\left(h'\right)\right)}.$$

**Anomaly Correlation Coefficient (ACC)** measures a model's ability to predict deviations from the mean. Higher ACC values indicate better accuracy in capturing anomalies, which is crucial in meteorology and climate science.

$$\text{ACC} = \frac{\sum_{k,h,w} \tilde{u}'_{k,h,w} u'_{k,h,w}}{\sqrt{\sum_{k,h,w} \alpha(h)(\tilde{u}'_{k,h,w})^2 \sum_{k,h,w} \alpha(h)(u'_{k,h,w})^2}},$$

where $u' = u - C$ and $\tilde{u}' = \tilde{u} - C$, with $C = \frac{1}{K} \sum_k \tilde{u}_k$ representing the temporal mean of the ground truth over the test set.

## F    IMPLEMENTATIONS OF BASELINES AND IFS

To make fair and comprehensive experimental validations, our experiments strictly adhere to **Resolution-matched "Apples-to-Apples" comparison principle**. Specifically, different methods may focus on different tasks (global or regional forecasting) and dataset with different resolutions (ranging from 0.25° to 5.625°). We first report the results from the original paper. Furthermore, we also replicated the Pangu and GraphCast models based on their official codebase[5][6] to conduct fair comparisons at a resolution of 5.625 degrees in Table 2.

### F.1    DETAILS OF BASELINES AND THE RESULT SOURCES

We summarize the details of baselines and the result sources of Table 2 and Table 3 in Table 16, including the original resolution setting of each method and the detailed result sources.

Table 16: **Details of Baselines and the Result Sources** of Table 2 and Tablle. 3. "Resolution" denotes the resolutions adopted in our experimental setting. "Original Resolution" denotes the resolution used in the original paper of the corresponding method.

| Resolution | Baselines | Paper | Original Resolution | Result Sources |
|---|---|---|---|---|
| 5.625° (Table 2) | IFS | (2023) | 5.625° (downsampled) | Cited from the paper of ClimaX (Nguyen et al., 2023) |
| | NODE | (2018) | 5.625° | Cited from the paper of ClimODE (Verma et al., 2024) |
| | FCN | (2022) | 5.625° | Cited from the paper of ClimODE (Verma et al., 2024) |
| | ClimaX* | (2023) | 5.625° | Cited from the paper of ClimODE (Verma et al., 2024) (without pre-training with CMIP6) |
| | ClimaX | (2023) | 5.625° | Cited from the paper of ClimaX (Nguyen et al., 2023) (pre-trained with CMIP6) |
| | GraphCast (5.625°) | (2023) | 0.25° | Reproduced from the official codes `https://github.com/google-deepmind/graphcast` |
| | Pangu (5.625°) | (2023) | 0.25° | Reproduced from the official codes `https://github.com/198808xc/Pangu-Weather` |
| | ClimODE | (2024) | 5.625° | Cited from the paper of ClimODE (Verma et al., 2024) |
| 1.40625° (Table 2) | IFS | (2023) | 1.40625° (downsampled) | Cited from the paper of ClimaX (Nguyen et al., 2023) |
| | NeuralGCM | (2024) | 1.40625° | Obtained from the official trained model checkpoint: `https://neuralgcm.readthedocs.io/en/latest/checkpoints.html` |
| | ClimaX | (2023) | 1.40625° | Cited from the paper of ClimaX (Nguyen et al., 2023) |
| | WeatherGFT | (2024) | 1.40625° | Cited from the paper of WeatherGFT (Xu et al., 2024) |
| 0.25° (Table 3) | IFS | (2023) | 0.25° | Obtained from WeatherBench2 `https://sites.research.google/weatherbench/deterministic-scores/` |
| | Pangu (0.25°) | (2023) | 0.25° | Obtained from WeatherBench2 `https://sites.research.google/weatherbench/deterministic-scores/` |
| | GraphCast | (2023) | 0.25° | Obtained from WeatherBench2 `https://sites.research.google/weatherbench/deterministic-scores/` |

---

[5]`https://github.com/google-deepmind/graphcast`
[6]`https://github.com/198808xc/Pangu-Weather`

Table 17: **Global weather forecasting results (ACC only)** based on $5.625°$ ERA5 dataset. The metric reported is latitude-weighted ACC ($\uparrow$).

| Variable | Lead-Time $\Delta t$ (h) | IFS (2023) | NODE (2018) | FCN (2022) | ClimaX* (2023) | ClimaX (2023) | ClimODE (2024) | DeepPrim$^\dagger$ Ours | DeepPrim Ours |
|---|---|---|---|---|---|---|---|---|---|
| $z500$ | 6 | 1.00 | 0.96 | 0.99 | 0.97 | 1.00 | 0.99 | 1.00 | 1.00 |
| | 12 | (N/A) | 0.88 | 0.99 | 0.96 | 1.00 | 0.99 | 1.00 | 1.00 |
| | 18 | (N/A) | 0.79 | 0.99 | 0.95 | 1.00 | 0.98 | 1.00 | 1.00 |
| | 24 | 1.00 | 0.70 | 0.99 | 0.93 | 1.00 | 0.98 | 0.99 | 1.00 |
| $t850$ | 6 | 0.99 | 0.94 | 0.99 | 0.94 | 0.98 | 0.97 | 0.98 | 0.99 |
| | 12 | (N/A) | 0.85 | 0.99 | 0.93 | 0.98 | 0.96 | 0.98 | 0.99 |
| | 18 | (N/A) | 0.77 | 0.99 | 0.92 | 0.98 | 0.96 | 0.98 | 0.98 |
| | 24 | 0.99 | 0.72 | 0.99 | 0.90 | 0.98 | 0.95 | 0.97 | 0.98 |
| $t2m$ | 6 | 0.99 | 0.82 | 0.99 | 0.92 | 0.98 | 0.97 | 0.99 | 0.99 |
| | 12 | (N/A) | 0.68 | 0.99 | 0.90 | 0.97 | 0.96 | 0.98 | 0.99 |
| | 18 | (N/A) | 0.69 | 0.99 | 0.88 | 0.97 | 0.96 | 0.97 | 0.98 |
| | 24 | 0.99 | 0.79 | 0.99 | 0.89 | 0.98 | 0.96 | 0.97 | 0.98 |
| $u10$ | 6 | 0.98 | 0.85 | 0.95 | 0.92 | 0.97 | 0.91 | 0.97 | 0.98 |
| | 12 | (N/A) | 0.70 | 0.93 | 0.88 | 0.95 | 0.89 | 0.96 | 0.97 |
| | 18 | (N/A) | 0.58 | 0.91 | 0.84 | 0.95 | 0.88 | 0.95 | 0.96 |
| | 24 | 0.97 | 0.50 | 0.89 | 0.80 | 0.94 | 0.87 | 0.94 | 0.95 |
| $v10$ | 6 | 0.98 | 0.81 | 0.94 | 0.92 | (N/A) | 0.92 | 0.97 | 0.98 |
| | 12 | (N/A) | 0.61 | 0.91 | 0.88 | (N/A) | 0.89 | 0.96 | 0.97 |
| | 18 | (N/A) | 0.46 | 0.86 | 0.83 | (N/A) | 0.88 | 0.95 | 0.96 |
| | 24 | 0.97 | 0.35 | 0.83 | 0.80 | (N/A) | 0.86 | 0.94 | 0.95 |

"DeepPrim" takes all surface and upper-air variables as output variables for training, whose full results are summarized in Table 19; "DeepPrim$^\dagger$" only outputs five key variables. "ClimaX*" denotes ClimaX that is not pre-trained on CMIP6 Eyring et al. (2016).

### F.2 CLARIFICATION OF THE IFS RESULTS IN TABLE 2 AND TABLE 3

In Table 2, we use the same IFS baselines as ClimaX (i.e., IFS HRES vs. ERA5), and we also adopt the same experimental settings. Therefore, we cited the results from the paper of ClimaX. Also, the results of IFS are also consistent with those in ClimODE and WeatherGFT. The results may be inconsistent with those reported in WeatherBench2 (Rasp et al., 2020). The main reason lies in **different regridding methods**. As the results of IFS on resolution of $5.625°$ and $1.4°$ are down-sampled from higher resolutions, the inconsistent results are mainly due to discrepancies of downsampling strategies (i.e., the ways of regridding from $0.25°$ to $5.6°$). Following ClimaX, we use bilinear interpolation, which considers 4 adjacent grids for regridding, while linear conservative regridding used in WeatherBench2 is more like avg. pooling. If the resolution reduction is relatively large, the latter would significantly reduce the data noise and the prediction error.

In Table 2, all the comparison results of IFS, Pangu ($0.25°$) and GraphCast ($0.25°$) are obtained from WeatherBench2 [7].

## G EXTERNAL EXPERIMENTS AND FULL QUANTITATIVE RESULTS

### G.1 GLOBAL WEATHER FORECASTING RESULTS EVALUATED BY ACC

Anomaly Correlation Coefficient (ACC) is a standard metric in meteorology and climate science to assess a model's skill in capturing anomalies (deviations from the climatological mean). Here we summarize the weather forecasting results based on ACC in Table 17. The results based on RMSE are in Table 2.

### G.2 GLOBAL WEATHER FORECASTING AT LONGER LEAD TIME HORIZONS AND DISCUSSIONS

Table 18 summarizes the comparison of our model with ClimaX (both the one that is pre-trained and the one that is trained from scratch) and ClimODE for longer lead time, i.e., $\Delta t = \{36h, 72h, 144h\}$ corresponding to $\{1.5 \text{ days}, 3 \text{ days}, 6 \text{ days}\}$. Empowered by pre-training on the CMIP6 dataset, ClimaX shows relatively stable performance for forecasting at longer lead time, as pre-training lays a

---

[7] https://sites.research.google/weatherbench/deterministic-scores/

great foundation for post-training and degrades the difficulty of long-lead time forecasting. Specifically, our DeepPrim, despite not being pre-trained on any external data, presents comparable results in forecasting $t850$ $t2m$, and $u10$ in comparison with pre-trained ClimaX. Moreover, our DeepPrim consistently achieves better performance than the non-pre-trained ClimaX* and ClimODE.

Actually, as pointed out in recent work (Gao et al.; Chen et al., 2023b), accurate long-term forecasting often requires specialized training techniques, model ensemble strategies (e.g., OneForecast (Gao et al.)), and autoregressive refinement (e.g., FuXi (Chen et al., 2023b)). These aspects are beyond the primary scope of our current work. In the future, it is of interest to incorporate such techniques to further enhance medium- and long-range forecasting of DeepPrim. Additionally, adopting more advanced ODE solvers (e.g., RK4 instead of Euler) may further improve long-term prediction stability and accuracy.

Table 18: **Long lead time prediction results of global weather forecasting** based on $5.625°$ ERA5 dataset. The evaluation metrics are Latitude-weighted RMSE and ACC.

| Variables | Lead-Time $\Delta t$ (h) | RMSE | | | | | | ACC | | | | | |
|---|---|---|---|---|---|---|---|---|---|---|---|---|---|
| | | IFS | ClimaX* | ClimaX | ClimODE | DeepPrim | DeepPrim* | IFS | ClimaX* | ClimaX | ClimODE | DeepPrim | DeepPrim* |
| z500 | 36 | 66.7 | 455.0 | 126.4 | 259.6 | 154.3 | 160.7 | 1.00 | 0.89 | 1.00 | 0.96 | 0.99 | 0.99 |
| | 72 | 147.0 | 687.0 | 244.1 | 478.7 | 319.4 | 331.9 | 0.98 | 0.73 | 0.97 | 0.88 | 0.95 | 0.94 |
| | 144 | 430.3 | 801.9 | 523.5 | 783.6 | 558.7 | 574.9 | 0.86 | 0.58 | 0.86 | 0.61 | 0.81 | 0.80 |
| t850 | 36 | 0.83 | 2.49 | 1.25 | 1.75 | 1.27 | 1.19 | 0.97 | 0.86 | 0.97 | 0.94 | 0.96 | 0.97 |
| | 72 | 1.19 | 3.17 | 1.59 | 2.58 | 1.88 | 1.78 | 0.94 | 0.76 | 0.98 | 0.85 | 0.94 | 0.94 |
| | 144 | 2.3 | 3.97 | 2.54 | 3.62 | 2.75 | 2.64 | 0.77 | 0.69 | 0.84 | 0.77 | 0.82 | 0.84 |
| t2m | 36 | 0.95 | 2.87 | 1.33 | 1.70 | 1.49 | 1.21 | 0.93 | 0.83 | 0.97 | 0.94 | 0.96 | 0.97 |
| | 72 | 1.15 | 3.97 | 1.43 | 2.75 | 1.88 | 1.57 | 0.90 | 0.83 | 0.98 | 0.85 | 0.95 | 0.95 |
| | 144 | 1.83 | 3.38 | 2.01 | 3.30 | 2.78 | 2.60 | 0.75 | 0.38 | 0.92 | 0.79 | 0.83 | 0.85 |
| u10 | 36 | 1.07 | 2.98 | 1.57 | 2.25 | 1.70 | 1.60 | 0.96 | 0.69 | 0.93 | 0.83 | 0.93 | 0.93 |
| | 72 | 1.69 | 3.70 | 2.18 | 3.19 | 2.55 | 2.44 | 0.90 | 0.30 | 0.94 | 0.66 | 0.81 | 0.82 |
| | 144 | 3.17 | 4.24 | 3.24 | 4.02 | 3.65 | 3.55 | 0.66 | 0.30 | 0.63 | 0.35 | 0.55 | 0.57 |
| v10 | 36 | 1.12 | 2.98 | N/A | 2.29 | 1.75 | 1.64 | 0.96 | 0.69 | N/A | 0.83 | 0.92 | 0.92 |
| | 72 | 1.75 | 3.80 | N/A | 3.30 | 2.58 | 2.43 | 0.90 | 0.39 | N/A | 0.63 | 0.79 | 0.80 |
| | 144 | 3.3 | 4.42 | N/A | 4.24 | 3.70 | 3.56 | 0.65 | 0.25 | N/A | 0.32 | 0.52 | 0.54 |

Kindly Note:
1. The results of IFS are obtained from WeatherBench2[8].
2. "DeepPrim†" denotes our model that takes all surface and upper-air variables as output variables for training;
3. "DeepPrim" denotes our model that only takes five key variables as output variables.
4. "ClimaX*" denotes ClimaX that is trained from scratch and "ClimaX" denotes ClimaX that is pre-trained with CMIP6 dataset.

### G.3 FULL GLOBAL WEATHER FORECASTING RESULTS

As mentioned in the main text, our DeepPrim receives all surface variables and upper-air variables as input variables to model the 3D atmospheric dynamics, and can also predict the future states of all surface and upper-air variables ($K = D_S + P \times D_U = 45$, where $D_S = 3$, $P = 7$, and $D_U = 6$). In Table 2, the forecasting performance of five key weather variables is reported in line with previous mainstream work (Nguyen et al., 2023; Verma et al., 2024). Here we summarize the forecasting performance of all surface and upper-air variables in Table 19.

### G.4 QUANTITATIVE RESULTS OF ABLATION STUDIES

The numerical quantitative results of ablation studies are summarized in Table 20, corresponding to Fig. 3 in the main text.

---

[8]https://sites.research.google/weatherbench/deterministic-scores/

Table 19: **Full results of DeepPrim[†] for global weather forecasting** based on ERA5 dataset with $5.625°$ resolution.

| Variable | Pressure level | RMSE | | | | ACC | | | |
|---|---|---|---|---|---|---|---|---|---|
| | | $\Delta t = 6$ | $\Delta t = 12$ | $\Delta t = 18$ | $\Delta t = 24$ | $\Delta t = 6$ | $\Delta t = 12$ | $\Delta t = 18$ | $\Delta t = 24$ |
| U wind $u10(m/s)$ | Surface | 0.89420 | 1.07571 | 1.24577 | 1.39002 | 0.9846 | 0.9780 | 0.9534 | 0.9417 |
| V wind $v10(m/s)$ | Surface | 0.92482 | 1.10320 | 1.27185 | 1.42697 | 0.9745 | 0.9637 | 0.9518 | 0.9390 |
| 2m Temperature $t2m(K)$ | Surface | 0.94463 | 1.12392 | 1.17341 | 1.19324 | 0.9737 | 0.9626 | 0.9738 | 0.9729 |
| Geopotential $z\,(m^2/s^2)$ | 50hPa | 90.86801 | 122.57426 | 152.14494 | 172.36325 | 0.9990 | 0.9983 | 0.9975 | 0.9968 |
| | 250hPa | 75.98683 | 108.39375 | 138.29964 | 174.10442 | 0.9988 | 0.9976 | 0.9962 | 0.9939 |
| | 500hPa | 50.13780 | 71.07771 | 94.17224 | 121.10972 | 0.9988 | 0.9975 | 0.9957 | 0.9928 |
| | 600hPa | 44.78042 | 62.57886 | 82.38021 | 105.66311 | 0.9985 | 0.9972 | 0.9951 | 0.9919 |
| | 700hPa | 42.91045 | 59.26258 | 77.16673 | 97.76039 | 0.9981 | 0.9964 | 0.9939 | 0.9902 |
| | 850hPa | 43.85304 | 59.68118 | 76.36211 | 95.31515 | 0.9972 | 0.9948 | 0.9915 | 0.9867 |
| | 900hPa | 46.10273 | 62.33511 | 79.26835 | 98.23368 | 0.9968 | 0.9942 | 0.9906 | 0.9855 |
| Relative Humidity $q\,(\%)$ | 50hPa | 0.94100 | 1.12839 | 1.27020 | 1.37069 | 0.9881 | 0.9829 | 0.9782 | 0.9748 |
| | 250hPa | 13.33168 | 15.45208 | 16.65293 | 17.70898 | 0.9092 | 0.8758 | 0.8541 | 0.8331 |
| | 500hPa | 11.41918 | 13.52854 | 14.85370 | 16.01361 | 0.9121 | 0.8742 | 0.8460 | 0.8184 |
| | 600hPa | 10.12321 | 12.19817 | 13.44334 | 14.59717 | 0.9283 | 0.8940 | 0.8696 | 0.8441 |
| | 700hPa | 9.63010 | 11.63048 | 12.75557 | 13.78103 | 0.9260 | 0.8901 | 0.8660 | 0.8415 |
| | 850hPa | 9.51776 | 11.36048 | 12.21194 | 12.94124 | 0.9034 | 0.8592 | 0.8353 | 0.8127 |
| | 900hPa | 6.90951 | 8.08307 | 8.60306 | 8.99389 | 0.9007 | 0.8613 | 0.8412 | 0.8248 |
| Specific Humidity $q\,(g/kg)$ | 50hPa | 6.686E-08 | 7.659E-08 | 8.281E-08 | 8.771E-08 | 0.9557 | 0.9419 | 0.9318 | 0.9234 |
| | 250hPa | 2.924E-05 | 3.378E-05 | 3.594E-05 | 3.768E-05 | 0.8943 | 0.8557 | 0.8348 | 0.8166 |
| | 500hPa | 3.166E-04 | 3.928E-04 | 4.299E-04 | 4.585E-04 | 0.9460 | 0.9156 | 0.8979 | 0.8829 |
| | 600hPa | 4.442E-04 | 5.556E-04 | 6.067E-04 | 6.479E-04 | 0.9524 | 0.9245 | 0.9092 | 0.8958 |
| | 700hPa | 6.031E-04 | 7.544E-04 | 8.212E-04 | 8.737E-04 | 0.9487 | 0.9185 | 0.9026 | 0.8890 |
| | 850hPa | 8.001E-04 | 9.851E-04 | 1.058E-03 | 1.116E-03 | 0.9393 | 0.9064 | 0.8911 | 0.8780 |
| | 900hPa | 6.591E-04 | 7.806E-04 | 8.275E-04 | 8.629E-04 | 0.9510 | 0.9304 | 0.9214 | 0.9142 |
| Temperature $t(K)$ | 50hPa | 0.84614 | 0.97900 | 1.08180 | 1.15405 | 0.9850 | 0.9801 | 0.9759 | 0.9727 |
| | 250hPa | 0.62926 | 0.80314 | 0.93336 | 1.05305 | 0.9878 | 0.9801 | 0.9731 | 0.9655 |
| | 500hPa | 0.59038 | 0.72667 | 0.82702 | 0.93176 | 0.9908 | 0.9861 | 0.9820 | 0.9771 |
| | 600hPa | 0.57439 | 0.70415 | 0.80529 | 0.90684 | 0.9914 | 0.9870 | 0.9830 | 0.9783 |
| | 700hPa | 0.60335 | 0.74169 | 0.84022 | 0.93270 | 0.9909 | 0.9862 | 0.9823 | 0.9781 |
| | 850hPa | 0.76288 | 0.93235 | 1.03605 | 1.12888 | 0.9882 | 0.9823 | 0.9781 | 0.9739 |
| | 900hPa | 0.74482 | 0.92397 | 1.03406 | 1.12530 | 0.9886 | 0.9824 | 0.9779 | 0.9737 |
| U wind component $u(m/s)$ | 50hPa | 1.41168 | 1.77854 | 1.99226 | 2.16728 | 0.9898 | 0.9839 | 0.9799 | 0.9763 |
| | 250hPa | 2.38616 | 2.96588 | 3.41439 | 3.85471 | 0.9859 | 0.9782 | 0.9711 | 0.9629 |
| | 500hPa | 1.75584 | 2.14611 | 2.44368 | 2.74203 | 0.9812 | 0.9718 | 0.9633 | 0.9536 |
| | 600hPa | 1.56071 | 1.90175 | 2.14444 | 2.39050 | 0.9794 | 0.9692 | 0.9607 | 0.9509 |
| | 700hPa | 1.48509 | 1.79435 | 2.00816 | 2.21522 | 0.9758 | 0.9645 | 0.9554 | 0.9454 |
| | 850hPa | 1.39223 | 1.67430 | 1.87418 | 2.07397 | 0.9733 | 0.9612 | 0.9512 | 0.9399 |
| | 900hPa | 1.27750 | 1.53114 | 1.73867 | 1.95376 | 0.9767 | 0.9664 | 0.9564 | 0.9446 |
| V wind component $v(m/s)$ | 50hPa | 1.42818 | 1.72542 | 1.91538 | 2.07149 | 0.9597 | 0.9408 | 0.9268 | 0.9143 |
| | 250hPa | 2.36407 | 2.95685 | 3.44890 | 3.94055 | 0.9829 | 0.9731 | 0.9633 | 0.9517 |
| | 500hPa | 1.74155 | 2.13558 | 2.45400 | 2.78292 | 0.9780 | 0.9667 | 0.9558 | 0.9427 |
| | 600hPa | 1.54916 | 1.88425 | 2.14326 | 2.41167 | 0.9757 | 0.9638 | 0.9529 | 0.9399 |
| | 700hPa | 1.45893 | 1.75918 | 1.97805 | 2.20431 | 0.9718 | 0.9588 | 0.9476 | 0.9345 |
| | 850hPa | 1.36526 | 1.64698 | 1.85200 | 2.06579 | 0.9701 | 0.9563 | 0.9444 | 0.9303 |
| | 900hPa | 1.27886 | 1.54310 | 1.76354 | 1.99765 | 0.9756 | 0.9643 | 0.9531 | 0.9393 |

Table 20: **Full ablation results of DeepPrim$^\dagger$ for global weather forecasting** based on ERA5 dataset with $5.625°$ resolution. The evaluation metric is Latitude-weighted RMSE.

| Variables | Lead time | w/o Source Sink | w/o $\nabla \mathbf{u}$ in Initialization Net | w/o $\dot{\mathbf{u}}$ in Initialization Net | w/o 3D modules | DeepPrim |
|---|---|---|---|---|---|---|
| $z500$ | 6 | 136.3 | 68.3 | 52.4 | 65.3 | 50.1 |
| | 12 | 149.2 | 84.2 | 75.4 | 83.2 | 71.0 |
| | 18 | 198.3 | 124.5 | 95.8 | 119.4 | 94.1 |
| | 24 | 258.5 | 155.7 | 127.7 | 139.4 | 121.0 |
| $t850$ | 6 | 1.34 | 0.88 | 0.77 | 0.84 | 0.76 |
| | 12 | 1.59 | 1.04 | 0.95 | 0.99 | 0.93 |
| | 18 | 1.70 | 1.15 | 1.07 | 1.13 | 1.04 |
| | 24 | 1.98 | 1.28 | 1.16 | 1.26 | 1.13 |
| $t2m$ | 6 | 2.43 | 1.03 | 0.90 | 1.01 | 0.89 |
| | 12 | 2.83 | 1.16 | 1.07 | 1.19 | 1.07 |
| | 18 | 2.51 | 1.29 | 1.18 | 1.32 | 1.17 |
| | 24 | 2.58 | 1.34 | 1.20 | 1.38 | 1.19 |
| $u10$ | 6 | 1.68 | 1.04 | 0.95 | 0.98 | 0.92 |
| | 12 | 1.81 | 1.25 | 1.13 | 1.19 | 1.10 |
| | 18 | 1.94 | 1.38 | 1.28 | 1.29 | 1.24 |
| | 24 | 2.30 | 1.60 | 1.42 | 1.52 | 1.39 |
| $v10$ | 6 | 1.70 | 1.10 | 1.0 | 1.05 | 0.94 |
| | 12 | 1.79 | 1.26 | 1.1 | 1.19 | 1.12 |
| | 18 | 1.95 | 1.41 | 1.3 | 1.38 | 1.27 |
| | 24 | 2.28 | 1.63 | 1.4 | 1.58 | 1.42 |

## G.5 ADDITIONAL ABLATION ANALYSIS OF BACKBONE ARCHITECTURES

As discussed in Appendix Section B.4, the backbones of the initialization network and the source-sink network are based on CNN (ResNet), and the backbone of the force network is based on the carefully designed 3D-BiViT. We further carry out backbone ablation experiments with graph networks modules using message passing and summarize the results in Table 21. From Table 21, we can observe that:

(1). Replacing the CNN(Resnet) in the Initialization Network and Source-Sink Network with GNNs yields comparable performance, as both architectures excel at modeling local node interactions for initial state estimation and residual corrections.

(2). Substituting the 3D-BiViT in Force Network with GNNs caused significant performance degradation, as our 3D-BiViT leverages two-stage attention to resolve anisotropic interactions (i.e., vertical and horizontal advection), GNNs, while effective for graph-structured systems, struggle to parameterize continuous 3D spatial relationships and pressure-level-aware dynamics without explicit inductive biases. The results further demonstrate the effectiveness of the designed 3D-BiViT.

(3). ForceNet (3D-BiViT) is designed to learn the force terms in the Navier-Stokes equations, thereby effectively modeling 3D atmospheric dynamics by capturing both inter- and intra-pressure-level interactions for upper-air and surface variables in a coordinated way. In our ablation study, replacing 3D-BiViT with vanilla ViT or CNN modules resulted in varying degrees of performance degradation. It is because neither vanilla ViT, GNN, nor CNN explicitly models 3D interactions between pressure levels, which may lead to bias in estimating the velocity term and thus degrade the forecasting performance. The additional ablations further demonstrate the necessity of our 3D design and its coherence with the overall model architecture.

## G.6 MORE EXPERIMENTS ON "APPLE-TO-APPLE" COMPARISON WITH CLIMODE

It is noted that ClimODE (Verma et al., 2024) experiments are conducted under an experimental setup that differs notably from ours: (i) only five variables are used in total, namely $\{z500, t2m, u10, v10, t850\}$, both as inputs and outputs, and (ii) the model is trained on only 10 years of data. In our main results, we report the results from the original ClimODE paper under this setting for completeness. In this subsection, we perform a more "apples-to-apples" comparison by aligning the data usage and variable configuration as closely as possible.

Table 21: **Additional ablation study results of DeepPrim and its variants**. In these variants, each architecture of DeepPrim is replaced with graph neural networks (GNNs) using message passing. The evaluation metric is Latitude-weighted RMSE, and the forecasting lead time is $\Delta t = 6h$.

| Model | Initilization Net | Force Net | Source-Sink Net | RMSE $\Delta t = 6h$ | | | | |
|---|---|---|---|---|---|---|---|---|
| | | | | z500 $(m^2/s^2)$ | t850 $(K)$ | t2m $(K)$ | u10 $(m/s)$ | v10 $(m/s)$ |
| DeepPrim | CNN (ResNet) | 3D-BiViT | CNN (ResNet) | 50.1 | 0.76 | 0.89 | 0.92 | 0.94 |
| Variant#1 | GNN | 3D-BiViT | CNN(ResNet) | 50.9 | 0.77 | 0.88 | 0.95 | 0.96 |
| Variant#2 | CNN(ResNet) | 3D-BiViT | GNN | 54.3 | 0.79 | 0.92 | 0.92 | 0.95 |
| Variant#3 | CNN (ResNet) | GNN | CNN (ResNet) | 69.3 | 0.91 | 0.98 | 1.01 | 1.02 |
| Variant#4 | CNN (ResNet) | Vanilia ViT | CNN (ResNet) | 64.4 | 0.94 | 0.99 | 1.04 | 1.06 |
| Variant#5 | CNN(ResNet) | CNN (ResNet) | CNN (ResNet) | 65.3 | 0.87 | 0.96 | 0.99 | 1.03 |

Our main ERA5–5.625° experiments follow the protocol used in ClimaX and are described in Section D. To disentangle the effects of data volume and variable selection from the architectural differences between DeepPrim and ClimODE, we consider the following controlled settings:

- **DeepPrim (full configuration).** Our default setting described above: 1979–2015 for training, 2016 for validation, 2017–2018 for testing; all selected variables and levels as inputs, all non-static variables as outputs.

- **DeepPrim (5-variable configuration).** We restrict DeepPrim to use exactly the same five variables as ClimODE, both as inputs and outputs, while keeping the same temporal split (1979–2015 train, 2016 validation, 2017–2018 test)

- **ClimODE (retrained with long-span data).** We reimplement ClimODE and retrain it using the same temporal split as DeepPrim (1979–2015 train, 2016 validation, 2017–2018 test), under its original five-variable configuration. This isolates the effect of increasing the training period.

The corresponding results are summarized in Table 22. First, when ClimODE is retrained on the longer 1979-2015 period, its performance improves compared to the originally reported 10-year setting, which is consistent with scaling-law behavior in data-driven forecasting. Second, when DeepPrim is restricted to the same five variables as ClimODE, its performance degrades relative to our full multi-variable, multi-pressure-level configuration, reflecting the loss of cross-pressure-level and cross-variable information that our model is designed to exploit. Third, and most importantly, under the *same* five-variable configuration and the *same* long-span data split, DeepPrim still outperforms the retrained ClimODE baseline across lead times and metrics.

These results indicate that: (i) both DeepPrim and ClimODE benefit from longer training records and richer input variables; and (ii) even under matched conditions in terms of variables and data span, DeepPrim achieves stronger performance. Therefore, the gains we report over ClimODE cannot be attributed solely to using more data or more variables, but also arise from our primitive-equation–inspired Neural ODE design, explicit 3D multi-level representation, and force-network–based treatment of source terms.

## H    LIMITATION AND FUTURE WORK

**Probabilistic and Long-term Forecast.** A limitation of DeepPrim is that it only supports deterministic weather forecasting in its current form. Extending DeepPrim to probabilistic forecasting is a promising future direction, and we will explore it in subsequent research. Feasible techniques could be reparameterizing the Source-Sink Network to output distribution parameters (e.g., mean and variance), enabling sampling-based uncertainty quantification while retaining the physics-guided ODE framework. Also, as pointed out in recent work (Gao et al.; Chen et al., 2023b), accurate long-term forecasting often requires specialized training techniques, model ensemble strategies (e.g., OneForecast (Gao et al.)), autoregressive refinement (e.g., FuXi (Chen et al., 2023b)), rolling initialization, and explicit error correction. These aspects are beyond the primary scope of our current work. In the future, it is of interest to incorporate such techniques to further enhance medium- and long-range forecasting of DeepPrim.

Table 22: **Global weather forecasting results on 5.625° ERA5 dataset (ClimODE vs. Deep-Prim).**

| Variable | Lead-Time $\Delta t$ (h) | ClimODE | | DeepPrim* | |
|---|---|---|---|---|---|
| | | 10 year for training | 1979–2015 for training | 5 variable input | full variable input |
| z500 | 6 | 102.9 | 79.1 | 69.5 | 53.6 |
| | 12 | 134.8 | 111.2 | 98.9 | 77.4 |
| | 18 | 162.7 | 148.9 | 129.8 | 102.2 |
| | 24 | 193.4 | 192.8 | 159.6 | 126.7 |
| t850 | 6 | 1.16 | 1.01 | 0.81 | 0.72 |
| | 12 | 1.32 | 1.23 | 0.98 | 0.87 |
| | 18 | 1.47 | 1.40 | 1.04 | 0.98 |
| | 24 | 1.55 | 1.52 | 1.16 | 1.06 |
| t2m | 6 | 1.21 | 1.13 | 0.79 | 0.73 |
| | 12 | 1.45 | 1.36 | 0.97 | 0.88 |
| | 18 | 1.43 | 1.54 | 1.02 | 0.97 |
| | 24 | 1.40 | 1.75 | 1.08 | 0.99 |
| u10 | 6 | 1.41 | 1.15 | 1.02 | 0.82 |
| | 12 | 1.81 | 1.41 | 1.13 | 1.00 |
| | 18 | 1.97 | 1.68 | 1.33 | 1.15 |
| | 24 | 2.01 | 1.83 | 1.51 | 1.25 |
| v10 | 6 | 1.53 | 1.21 | 1.03 | 0.85 |
| | 12 | 1.81 | 1.47 | 1.19 | 1.03 |
| | 18 | 1.96 | 1.65 | 1.35 | 1.19 |
| | 24 | 2.04 | 1.86 | 1.52 | 1.24 |

**Advanced Time-stepping Method**. In our implementation, we support Euler, Runge-Kutta (RK4), and other ODE solvers (through `torchdiffeq` [9]) for discretizing the ODEs, and we previously used the Euler method to solve the ODE system, as it is simple, efficient, and adequate for short-term forecasting. Compared to the simple Euler method, RK4 may require more computational effort but offers higher-order accuracy, improved stability, and better control of local truncation error, which can help capture the complex temporal dynamics in atmospheric modeling, thereby helping mitigate error accumulation, especially for medium- and long-range forecasts. Exploring these advanced time-stepping methods for longer-term forecasting remains a promising direction for future work.

**Data Seperation Pipeline**. Our experimental setting aligns with ClimaX, i.e., the separation of train (1979∼2015)/validation(2016)/test(2018), which may also have some limitations and biases. The dataset separation could be more interesting and fair by employing a more expensive but more robust scheme by keeping a rolling train/val/test set, e.g.,, 5 consecutive days for train, 1 for val, 1 for test, than 5 for train etc, for all the data. It could allow for avoiding biases or global changes, potentially from global warming between 1980 and 2018.

# I   VISUALIZATION OF RMSE CURVES FOR GLOBAL AND REGIONAL WEATHER FORECASTING

To provide a clear understanding of the forecasting performance comparison, we provide the RMSE curves at different lead times for global and regional weather forecasting tasks in Fig. 9, Fig. 10, Fig. 11, Fig. 12, Fig. 13.

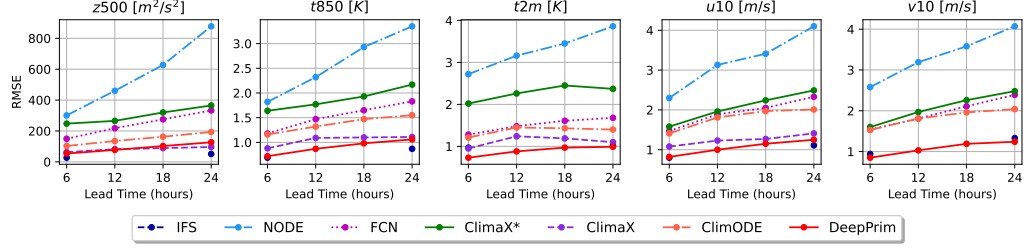

Figure 9: Global weather forecasting results based on 5.625° ERA5 dataset evaluated by latitude-weighted RMSE. It corresponds to Table 2 in our paper.

---

[9] https://github.com/rtqichen/torchdiffeq

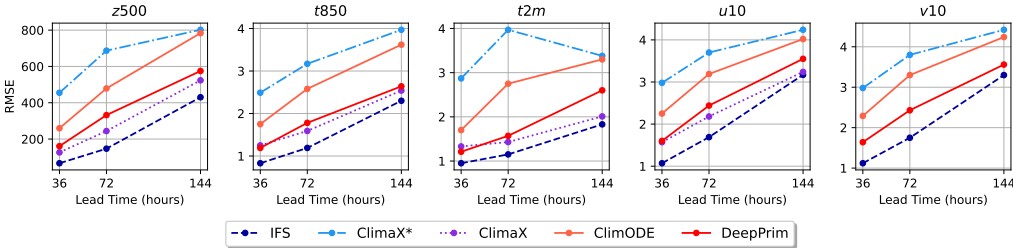

Figure 10: Long lead time prediction results of global weather forecasting based on $5.625°$ ERA5 dataset with latitude-weighted RMSE. It corresponds to Table 18 in our paper.

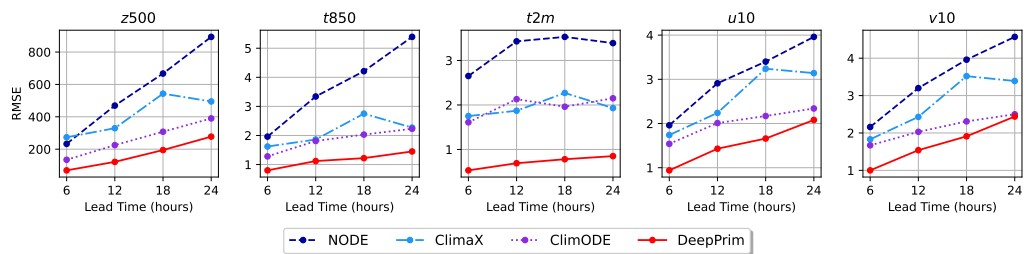

Figure 11: North America regional weather forecasting results based on $5.625°$ ERA5 dataset with latitude-weighted RMSE. It corresponds to Table 4 in our paper.

## J    VISUALIZATION OF GLOBAL FORECASTING MAPS

To intuitively demonstrate the forecasting capacity of our method, we present the showcases of weather forecasting results and bias maps with lead time $\Delta t = \{6, 12, 18, 24\}$ in Fig. 14, Fig. 15, Fig. 16, and Fig. 17, respectively. Each figure includes the initial conditions, the DeepPrim's predictions of future states, the ground truth of further states, and the bias map of five key weather variables, i.e., $t2m$, $u10$, $v10$, $g500$, and $t850$. Furthermore, we provide showcases to demonstrate the forecasting accuracy of DeepPrim in comparison with ClimaX as in Fig. 18, 19, 20, 21 with lead time $\Delta t = \{6, 12, 18, 24\}$, respectively. As presented, our DeepPrim can provide more accurate predictions via physics-informed 3D atmospheric modeling in comparison with ClimaX.

## K    BROADER IMPACTS

This research focuses on weather forecasting, which has an essential influence on relevant fields such as energy, transportation, and agriculture. As an AI application for social good, our model boosts predictions for various weather factors such as temperature, wind speed, and geopotential in both global and regional perspectives. It is essential to note that our work focuses solely on scientific issues, and we also ensure that ethical considerations are carefully taken into account. All

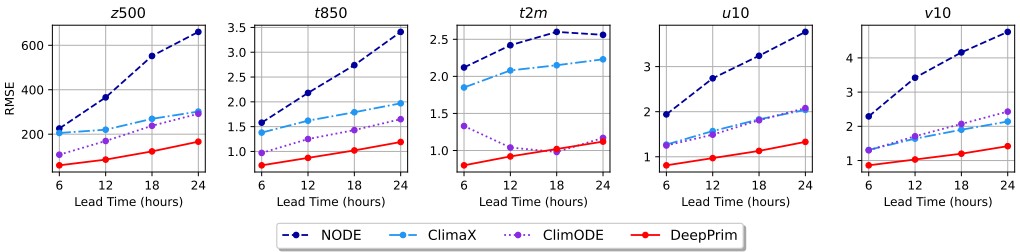

Figure 12: South America regional weather forecasting results based on $5.625°$ ERA5 dataset with latitude-weighted RMSE. It corresponds to Table 4 in our paper.

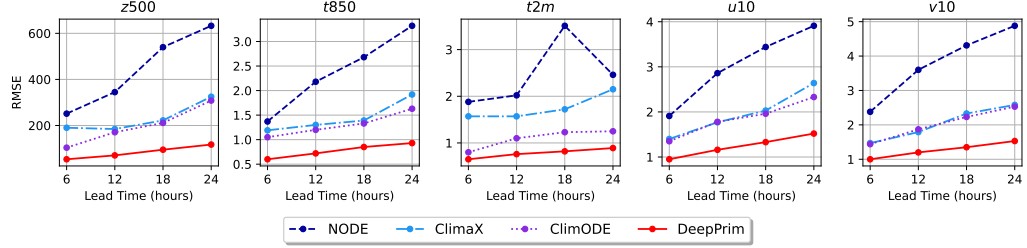

Figure 13: Australia regional weather forecasting results based on 5.625° ERA5 dataset with latitude-weighted RMSE. It corresponds to Table 4 in our paper.

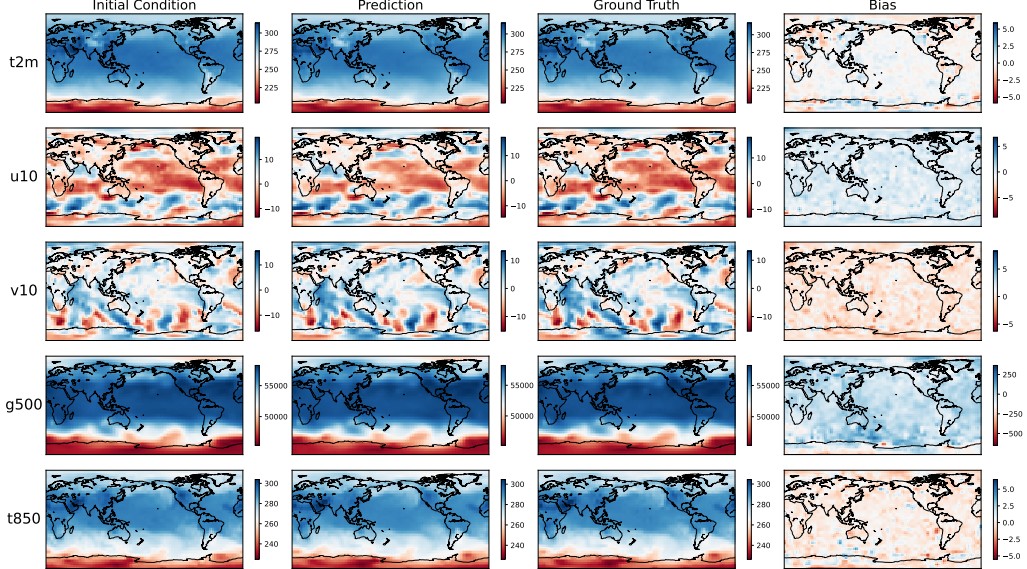

Figure 14: Visualization showcase of DeepPrim[†] for global weather forecasting with leat time $\Delta t = 6h$ based on ERA5 dataset with $5.625°$ resolution.

the weather datasets are publicly available for scientific research. Thus, we believe that there is no ethical risk associated with our research.

## L  DECLARATION OF LLM USAGE

The authors used LLM solely as a general-purpose assistive tool for grammar refinement and minor formatting suggestions. LLM did not contribute to research ideation, experimental design, data analysis, or interpretation, and all content was reviewed and is the full responsibility of the authors.

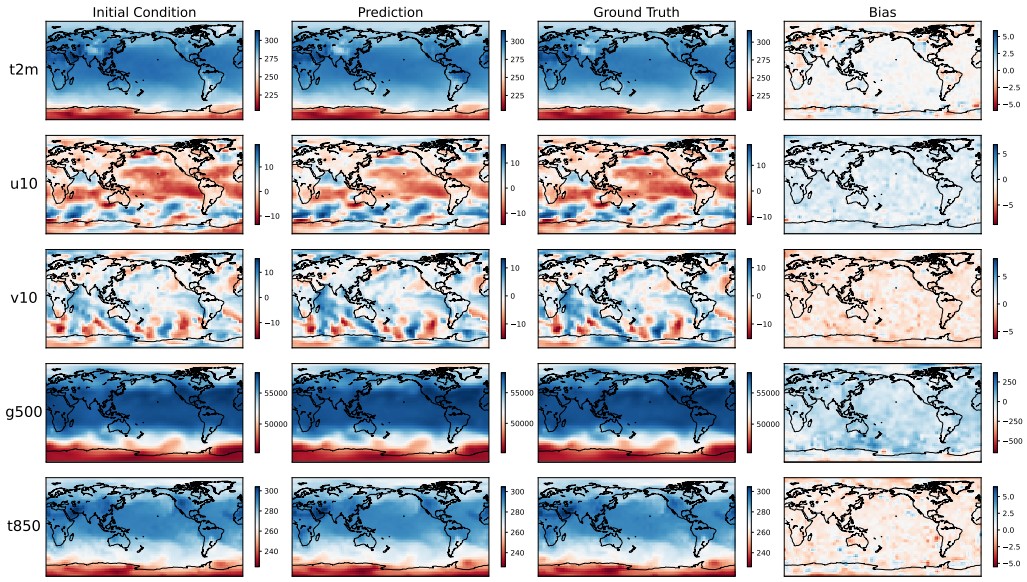

Figure 15: Visualization showcase of DeepPrim[†] for global weather forecasting with leat time $\Delta t = 12h$ based on ERA5 dataset with $5.625°$ resolution.

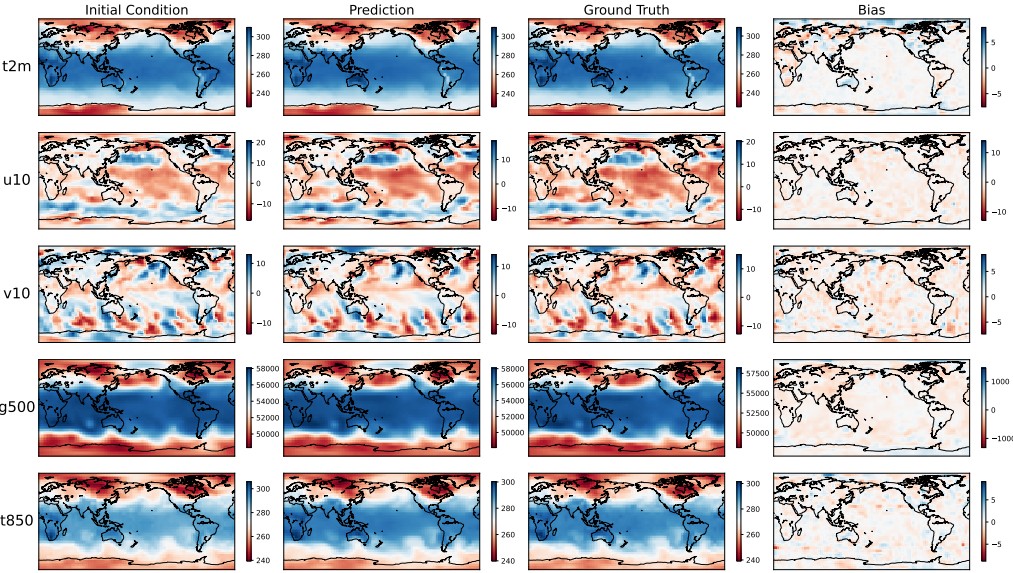

Figure 16: Visualization showcase of DeepPrim[†] for global weather forecasting with leat time $\Delta t = 18h$ based on ERA5 dataset with $5.625°$ resolution.

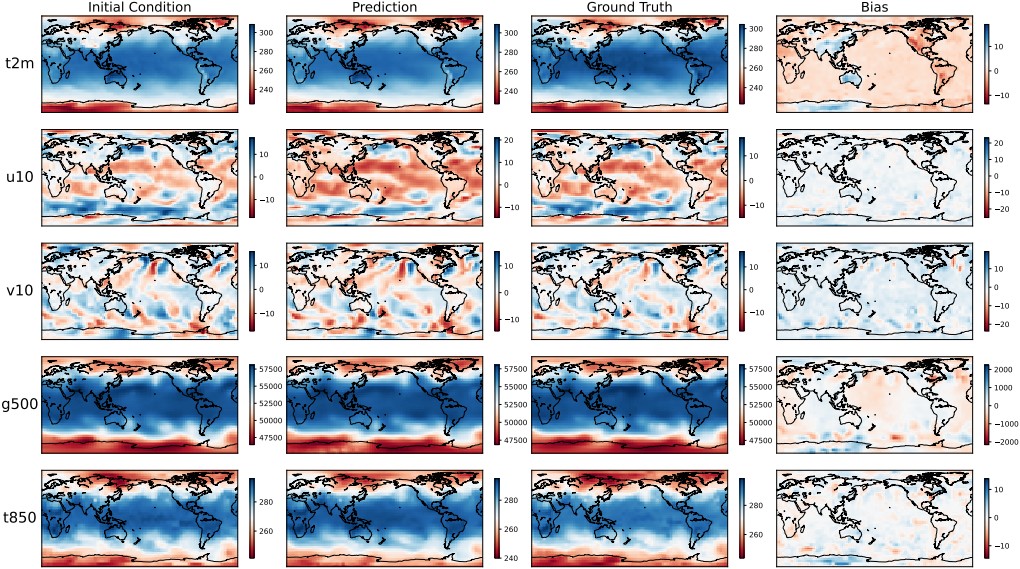

Figure 17: Visualization showcase of DeepPrim† for global weather forecasting with leat time $\Delta t = 24h$ based on ERA5 dataset with $5.625°$ resolution.

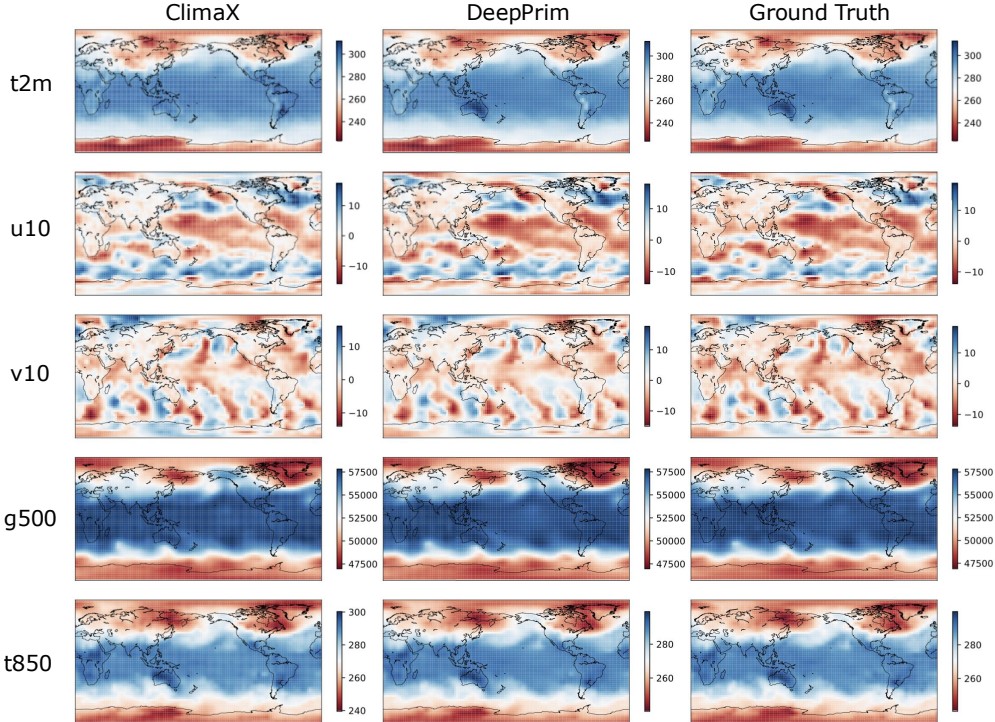

Figure 18: Prediction comparison of DeepPrim† and ClimaX for global weather forecasting with lead time $\Delta t = 6h$ based on ERA5 dataset with $5.625°$ resolution.

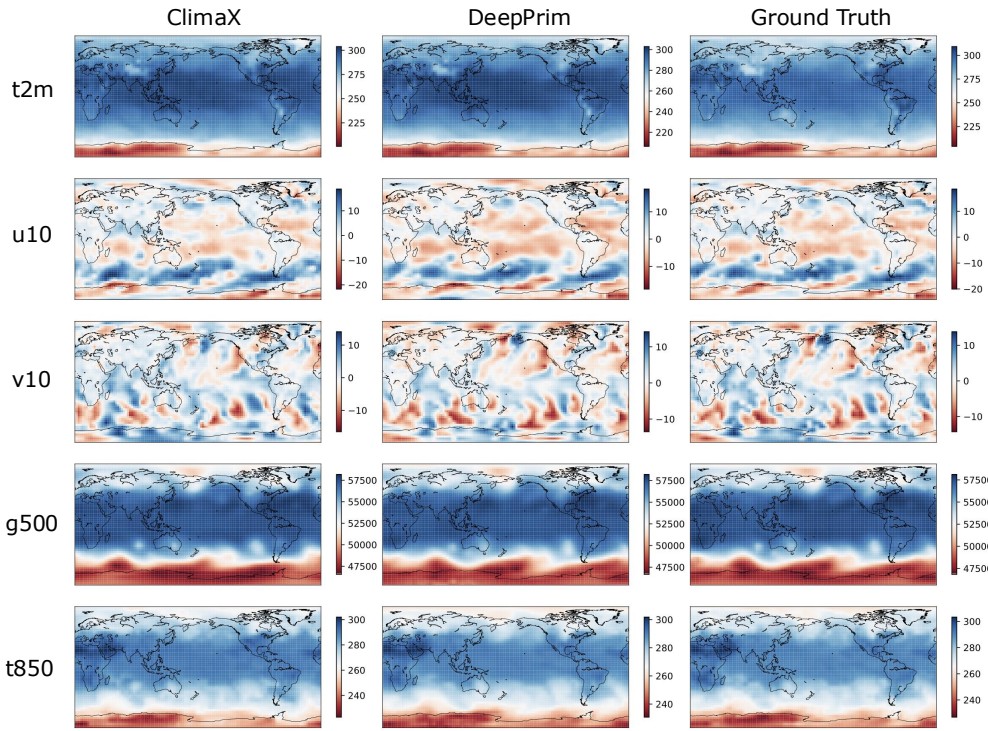

Figure 19: Prediction comparison of DeepPrim[†] and ClimaX for global weather forecasting with lead time $\Delta t = 12h$ based on ERA5 dataset with $5.625°$ resolution.

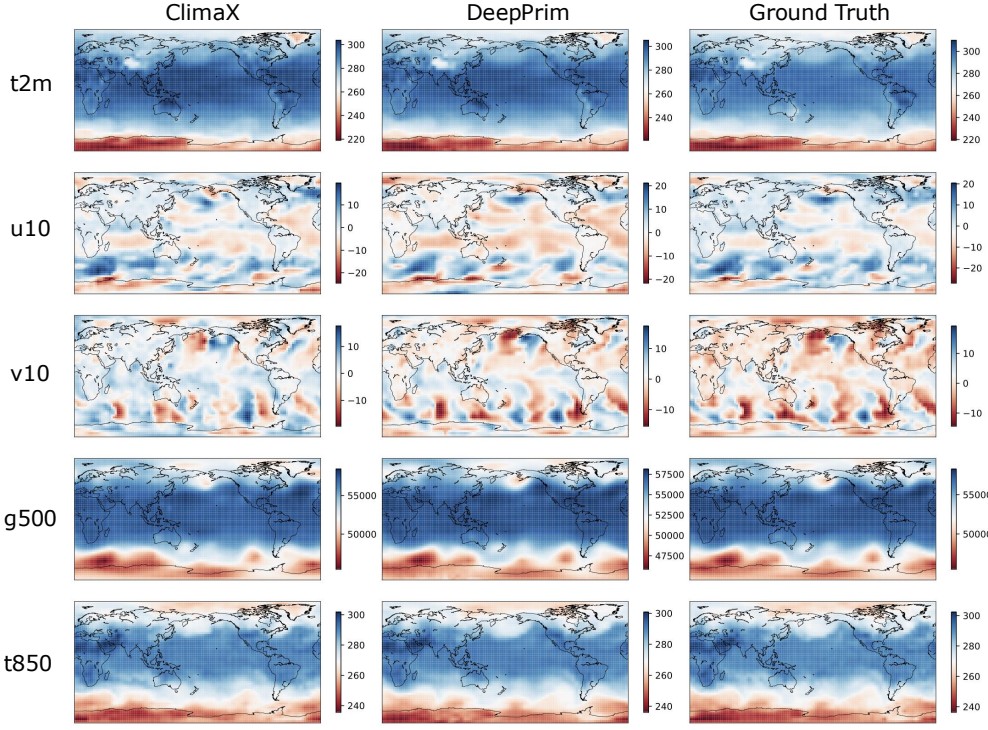

Figure 20: Prediction comparison of DeepPrim[†] and ClimaX for global weather forecasting with lead time $\Delta t = 18h$ based on ERA5 dataset with $5.625°$ resolution.

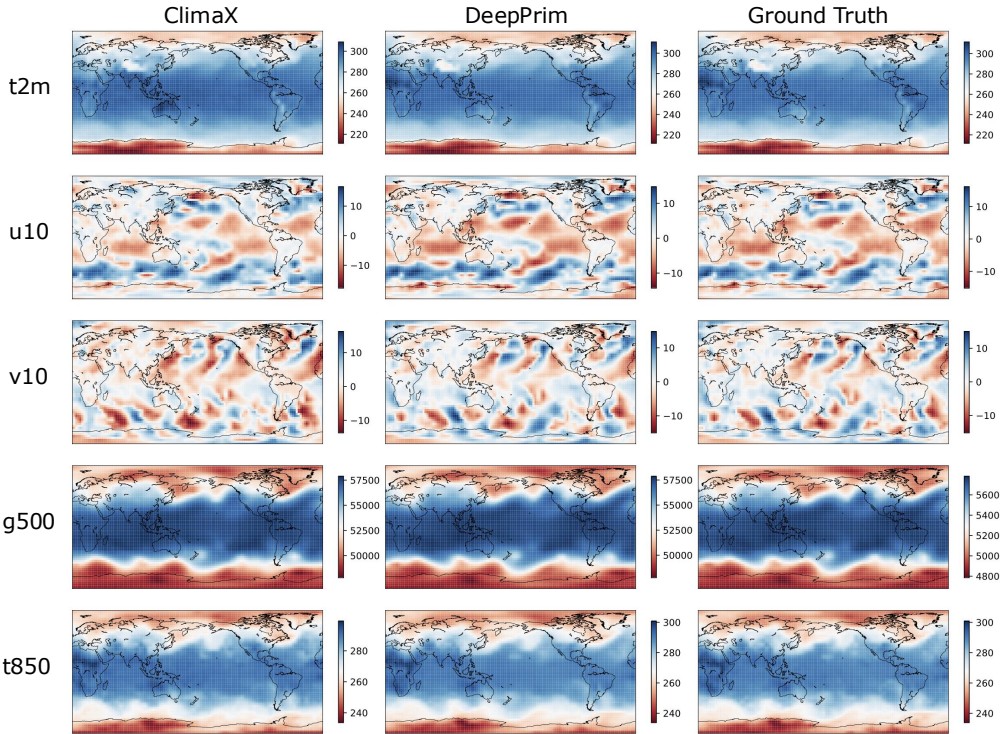

Figure 21: Prediction comparison of DeepPrim[†] and ClimaX for global weather forecasting with lead time $\Delta t = 24h$ based on ERA5 dataset with $5.625°$ resolution.

