# OpenReview forum: "DeepPrim: a Physics-Driven 3D Short-term Weather Forecaster via Primitive Equation Learning"
_ICLR.cc/2026/Conference — ICLR 2026 Poster_

### Official Review · Reviewer_Rdm2 · 2025-10-30

**Soundness:** 3
**Presentation:** 3
**Contribution:** 3
**Rating:** 6
**Confidence:** 3

**Summary:**

A novel 3D deep weather forecaster that explicitly integrates physical principles for short-term, continuous-time prediction. It addresses the trade-off between traditional Numerical Weather Prediction (NWP) models and purely data-driven deep learning models.

**Strengths:**

1.The model pioneers the explicit learning of the Primitive Equations (Navier-Stokes Force Terms) via a Neural ODE system. This design is highly original, ensuring physical consistency and enabling continuous-time prediction, which is superior to fixed-step forecasting methods.

2.It introduces the novel 3D Bi-Component Vision Transformer, which provides a physics-motivated inductive bias for modeling the atmosphere. This architecture is specifically designed to accurately capture heterogeneous inter- and intra-pressure-level interactions, which are crucial for 3D atmospheric motion.

**Weaknesses:**

1.While the Learnable Source-Sink Network is an innovative structural component, it essentially replaces complex, semi-empirical parameterization schemes used in NWP with a data-driven black-box parameterization. The network lacks explicit physical constraints or built-in interpretability, limiting the scientific novelty. It represents a missed opportunity to provide causal or physically restricted learned parameterizations that could advance our understanding of unresolved atmospheric processes.

2.The empirical validation is heavily concentrated on short-term forecasting (sub-24h). While the paper provides some longer-term results (up to 5-6 days) in the Appendix, the core analysis and demonstrated dominance are not extended to medium-to-long-range prediction. This limitation leaves the model's robustness and scalability in capturing cumulative errors over longer time horizons largely unexplored in the main text.

3.The paper lacks theoretical guarantees (e.g., stability analysis, energy conservation) that typically accompany physics-driven models. Without proof that the learned dynamics $f_{\theta}$ adhere to fundamental laws beyond empirical fitting, the claimed "physics-driven" novelty is limited to an architectural constraint rather than a proven theoretical foundation.

4.The paper omits a crucial analysis of inference time/computational complexity (FLOPs). Given the complexity of the 3D-BiViT architecture combined with the iterative nature of the Neural ODE solver, DeepPrim is likely more computationally expensive for real-time deployment than direct mapping models, a trade-off that is not quantified.

**Questions:**

1.Can the authors provide a targeted ablation study comparing the current Euler solver against a higher-order numerical integrator (e.g., RK4) for lead times of 36h and 72h? This is crucial for determining if the model's reduced performance at longer ranges is due to the learned dynamics $f_{\theta}$ or the numerical instability of the integration method.

2.What is the specific physical interpretation and necessity for introducing the "intermediate atmospheric motion" $v^*$ separate from the horizontal wind components $(v_x, v_y)$ within the prognostic variable $\mathbf{u}$? Is this required for numerical stability or does it represent a specific component of the flow?

3.The paper emphasizes capturing interactions with the solved advection. How is the advection term itself computed? Is it handled entirely implicitly by the neural network, or is it computed explicitly using a traditional numerical scheme (like semi-Lagrangian or finite differencing) and then corrected/fused by the network?

---

> ### Author Response · Authors · 2025-11-20
> **Response to Reviewer Rdm2 (Part 1)**
>
> Dear reviewer Rdm2:
>
> We sincerely thank you for your positive evaluation of our work's originality, physical consistency, and technical design. We deeply appreciate your professional review work, constructive comments, and invaluable suggestions, which are immensely helpful in enhancing our work. Rest assured, we are glad to address your concerns point-to-point as follows.
>
>
> ### ***W1: About Source-Sink Network***
>
> We fully agree that ideal parameterizations should be both physically consistent and interpretable. However, we'd like to clarify that, currently, it is computationally infeasible to fully and explicitly embed the physical mechanisms of all unresolved processes (such as turbulence and radiative transport), and the theory itself still has uncertainties. **In summary**, the Source-Sink Network is a pragmatic, data-driven approximation of *inherently uncertain* subgrid processes—used *only after* ensuring that resolved dynamics are explicitly physics-constrained. Specifically, DeepPrim adopts a **hierarchical physics-integration strategy**, balancing rigor with practicality:
>
> 1. **Explicit physics in core dynamics**: The Force Network (3D-BiViT) explicitly learns the *force terms* in the Navier-Stokes equation—namely, pressure gradient, Coriolis, and viscous friction forces (Section 3.2.2). This ensures that large-scale 3D atmospheric motion adheres to fundamental conservation laws and known physical structure.
>
> 2. **Source-Sink as a physics-inspired residual model**: Traditional NWP parameterizations (e.g., for convection, radiation, boundary-layer turbulence) are themselves *semi-empirical approximations* with significant structural uncertainty [1]. Our Source-Sink Network is not a generic black box; it is:
>    - **Physically motivated**: It models diabatic heating ($Q$) and moisture source/sink terms ($S_q$) from the thermodynamic and water vapor equations (as discussed in $\underline{\text{Appendix C.3–C.4 on Page 21}}$).
>    - **Informed by physical priors**: Its inputs include initial states, intermediate motion fields, ODE trajectories, and spatiotemporal embeddings encoding diurnal/seasonal cycles and static geography (Section 3.2.3).
>    - **A correction mechanism**: As Eq. (8) shows, it *residually corrects* the ODE solution rather than predicting from scratch, consistent with how "physics packages" refine dynamical cores in NWP.
>
> Crucially, our ablation study (in $\underline{\text{Section 4.4 on Page 10 and Fig. 3 on Page 8}}$) confirms that the Source-Sink Network significantly mitigates ODE error accumulation, demonstrating its capacity to capture real physical effects.
>
>
> *Ref:*
>
> [1]. Neural general circulation models for weather and climate. Nature, 2024.
>
> [2]. Machine learning for the physics of climate. Nature Reviews Physics, 2025.

---

> ### Author Response · Authors · 2025-11-20
> **Response to Reviewer Rdm2 (Part 2)**
>
> ### ***W2: About longer-term results***
>
> Thanks for your insightful feedback and for carefully examining our longer‑range results in the Appendix. While DeepPrim is primarily motivated by short‑range (sub‑24h) forecasting—driven by practical demands [1][2] in, e.g., renewable energy integration [3], event scheduling, and aviation planning [4], where**3D modeling and physics-informed continuous-time approaches are particularly critical**—we agree that a more explicit discussion of medium‑ to long‑range behavior is valuable.
>
> In the current work, we intentionally focus our main‑text analysis on short‑range lead times to (i) isolate and highlight the benefits of the primitive‑equation ODE formulation and Source–Sink design on high‑frequency evolution, and (ii) ensure a fair,comprehensive‑resolution comparison within the standard ERA5/WeatherBench protocol. Nevertheless, as the reviewer notes, we do provide extended experiments up to 5–6 days in the Appendix, where DeepPrim remains competitive and exhibits controlled error growth over longer horizons. These results indicate that the proposed continuous‑time, physics‑structured design does not break down when integrated forward and can serve as a solid backbone for longer‑range forecasting.
>
> At the same time, recent studies on medium‑ and long‑range forecasting (e.g., Pangu, GraphCast, OneForecast [5], FuXi [6]) suggest that pushing skill far beyond several days typically requires specialized training techniques, model ensemble strategies (e.g., OneForecast [5]), and autoregressive refinement (e.g., FuXi [6]). **These techniques are orthogonal to our core architectural contribution and can, in principle, be combined with DeepPrim to further strengthen its long‑range robustness and mitigate cumulative error**. We'd like to note that **these aspects are beyond the primary scope of our current work, but they are orthogonal and complementary directions**. Empowering our approach with these techniques could further enhance medium- and long-range forecasting of our DeepPrim, which we see as a promising avenue for future research.
>
> In light of your feedback, we have also supplemented further discussion on enhancement for long lead-time forecasting in $\underline{\text{Appendix H on Page 31}}$ of the revised paper.
>
>
> *Reference:*
>
> [1]. Radar refractivity retrieval: Validation and application to short-term forecasting. Journal of Applied Meteorology, 2005.
>
> [2]. A dynamic convolutional layer for short-range weather prediction. CVPR, 2015.
>
> [3]. Fusionsf: Fuse heterogeneous modalities in a vector quantized framework for robust solar power forecasting. KDD, 2024.
>
> [4]. No more flying blind: Leveraging weather forecasting for clear-cut risk-based decisions. Transportation Research Interdisciplinary Perspectives, 2025.
>
> [5]. OneForecast: A Universal Framework for Global and Regional Weather Forecasting. ICML, 2025.
>
> [6]. Fuxi: A cascade machine learning forecasting system for 15-day global weather forecast. npj Climate and Atmospheric Science, 2023.
>
>
> ### ***W3: About physics-driven and theoretical guarantee.***
>
> Thank you for highlighting the importance of theoretical guarantees such as stability and energy conservation in physics‑driven models.
>
> In this work, our use of the term “physics‑driven” refers to the fact that **DeepPrim is explicitly derived from, and constrained by, the primitive equations in pressure coordinates**: the dynamical core follows the structure of the momentum, continuity, thermodynamic, and moisture equations, and each learnable component parameterizes a specific physical term rather than acting as an unconstrained black‑box predictor (the detailed formulation of Primitive Equations of the atmosphere in $\underline{\text{Appendix. C on Pages 21 - 25}}$). This architectural design enforces mass‑consistent computation of vertical velocity, physically motivated advection and pressure‑gradient operators, and structured source–sink parameterizations, ensuring that the learned dynamics are guided by the underlying PDE formulation rather than purely empirical fitting.
>
> While a full, formal analysis of global stability and exact conservation properties for such high‑dimensional neural ODE systems is highly nontrivial and typically beyond the scope of current deep learning–based weather models, in this work we focus on demonstrating the practical benefits of this physics‑derived architecture through extensive empirical evaluation across multiple datasets under different resolutions (from $5.625°$ to $1.4°$ and $0.25°$). According to your feedback, we have revised the manuscript to clarify this notion of “physics‑driven” as a principled architectural and inductive‑bias design grounded in the primitive equations, and we now explicitly point out rigorous stability and conservation analysis as an important direction for future work in $\underline{\text{Section5: Conclusion on Page 10}}$.

---

> ### Author Response · Authors · 2025-11-20
> **Response to Reviewer Rdm2 (Part 3)**
>
> ### ***W4: Inference time/computational complexity***
>
> We greatly agree that computational complexity and effort analysis are critical. Due to space limitations, we have had to include the detailed training and inference time statistics and discussions in $\underline{\text{Appendix B.8 on Page 21}}$ in our original submission. Here we summarize the results in Tables R1 and R2 for your convenience.
>
> **Table R1: Inference time of DeepPrim** for each sample at different forecasting lead times based on a single NVIDIA A800-SXM4-80GB GPU. The evaluation metric is seconds (s).
>
> |Resolution|$\Delta t=6$|$\Delta t=12$|$\Delta t=18$|$\Delta t=24$|
> |---|---|---|---|---|
> |DeepPrim(5.625°)|0.16s|0.33s|0.47s|0.66s|
> |DeepPrim(1.4°)|0.64s|1.23s|1.81s|2.31s|
> |DeepPrim(0.25°)|2.48s|4.98s|7.47s|9.48s|
>
> **Table R2: Training time of DeepPrim** at different forecasting lead times. The maximum epoch is set as 50, and we use an early-stopping strategy with a patience of 3.
>
> |Resolution|Hardware|$\Delta t=6$|$\Delta t=12$|$\Delta t=18$|$\Delta t=24$|
> |---|---|---|---|---|---|
> |DeepPrim(5.625°)|4 NVIDIA A800-SXM4-80GB GPUs|13hours|1day2hours|1day7hours|1day19hours|
> |DeepPrim(1.4°)|4 NVIDIA A800-SXM4-80GB GPUs|5days|6days|6days|7days|
> |DeepPrim(0.25°)|8 NVIDIA A800-SXM4-80GB GPUs|8days|-|-|15days|
>
>
> For comparison, Pangu (0.25°)[1][2] reports a training time of 16 days on 192 V100 GPUs and an inference time of 1.4 seconds per sample.
>
> Overall, while our physics-informed Neural ODE approach introduces additional computational steps, **the inference time remains within a practical range for short-term (6–24h) forecasting**, supporting real-time or near-real-time decision-making.
>
> According to your suggestion, we are dedicated to explicitly highlighting these trade-offs and computational details in the main paper to provide a clearer picture of DeepPrim’s practical viability. We hope this addresses your concerns, and we sincerely appreciate your helpful feedback.
>
>
> ***Reference:***
>
>
> [1]. Accurate medium-range global weather forecasting with 3d neural networks. Nature, 2023.
>
> [2]. Pangu-weather: A 3d high-resolution model for fast and accurate global weather forecast. arXiv:2211.02556.
>
>
> ### ***Q1: About Euler solver against higher-order numerical integrators***
>
>
> Thank you for this constructive suggestion, and we appreciate your insightful expertise on ODE solvers. In response, we have conducted a targeted ablation study as follows.
>
> **Table R2: Comparison of Euler and RK4 solvers for DeepPrim in 36h and 72h global weather forecasting**
>
> |Variable|Lead time|RMSE||Acc||
> |-|-|-|-|-|-|
> |||DeepPrim (Euler)*|DeepPrim (RK4)*|DeepPrim (Euler)*|DeepPrim (RK4)*|
> |z500|36|160.7|154.8|0.99|0.99|
> |z500|72|331.9|311.7|0.94|0.95|
> |t850|36|1.19|1.10|0.97|0.97|
> |t850|72|1.78|1.79|0.94|0.94|
> |t2m|36|1.21|1.19|0.97|0.97|
> |t2m|72|1.57|1.55|0.95|0.95|
> |u10|36|1.60|1.50|0.93|0.94|
> |u10|72|2.44|2.44|0.82|0.82|
> |v10|36|1.64|1.65|0.92|0.92|
> |v10|72|2.43|2.44|0.80|0.80|
>
> In our implementation, we support both Euler and Runge-Kutta (RK4) schemes and other ODE solvers (through $\texttt{torchdiffeq}$) for discretizing the ODEs, and we previously used the Euler method to solve the ODE system, as it is simple, efficient, and adequate for short-term forecasting. **Compared to the simple Euler method, RK4 may require more computational effort but offers higher-order accuracy, improved stability, and better control of local truncation error**, which can help capture the complex temporal dynamics in atmospheric modeling, thereby helping mitigate error accumulation, especially for medium- and long-range forecasts. Exploring these advanced time-stepping methods for longer-term forecasting remains a promising direction for future work. The relevant discussion is involved in $\underline{\text{Appendix H on Page 32}}$.
>
> The new experiments confirm that, compared to Euler, RK4 provides higher-order accuracy by improving the stability, leading to slightly numerical error and marginal improvements in forecast skill at 36 h and 72 h. However, these gains are not large enough to substantially close the performance gap at longer lead times, suggesting that the reduced performance is mainly attributable to the learned dynamics and the intrinsic difficulty of long-range forecasting, except for numerical instability of the Euler integrator. Also, as mentioned in our response to your W2, effective long-range prediction additionally benefits from training and inference strategies such as model ensembling, rolling forecasts, and autoregressive refinement. These techniques are orthogonal to our core architectural contribution and can, in principle, be combined with DeepPrim to further enhance its long-range robustness and mitigate cumulative error.

---

> ### Author Response · Authors · 2025-11-20
> **Response to Reviewer Rdm2 (Part 4)**
>
> ### ***Q2: About the physical interpretation and necessity for introducing the "intermediate atmospheric motion***
>
>
> Thanks for your insightful feedback, and we are glad to address it. Actually, the intermediate atmospheric motion $v^*$ is **not a new physical variable**, but a modeling trick introduced for modeling flexibility and numerical stability. Specifically,
>
> 1. **Decoupling motion/force learning from state evolution**: In DeepPrim, we aim to *explicitly learn the force terms* in the momentum equation (rather than directly predicting wind tendencies). Thus, $v^{\*}$ serves as an auxiliary state whose time derivative $\dot{v}^*$ is directly output by the Force Network (Eq. 5). The actual wind components are part of the prognostic state $\mathbf{u}$, and their final prediction is refined by the Source-Sink Network (Eq. 8).
>
> 2. **Enhanced initial condition estimation**: The true initial wind field $[v_x, v_y]$ may not be optimal for ODE integration due to unresolved subgrid influences. The Initialization Network (Eq. 7) computes a *residual correction* using 3D spatial gradients ($\nabla u$) and temporal variations ($\dot u$), yielding a more physically consistent $v^*$ .
>
> **In essence**, $v^*$ enables the model to learn a more accurate *effective motion field* that better aligns with the learned force dynamics, improving ODE integration stability without violating physical principles.
>
>
>
> ### ***Q3: How is the advection term computed?***
>
> We appreciate your detailed review of our technical implementations. We'd like to clarify that the **advection term is computed explicitly** and is **not learned implicitly** by the network. Specifically ($\underline{\text{Algorithm 1, Line 7, on Page 15}}$), for any prognostic variable, the time tendency includes an explicit advection term:
>  $$
>  −(v^* ⋅ \nabla_p u + \omega \partial u/ \partial p),
>  $$
> where $v^*$ is the intermediate motion field, $\nabla_p u$ is the horizontal gradient (implemented with `torch.gradient()`, a second-order differential operator), $\omega$ is the vertical velocity in pressure coordinates, derived *explicitly* from the continuity equation (Eq. 4), where the integration is implemented with torch.cumsum():
>  $$
>  \omega=-\int\left(\frac{\partial v_{x}}{\partial x}+\frac{\partial v_{y}}{\partial y}\right) \mathrm{d} p
>  $$
>
> **The neural networks only model non-advective terms**, where **Force Network**: Learns the forces (pressure gradient, Coriolis, friction), and **Source-Sink Network**: Learns *non-advective source/sink processes* (radiative heating, latent heat release, etc.). This design ensures that **advection—the dominant transport mechanism in atmospheric dynamics—is treated with strict numerical and physical fidelity**, while only the more uncertain subgrid processes are delegated to data-driven learning.
>
>
> We sincerely appreciate your careful review of our paper and the technical details. Hope our response can address your concerns.

---

### Official Review · Reviewer_ZswK · 2025-11-01

**Soundness:** 1
**Presentation:** 1
**Contribution:** 2
**Rating:** 2
**Confidence:** 4

**Summary:**

Physics-informed modeling of weather has recently gained attention. Building on the existing work on neural advection, this work advocates modeling 3D dynamics of weather evolution with both inter- and intra-pressure-level interactions pertaining to upper-air and surface variables, drawing inspiration from the Navier-Stokes equation. A bicomponent vision transformer architecture is invoked for this purpose. Experimental focusing on short-term global and regional weather forecasting tasks are also provided.

**Strengths:**

--- Weather modeling has clearly emerged as an important area of research within the ML community, so the topic is of broad interest.

--- Incorporating additional physics-inspired priors (e.g., using pressure gradients), to augment data-driven pipelines can potentially provide a useful inductive bias.

**Weaknesses:**

--- Novelty of the proposed approach is limited. Essentially, with the exception of  a component called 'force network' that models external forces such as pressure gradients, the entire approach is directly adopted from ClimODE. Surprisingly, in the entire paper, the authors fail to acknowledge ClimODE for proposing and implementing neural advection as a fundamental principle for weather modeling.

--- The implementation is not consistent with the formulation. For instance, equation 5 describes a source-sink network operating on  \dot{v} (the gradient of intermediate atmospheric motion), but in practice, a ClimODE-style approach is adopted, wherein the network is applied on ODE predictions instead of ˙\dot{v}˙.

--- It is unclear how the pressure-based information is handled in practice. The dataset contains a few variables pertaining to pressure, and estimating the gradient with respect to pressure (equation 5) would necessitate segregating the equations for each variable. Absent this, information leakage due to correlations between pressure gradients across different variables cannot be ruled out.

--- The evaluation and reporting of results (e.g., in Table 2) is also unsatisfactory, and oftentimes, even misleading. For instance, it is mentioned in Appendix D that "In line with (Nguyen et al., 2023), we selected 6 atmospheric variables at 7 pressure levels, 3 surface variables, and 3 static variables for the ERA5 dataset with 5.625° resolution, as detailed in Table 14. In our model training, we choose all variables as input variables, and all variables except three static variables as output variables that are used for loss calculation." This is in contrast to ClimODE that reported use of only five variables in total, namely {z, t2m, u10, v10, t at 500hPa}, for training.

--- Comparisons are also unfair in terms of size of data used for training. DeepPrim used data from 1979–2015 for training, while ClimODE was trained with only 10 years of data.

Overall, the methodology (including evaluation) falls considerably short of expected standards.

**Questions:**

Could the authors please address my concerns mentioned in the Weaknesses section?

---

> ### Author Response · Authors · 2025-11-20
> **Response to Reviewer ZswK (Part 1)**
>
> Dear Reviewer ZswK:
>
> We deeply appreciate your positive recognition of our investigated physics-informed weather modeling and the physics-inspired priors. We are especially grateful for the detailed and insightful review provided, which is immensely beneficial in enhancing the overall quality of our work. Here we are glad to address your concerns point by point as follows.
>
>
> ### ***W1: About ClimODE and our work's novelty***
>
> Thanks for raising this point, and we appreciate your expertise in physis-infomed weather forecasting. There is no doubt that **ClimODE is indeed a pioneering work**, and actually, our interest in physics‑informed deep weather forecasting was partly influenced by the ICLR oral presentation “Climate and Weather Forecasting with Physics‑Informed Neural ODEs.” We acknowledge that our original submission may not sufficiently emphasize ClimODE’s contribution, and we have revised the text to more clearly and explicitly credit this contribution in $\underline{\text{Section 2: Related Works on Page 3}}$.
>
> Also, we'd like to clarify our work's **Conceptual and technical differences of ClimODE**:
>
> (1). Our work builds on the idea of neural advection from ClimODE but goes beyond a purely continuity-based formulation by using a primitive-equation–inspired Neural ODE that evolves multiple prognostic variables within a shared, physics-motivated representation.
>
> (2). Recognizing the importance of explicitly modeling the 3D structure of the atmosphere, this dynamics is instantiated with a 3D bi-directional ViT backbone for multi-level atmospheric fields and an explicit force network for pressure-gradient and other source terms, so that physics is embedded as the core forecasting backbone.
>
> (3). Experimentally, in comparison with ClimODE that mainly focuses on experiments on $5.625°$ ERA5 datasets, we instantiate our framework on ERA5 datasets at $1.4°$, and $0.25°$ resolutions and compare against a broader set of strong SOTA deep learning weather models, demonstrating the potential of physics-informed continuous-time modeling for high-resolution weather forecasting.
>
>
>
> ### ***W2: About the technical implementation of the source-sink network***
>
> Thank you for kindly raising this insightful question. We appreciate the opportunity to clarify this aspect of our design.
>
> As discussed in our physical analysis (in $\underline{\text{Section 3.1 on Pages 3 and 4}}$), modeling source and sink terms at each ODE step provides a theoretical framework that aligns with the well-established physical processes. However, in practice, **to enhance training stability and reduce the risk of overfitting, we chose to directly model the accumulated source and sink contribution over the entire forecast window**, rather than their instantaneous rates at each step. This approach offers practical benefits, including improved numerical stability during ODE integration and more robust generalization in data-driven learning. Conceptually, the Source-Sink network thus captures the accumulated effect of these processes from the initial to the forecast time, rather than serving as an additive term to the instantaneous temporal derivative.
>
> ### ***W3: How the pressure-based information is handled in practice***
>
> Thank you for this thoughtful question and care of the implementation of the pressure‑based information.
> In our implementation, the pressure coordinate is modeled as a discrete vertical axis, and except for few static variables live purely on the surface, all key prognostic upper‑air variables (e.g., temperature $t$, wind speed $u,v$, geopotential $z$, and humidity $q,r$) are available on multiple pressure levels, as summarized in $\underline{\text{Tables 13–14 on Pages. 24–25}}$. This provides sufficient vertical structure to define pressure‑based operators.
>
> Concretely, the pressure derivative $\partial(\cdot)/\partial p$ in Eq. (5) is implemented as part of the 3D gradient operator $\nabla = [\partial/\partial x, \partial/\partial y, \partial/\partial p]$ ($\underline{\text{Eq. (7) on Page 6, and Algorithm 1 on Page 15}}$), applied **variate‑wise** to each variable using finite differences along the pressure axis (e.g., computing the vertical gradient of temperature at different pressure levels). Thus, each variable’s pressure gradient is computed from its own vertical profile, and we do not share or mix pressure gradients across different variables, which avoids the type of information leakage you are concerned about. Pressure information is further injected in the learnable components through explicit pressure‑level embeddings and intra‑/cross‑level attention in the Force Net ($\underline{\text{Algorithm 2 on Page 16 and Fig. 6 on Page 19}}$).

---

> ### Author Response · Authors · 2025-11-20
> **Response to Reviewer ZswK (Part 2)**
>
> ### ***W4: The evaluation details of model input variables***
>
> We sincerely thank you for carefully examining our experimental setup and for pointing out this important issue. We'd like to clarify that our main ERA5‑5.625° experiments follow the setting used in ClimaX and several recent SOTA models (e.g., WeatherGFT, Pangu‑Weather): we use all variables at different pressure levels as model input. Also, as our method is designed to explicitly model 3D (multi‑level) and multi‑variable interactions within a primitive‑equation–inspired framework, which requires access to variables across all pressure levels. We agree that this is different from the original ClimODE setting, which uses only five variables $\{z500,t2m,u10,v10,t850\}$ as the model's input and output.
>
> To enable a more apples-to-apples comparison with ClimODE, we additionally train our model using exactly the same five variables as ClimODE, both as inputs and outputs. The corresponding results are now reported in the revised Table R1. As expected, this five‑variable setting performs worse than our full multi‑variable, multi‑pressure-level setup, because it no longer fully leverages cross‑level and cross‑variable information. However, under the same five‑variable configuration, our method still outperforms ClimODE, indicating that the performance gains are not solely due to using more variables, but also stem from our primitive‑equation–inspired Neural ODE design.
>
> **Table R1: Comparison of ClimODE and DeepPrim under different input variables based on $5.625°$ ERA5 datasets with RMSE**
>
> |Variable|Lead Time|ClimODE (5 variable input)|DeepPrim* (5 variable input)|DeepPrim* (full variable input)|
> |-|-|-|-|-|
> |z500|6|102.9|69.5|53.6|
> ||12|134.8|98.9|77.4|
> ||18|162.7|129.8|102.2|
> ||24|193.4|159.6|126.7|
> |t850|6|1.16|0.81|0.72|
> ||12|1.32|0.98|0.87|
> ||18|1.47|1.04|0.98|
> ||24|1.55|1.16|1.06|
> |t2m|6|1.21|0.79|0.73|
> ||12|1.45|0.97|0.88|
> ||18|1.43|1.02|0.97|
> ||24|1.4|1.08|0.99|
> |u10|6|1.41|1.02|0.82|
> ||12|1.81|1.13|1.0|
> ||18|1.97|1.33|1.15|
> ||24|2.01|1.51|1.25|
> |v10|6|1.53|1.03|0.85|
> |12|1.81|1.19|1.03|
> ||18|1.96|1.35|1.19|
> ||24|2.04|1.52|1.24|
>
> ### ***W5: More apples-to-apples comparison results***
>
>
> Thank you for your careful and thorough assessment of our experimental setup, and we appreciate the opportunity to clarify the data usage in our comparisons.
>
> In our main ERA5 experiments, we follow the ClimaX protocol, i.e., we use data from 1979–2015 for training, 2016 for validation, and 2017–2018 for testing. This choice is intended to enable more comprehensive comparisons with recent large‑scale data‑driven weather models that use long training periods.
>
> We are aware that ClimODE was originally trained using only 10 years of data. To make the comparison fairer, we reimplemented ClimODE and retrained it under the same data setting as ours (1979–2015 for training, 2016 for validation, and 2017–2018 for testing). As expected, when given more years of data, ClimODE’s performance improves, which is consistent with scaling‑law observations in data‑driven forecasting. However, even under this strengthened baseline, our method still achieves better performance. This suggests that the gains are not solely due to using a longer training period, but also stem from our primitive‑equation–inspired Neural ODE with explicit 3D multi‑level modeling and force network design.
>
>
> **Table R2: Comparison of ClimODE and DeepPrim under different input variables and different training years based on $5.625°$ ERA5 datasets with RMSE**
> |Variable|Lead Time|ClimODE (10 year for training)|ClimODE (1979–2015 for training)|DeepPrim* (5 variable input)|DeepPrim* (full variable input)|
> |--------|---------|-----------------|--------------------------------|----------------------------|-------------------------------|
> |z500|6|102.9|79.1|69.5|53.6|
> ||12|134.8|111.2|98.9|77.4|
> ||18|162.7|148.9|129.8|102.2|
> ||24|193.4|192.8|159.6|126.7|
> |t850|6|1.16|1.01|0.81|0.72|
> ||12|1.32|1.23|0.98|0.87|
> ||18|1.47|1.4|1.04|0.98|
> ||24|1.55|1.52|1.16|1.06|
> |t2m|6|1.21|1.13|0.79|0.73|
> ||12|1.45|1.36|0.97|0.88|
> ||18|1.43|1.54|1.02|0.97|
> ||24|1.4|1.75|1.08|0.99|
> |u10|6|1.41|1.15|1.02|0.82|
> ||12|1.81|1.41|1.13|1.0|
> ||18|1.97|1.68|1.33|1.15|
> ||24|2.01|1.83|1.51|1.25|
> |v10|6|1.53|1.21|1.03|0.85|
> ||12|1.81|1.47|1.19|1.03|
> ||18|1.96|1.65|1.35|1.19|
> ||24|2.04|1.86|1.52|1.24|
>
>
> Following your valuable suggestions, we have added a new section, **Appendix G.6: More Experiments on``Apple-to-apple'' Comparison with ClimODE** on $\underline{\text{ Pages 31, 32 and 33}}$ together with
> the experimental results summarized in $\underline{\text{Table 22 on Page 33}}$, where we conduct additional experiments to enable a more rigorous and fair comparison with ClimODE with according discussions. The revisions are marked in red.
>
> Again, we appreciate your expertise in physics-informed weather forecasting. Thank you again for the careful review of our paper, and we hope our response addresses your concerns.

---

### Official Review · Reviewer_k1Xf · 2025-11-01

**Soundness:** 3
**Presentation:** 2
**Contribution:** 2
**Rating:** 6
**Confidence:** 4

**Summary:**

The paper presents DeepPrim, a physics-driven 3D deep learning framework for short-term weather forecasting. DeepPrim combines the learning of primitive equations—including the 3D Navier-Stokes equations for atmospheric dynamics—with data-driven neural modules, specifically a 3D bicomponent Vision Transformer (3D-BiViT) to capture both horizontal and vertical (pressure-level) interactions, source-sink networks for unresolved processes, and explicit ODE solvers for continuous-time forecasting. Extensive experiments on global and regional ERA5 datasets show that DeepPrim outperforms recent deep learning and NWP baselines in RMSE and demonstrates improved physical fidelity in 3D temporal weather modeling

**Strengths:**

1. Integration of Physics and Deep Learning: DeepPrim thoughtfully incorporates the full set of primitive atmospheric equations—including the 3D Navier-Stokes equation in pressure coordinates—into a neural architecture
2. Comprehensive Ablations and Interpretability: The authors ablate every core module—initialization, source-sink, and especially 3D pressure-coupling
3. Architectural Innovation for 3D Coupling: The force network introduces explicit intra-pressure and cross-pressure attention mechanisms, directly modeling vertical and horizontal dynamics.

**Weaknesses:**

1. Limited Experimental Scope for Longer-term Forecasting: Despite strong short-term (sub-24h) results, the paper’s focus on short-range forecasting leaves a gap in evaluating DeepPrim’s stability and error accumulation at multi-day or weekly horizons.
2. Insufficient Comparison on Precipitation and Rare-event Metrics: Although DeepPrim is evaluated on standard atmospheric state variables (z500, t850, t2m, u10, v10), it lacks explicit performance metrics for short-term precipitation, extreme wind/storm events, or severe weather proxies restricts insight into the practical significance for domains that are most sensitive to rare or impactful events.
3. Absence of Explicit Uncertainty Quantification: Modern operational forecasting demands not just pointwise predictions, but calibrated uncertainty—either through ensembles or explicit probabilistic modeling. DeepPrim delivers deterministic outputs, and there is no discussion of uncertainty, confidence intervals, or systematic error analysis.

**Questions:**

1. Out-of-Sample Evaluation: Have the authors tested DeepPrim on years/geographies not included in the training/discussed regions, or on different reanalysis datasets (e.g., JRA-55, MERRA-2), to further demonstrate generalization capabilities?
2. Physical Interpretability of the Source-Sink Module: Please elaborate on how the source-sink network is parameterized. Can its outputs be mapped back to interpretable physical processes? Are particular source/sink terms discoverable in the learned model, and how does this inform trust in the system for operational forecasters?
3. Long-range Forecasting and Accumulated Error: Can the authors provide quantitative results for forecast horizons >24h, using the full DeepPrim model, to assess whether short-term gains remain stable at longer lead times? If degradation is significant, what additional mitigations (e.g., rolling initialization, explicit error correction) might be needed?

---

> ### Author Response · Authors · 2025-11-20
> **Response to Reviewer  k1Xf (Part 1)**
>
> Dear Reviewer k1Xf:
>
> We sincerely thank you for your positive acknowledgment of our work's physics-deep learning integration, 3D architectural innovation, and comprehensive experimental validation. We deeply appreciate your detailed and perceptive feedback. Rest assured, we are committed to addressing your concerns.
>
>
> ### ***W1&Q3: About long lead-time evaluation***
>
> Thank you for your constructive suggestion. We fully agree that long-range forecasting is an important research direction, as demonstrated by several outstanding works such as Pangu and GraphCast. In contrast, **our work is primarily motivated by the demands of short-term forecasting, which is highly relevant for many practical applications [1][2] such as renewable energy prediction** (e.g., wind and solar power integration) [3], **event scheduling and management** (e.g., outdoor sports or public gatherings), and **aviation planning** (e.g., flight routing and safety) [4]. In this context, **3D modeling and physics-informed continuous-time approaches are particularly critical**, so we aim to address this specific research gap.
>
> While our main paper focuses on short-term results, we have included some medium and long-range forecasting experiments in $\underline{\text{Table 18 and Appendix G.2 on Page 29}}$. For your convenience, we'd like to present the results below.
>
> **Table R1:Long lead time prediction results of global weather forecasting based on $5.625°$ ERA5
> dataset.**
>
> |Variables|Lead-Time Δt (h)|IFS|ClimaX*|ClimaX|ClimODE|DeepPrim|DeepPrim*|IFS|ClimaX*|ClimaX|ClimODE|DeepPrim|DeepPrim*|
> |---|---|---|---|---|---|---|---|---|---|---|---|---|---|
> | | | |**RMSE**| | | | | |**ACC**| | | | |
> |z500|36|66.7|455.0|126.4|259.6|154.3|160.7|1.00|0.89|1.00|0.96|0.99|0.99|
> ||72|147.0|687.0|244.1|478.7|319.4|331.9|0.98|0.73|0.97|0.88|0.95|0.94|
> ||144|430.3|801.9|523.5|783.6|558.7|574.9|0.86|0.58|0.86|0.61|0.81|0.80|
> |t850|36|0.83|2.49|1.25|1.75|1.27|1.19|0.97|0.86|0.97|0.94|0.96|0.97|
> ||72|1.19|3.17|1.59|2.58|1.88|1.78|0.94|0.76|0.98|0.85|0.94|0.94|
> ||144|2.3|3.97|2.54|3.62|2.75|2.64|0.77|0.69|0.84|0.77|0.82|0.84|
> |t2m|36|0.95|2.87|1.33|1.70|1.49|1.21|0.93|0.83|0.97|0.94|0.96|0.97|
> ||72|1.15|3.97|1.43|2.75|1.88|1.57|0.90|0.83|0.98|0.85|0.95|0.95|
> ||144|1.83|3.38|2.01|3.30|2.78|2.60|0.75|0.38|0.92|0.79|0.83|0.85|
> |u10|36|1.07|2.98|1.57|2.25|1.70|1.60|0.96|0.69|0.93|0.83|0.93|0.93|
> ||72|1.69|3.70|2.18|3.19|2.55|2.44|0.90|0.30|0.94|0.66|0.81|0.82|
> ||144|3.17|4.24|3.24|4.02|3.65|3.55|0.66|0.30|0.63|0.35|0.55|0.57|
> |v10|36|1.12|2.98|N/A|2.29|1.75|1.64|0.96|0.69|N/A|0.83|0.92|0.92|
> ||72|1.75|3.80|N/A|3.30|2.58|2.43|0.90|0.39|N/A|0.63|0.79|0.80|
> ||144|3.3|4.42|N/A|4.24|3.70|3.56|0.65|0.25|N/A|0.32|0.52|0.54|
>
>
>  Empowered by pre-training on the CMIP6 dataset, ClimaX shows relatively stable performance for forecasting at longer lead time, as pre-training lays a great foundation for post-training and degrades the difficulty of long-lead time forecasting. Specifically, our DeepPrim, despite not being pre-trained on any external data, presents comparable results in forecasting $t850$ $t2m$, and $u10$ in comparison with pre-trained ClimaX. Moreover, our DeepPrim consistently achieves better performance than the non-pre-trained ClimaX $^{*}$ and ClimODE.
>
> Also, **we fully agree with your insights** that `"additional mitigations (e.g., rolling initialization, explicit error correction) might be needed"`. In fact, as pointed out in recent work [5][6], accurate long-term forecasting often requires **specialized training techniques, model ensemble strategies (e.g., OneForecast [5]), and autoregressive refinement (e.g., FuXi [6])**
>
> We'd like to note that **these aspects are beyond the primary scope of our current work, but they are orthogonal and complementary directions**. Empowering our approach with these techniques could further enhance medium- and long-range forecasting of our DeepPrim, which we see as a promising avenue for future research.
>
> In light of your feedback, we have also supplemented further discussion on enhancement for long lead-time forecasting in $\underline{\text{Appendix H on Pages 32 and 33}}$ of the revised paper.
>
> *Reference:*
>
>
> [1]. Radar refractivity retrieval: Validation and application to short-term forecasting. Journal of Applied Meteorology, 2005.
>
> [2]. A dynamic convolutional layer for short-range weather prediction. CVPR, 2015.
>
> [3]. Fusionsf: Fuse heterogeneous modalities in a vector quantized framework for robust solar power forecasting. KDD, 2024.
>
> [4]. No more flying blind: Leveraging weather forecasting for clear-cut risk-based decisions. Transportation Research Interdisciplinary Perspectives, 2025.
>
> [5]. OneForecast: A Universal Framework for Global and Regional Weather Forecasting. ICML, 2025.
>
> [6]. Fuxi: A cascade machine learning forecasting system for 15-day global weather forecast. npj Climate and Atmospheric Science, 2023.

---

> ### Author Response · Authors · 2025-11-20
> **Response to Reviewer k1Xf (Part 2)**
>
> ### ***W2: Precipitation and Rare-event Metrics***
>
>
> Thanks for pointing out the importance of precipitation and rare-event metrics. We'd like to clarify that our current evaluation protocol follows the widely adopted setup of ClimaX [7] and ClimODE [8], using the WeatherBench-processed ERA5 dataset and focusing on standard atmospheric state variables. Furthermore, we conduct a more systematic study across multiple spatial resolutions (from $5.625°$, $1.4°$ to $0.25°$), which allows us to assess the robustness and scalability of DeepPrim under a broad range of grid configurations.
>
> We fully agree that short-term precipitation and severe-weather-related metrics (e.g., extreme wind and storm events) are of high practical importance. Accurately capturing these phenomena typically requires more detailed microphysical parameterizations and additional physical priors beyond those considered in our current primitive-equation-based formulation. Extending DeepPrim with explicit moist-physics priors and precipitation-related outputs is, therefore, a natural and promising direction for future work, and we will clarify this limitation and outlook in the revised manuscript. We have also supplemented improvements on enhancement for precipitation nowcasting in $\underline{\text{Appendix H on Page 32}}$. The revisions are marked in red.
>
> Regarding your concern about **rare-event metrics**, here we provide additional results on the ERA5 test set using the Anomaly Correlation Coefficient (ACC), a standard metric in meteorology and climate science to assess a model’s skill in capturing anomalies (deviations from the climatological mean). Specifically, we compute
>
> $\text{ACC} = \frac{\sum\_{k,h,w}\tilde{u}^{\prime}\_{k, h, w} u^{\prime}\_{k, h, w}}{\sqrt{\sum\_{k,h,w} \alpha(h) (\tilde{u}^{\prime}\_{k, h, w})^2 \sum\_{k,h,w} \alpha(h) (u^{\prime}\_{k, h, w})^2 }}$
>
>
> where $u^{\prime} = u - C $ and $\tilde{u}^{\prime} = \tilde{u} - C $, with $ C = \frac{1}{K}\sum_k \tilde{u}_k $ representing the temporal mean of the ground truth over the test set.
>
>
>
> **Table R2:Global weather forecasting results based on ACC evaluation metric.**
>
> |Variable|Δt (h)|IFS|NODE|FCN|ClimaX*|ClimaX|ClimODE|DeepPrim†|DeepPrim|
> |---|---|---|---|---|---|---|---|---|---|
> |z500|6|1.00|0.96|0.99|0.97|1.00|0.99|1.00|1.00|
> ||12|(N/A)|0.88|0.99|0.96|1.00|0.99|1.00|1.00|
> ||18|(N/A)|0.79|0.99|0.95|1.00|0.98|1.00|1.00|
> ||24|1.00|0.70|0.99|0.93|1.00|0.98|0.99|1.00|
> |t850|6|0.99|0.94|0.99|0.94|0.98|0.97|0.99|0.98|
> ||12|(N/A)|0.85|0.99|0.93|0.98|0.96|0.99|0.98|
> ||18|(N/A)|0.77|0.99|0.92|0.98|0.96|0.98|0.98|
> ||24|0.99|0.72|0.99|0.90|0.98|0.95|0.98|0.97|
> |t2m|6|0.99|0.82|0.99|0.92|0.98|0.97|0.99|0.99|
> ||12|(N/A)|0.68|0.99|0.90|0.97|0.96|0.99|0.98|
> ||18|(N/A)|0.69|0.99|0.88|0.97|0.96|0.98|0.97|
> ||24|0.99|0.79|0.99|0.89|0.98|0.96|0.98|0.97|
> |u10|6|0.98|0.85|0.95|0.92|0.97|0.91|0.98|0.97|
> ||12|(N/A)|0.70|0.93|0.88|0.95|0.89|0.97|0.96|
> ||18|(N/A)|0.58|0.91|0.84|0.95|0.88|0.96|0.95|
> ||24|0.97|0.50|0.89|0.80|0.94|0.87|0.95|0.94|
> |v10|6|0.98|0.81|0.94|0.92|(N/A)|0.92|0.98|0.97|
> ||12|(N/A)|0.61|0.91|0.88|(N/A)|0.89|0.97|0.96|
> ||18|(N/A)|0.46|0.86|0.83|(N/A)|0.88|0.96|0.95|
> ||24|0.97|0.35|0.83|0.80|(N/A)|0.86|0.95|0.94|
>
> Anomaly Correlation Coefficient (ACC) measures a model’s ability to predict deviations from the mean. Higher ACC values indicate better accuracy in capturing anomalies, which is crucial in meteorology and climate science. As shown in the table, DeepPrim also shows strong prediction accuracy in ACC.
>
> According to your suggestion, we have added the experimental results based on ACC metric in $\underline{\text{Appendix G.1 and Table. 17 on Page 28}}$.
>
>
>
> ### ***W3: About Explicit Uncertainty Quantification***
>
> Thanks for your insightful question on uncertainty quantification. We fully agree that calibrated uncertainty (e.g., via ensembles or explicit distributional outputs) is crucial for operational deployment and risk-aware decision-making. In $\underline{\text{Appendix H on Page 31}}$, we discussed how DeepPrim can be extended to support uncertainty quantification while preserving its physics-guided ODE structure. In particular, a feasible direction is to reparameterize the Source–Sink Network to output distributional parameters (e.g., mean and variance) instead of only deterministic tendencies, which would enable sampling-based uncertainty estimation on top of the learned dynamical system. This probabilistic extension, together with ensemble-style perturbations of initial states or forcing terms, represents an important avenue for future work toward uncertainty-aware, operationally relevant deployments of DeepPrim.

---

> ### Author Response · Authors · 2025-11-20
> **Response to Reviewer k1Xf (Part 3)**
>
> ### ***Q1: Out-of-Sample Evaluation***
>
> We thank the reviewer for highlighting the OOS evaluation and generalization across time and geography.  In this work, our experimental setup follows the widely adopted WeatherBench/ERA5 protocol used by ClimaX [7] and ClimODE [8], which allows for a direct and fair comparison with strong baselines under a shared setting.
>
> Our experimental setting aligns with ClimaX, i.e., we use 1979–2015 for training, 2016 for validation, and 2017–2018 for testing, so the test period is intrinsically out-of-sample in time and already reflects some degree of climatic non-stationarity (e.g., due to long-term climate change). Beyond these prior works, we further conduct a more comprehensive evaluation across multiple spatial resolutions (5.625°, 1.4° and 0.25°) and include dedicated regional forecasting experiments over North America, South America, and Australia ($\underline{\text{Section 4.2, Page 8}}$), providing additional evidence that DeepPrim generalizes across distinct geophysical regimes and grid configurations within ERA5.
>
> *Reference:*
>
> [7]. ClimaX: A foundation model for weather and climate. ICML, 2023
>
> [8]. Climode: Climate and weather forecasting with physics-informed neural odes. ICLR, 2024.
>
>
>
> ### ***Q2: Physical Interpretability of the Source-Sink Module***
>
>
> We thank the reviewer for highlighting the importance of the physical interpretability of the Source–Sink module. In DeepPrim, this network is embedded in the primitive‑equation ODE and is explicitly designed to model only non‑advective tendencies, analogous to physical parameterizations in NWP (radiative heating, boundary‑layer mixing, moist processes, surface fluxes). Given the initial state $u_{t_{0}}$, the intermediate flow field $v^{*}\_{t\_{0}}$, the ODE‑integrated “dynamics‑only’’ forecasts $\{ \mathbf{u}^{\prime}( t\_i ) \}\_{i=1}^{N}$ and a space–time embedding  $\psi\_{st}$ (latitude/longitude, diurnal, seasonal cycles), the Source–Sink Network (a compact ResNet‑style CNN) outputs cumulative source–sink contributions $\{ \mathbf{u}\_{source-sink} ( t\_i ) \} \_{i=1}^{N}$, and the final prediction is $\{ \mathbf{\hat{u}}( t\_i ) \} \_{i=1}^{N} =\{ \mathbf{u}^{\prime}( t\_i ) \}\_{i=1}^{N}+\{ \mathbf{u}\_{source-sink} ( t\_i ) \}\_{i=1}^{N} $  (see $\underline{\text{ Sec. 3.2.3 and Appendix B.2/B.8}}$), which structurally disentangles advective and non‑advective tendencies.
>
> As discussed in our physical analysis (in $\underline{\text{Section 3.1 on Pages 3 and 4}}$), modeling source and sink terms at each ODE step provides a theoretical framework that aligns with the well-established physical processes. However, in practice, **to enhance training stability and reduce the risk of overfitting, we chose to directly model the accumulated source and sink contribution over the entire forecast window**, rather than their instantaneous rates at each step. This approach offers practical benefits, including improved numerical stability during ODE integration and more robust generalization in data-driven learning. Conceptually, the Source-Sink network thus captures the accumulated effect of these processes from the initial to the forecast time, rather than serving as an additive term to the instantaneous temporal derivative.
>
>
> Additionally, our ablations (in $\underline{\text{Figure. 3 on Page. 8}}$) show that removing the Source–Sink module degrades diurnal cycles and increases long‑range error accumulation, indicating that it captures physically meaningful non‑advective processes rather than arbitrary numerical corrections.
>
>
> Thanks again for your professional rigor and careful review. Hope our response addresses your concerns.

---

### Official Review · Reviewer_axZ9 · 2025-11-06

**Soundness:** 3
**Presentation:** 3
**Contribution:** 2
**Rating:** 6
**Confidence:** 4

**Summary:**

The authors highlight limitations of both numerical weather prediction (NWP) methods and deep learning weather prediction (DLWP) models. They propose DeepPrim that learns primitive equations of the Earth's atmosphere. DeepPRIM models 3D atmospheric motion using the Navier-Stokes equation for the pressure and captures the interactions between the advective and other variables.

This is a good applied paper but is lacking some algorithmic novelty since it is using a standard Vision Transformer.

**Strengths:**

- Nice combination of physical/numerical methods within deep learning methods.
- Authors clearly identify 3 main challenges with current DLWP methods
- Nice emphasis on the importance of physical constraints
- Strong results of RMSE improvement by ~35-38%
- Nice method overview in Table 1.
- Nice use of RK4 instead of Euler time-stepping
- Nice use of ERA dataset at various resolutions over various predictive variables, e.g., geopotential, air temperature, wind speeds.
- Strong results in most cases.
- Good comparisons to IFS, Pangu-Weather and GraphCast in Table 3.
- Nice that the method can learn the BCs
- Nice ablation studies

**Weaknesses:**

- Missing reference to DLWP benchmarking paper across the various SOTA methods Karlbauer et al., "Comparing and contrasting deep learning weather prediction backbones on navier-stokes and atmospheric dynamics", 2024.
- Missing references on physical constraints in deep learning models for PDEs especially to hard-constrained methods:
  -  Negiar et al., "Learning differentiable solvers for systems with hard constraints", ICLR, 2023
  - Chalapathi et al., "Scaling physics-informed hard constraints with mixture-of-experts", ICLR 2024
   - Hansen et al., "Learning Physical Models that Can Respect Conservation Laws", ICML 2024
    - Mouli et al., "Using uncertainty quantification to characterize and improve out-of-domain learning for pdes", ICML, 2024
  - Saad et al., "Guiding continuous operator learning through Physics-based boundary constraints", ICLR, 2023
   - Utkarsh, U., "End-to-End Probabilistic Framework for Learning with Hard Constraints", https://arxiv.org/pdf/2506.07003?, 2025.
- Metrics other than point-wise ones should be considered, e.g., probabilistic ones like CRPS or a constraint/physics-based metric

**Questions:**

1. The authors mention that the proposed method does well on short-term prediction. How would it extend to longer term weather prediction?
2. What made the authors choose a Vision Transformer backbone?
4. The ODE in Eqn, 6 looks like it is just solved with Neural ODE?
5. Do the authors know why some of the other baselines do better on the z500 variable?
6. Do the authors know why GraphCast has better performance in several cases in Table 3?

---

> ### Author Response · Authors · 2025-11-20
> **Response to Reviewer axZ9 (Part 1)**
>
> Dear reviewer axZ9:
>
> We sincerely appreciate your encouraging feedback and positive acknowledgment of our work's motivation, physics insights, technical design, and experiments. We are especially grateful for the constructive and insightful review provided, which is critical in enhancing our paper. Rest assured, we are dedicated to addressing your concerns.
>
> ### ***W1&2: More Related Reference on DLWP and PDEs***
>
> Thank you for pointing out these important references. We greatly appreciate your expertise in DLWP benchmarking and, specifically, physical constraints in deep learning for PDE. In light of your suggestions, we've carefully revised the Related Work section. The revisions are marked in red. Specifically:
>
> * **DLWP benchmarking paper.** We have explicitly cited and discussed ref [1] in $\underline{\text{Section II Related Works on Page 3}}$. We highlight how their systematic comparison of DL backbones for Navier–Stokes and atmospheric dynamics provides a useful reference point for backbone choices in data-driven weather prediction.
>
> * **Physical constraints and hard-constrained PDE methods.** We have added a new subsection **“Deep Learning for PDEs”** in the $\underline{\text{Section II Related Work on Page 3}}$, where the reference recommended by you [2,3,4,5,6,7] are carefully discussed. Also, we'd like to clarify that our work is complementary to these efforts: rather than enforcing strict hard constraints, we embed primitive-equation structure and continuous-time Neural ODE dynamics directly into the forecasting backbone, and view integrating such hard-constrained and probabilistic techniques into our framework as an exciting direction for future research.
>
> *Reference:*
>
> [1]. Comparing and contrasting deep learning weather prediction backbones on Navier-Stokes and atmospheric dynamics, 2024.
>
> [2].Learning differentiable solvers for systems with hard constraints, ICLR, 2023.
>
> [3]. Scaling physics-informed hard constraints with mixture-of-experts, ICLR 2024.
>
> [4]. Learning Physical Models that Can Respect Conservation Laws, ICML 2024.
>
> [5]. Using uncertainty quantification to characterize and improve out-of-domain learning for pdes, ICML, 2024.
>
> [6]. Guiding continuous operator learning through Physics-based boundary constraints, ICLR, 2023.
>
> [7]. End-to-End Probabilistic Framework for Learning with Hard Constraints, 2025.
>
>
> ### ***W3: Metrics other than point-wise ones***
>
> We thank the reviewer for this constructive suggestion. We agree that going beyond point-wise metrics and including additional skill measures can provide a more comprehensive evaluation.
>
> Due to space constraints, we mainly reported the RMSE results in the main text. Enlightened by your feedback, we have also supplemented the results based on the Anomaly Correlation Coefficient (ACC) metric in $\underline{\text{Appendix G.1 and Table. 17 on Page 28}}$ and marked them in red. ACC is a standard benchmark metric in meteorology and climate science for assessing the skill in predicting anomalies (deviations from climatology). ACC directly measures the ability of the model to capture anomaly patterns; higher ACC values indicate better anomaly prediction, which is crucial for assessing forecast skill beyond simple point-wise errors. Specifically,
>
> $\text{ACC} = \frac{\sum\_{k,h,w}\tilde{u}^{\prime}\_{k, h, w} u^{\prime}\_{k, h, w}}{\sqrt{\sum\_{k,h,w} \alpha(h) (\tilde{u}^{\prime}\_{k, h, w})^2 \sum\_{k,h,w} \alpha(h) (u^{\prime}\_{k, h, w})^2 }}$
>
> where $u^{\prime} = u - C $ and $\tilde{u}^{\prime} = \tilde{u} - C $, with $ C = \frac{1}{K}\sum_k \tilde{u}_k $ representing the temporal mean of the ground truth over the test set.
>
> For your convenience, here we present the results in Table R1 where DeepPrim also shows strong performance in ACC, showing its capacity in capturing anomalies and deviations from climatology.
>
> **Table R1:Global weather forecasting results based on ACC evaluation metric.**
>
> |Variable|Δt (h)|IFS|NODE|FCN|ClimaX*|ClimaX|ClimODE|DeepPrim†|DeepPrim|
> |-|-|-|-|-|-|-|-|-|-|
> |z500|6|1.00|0.96|0.99|0.97|1.00|0.99|1.00|1.00|
> ||12|(N/A)|0.88|0.99|0.96|1.00|0.99|1.00|1.00|
> ||18|(N/A)|0.79|0.99|0.95|1.00|0.98|1.00|1.00|
> ||24|1.00|0.70|0.99|0.93|1.00|0.98|0.99|1.00|
> |t850|6|0.99|0.94|0.99|0.94|0.98|0.97|0.99|0.98|
> ||12|(N/A)|0.85|0.99|0.93|0.98|0.96|0.99|0.98|
> ||18|(N/A)|0.77|0.99|0.92|0.98|0.96|0.98|0.98|
> ||24|0.99|0.72|0.99|0.90|0.98|0.95|0.98|0.97|
> |t2m|6|0.99|0.82|0.99|0.92|0.98|0.97|0.99|0.99|
> ||12|(N/A)|0.68|0.99|0.90|0.97|0.96|0.99|0.98|
> ||18|(N/A)|0.69|0.99|0.88|0.97|0.96|0.98|0.97|
> ||24|0.99|0.79|0.99|0.89|0.98|0.96|0.98|0.97|
> |u10|6|0.98|0.85|0.95|0.92|0.97|0.91|0.98|0.97|
> ||12|(N/A)|0.70|0.93|0.88|0.95|0.89|0.97|0.96|
> ||18|(N/A)|0.58|0.91|0.84|0.95|0.88|0.96|0.95|
> ||24|0.97|0.50|0.89|0.80|0.94|0.87|0.95|0.94|
> |v10|6|0.98|0.81|0.94|0.92|(N/A)|0.92|0.98|0.97|
> ||12|(N/A)|0.61|0.91|0.88|(N/A)|0.89|0.97|0.96|
> ||18|(N/A)|0.46|0.86|0.83|(N/A)|0.88|0.96|0.95|
> ||24|0.97|0.35|0.83|0.80|(N/A)|0.86|0.95|0.94|

---

> ### Author Response · Authors · 2025-11-20
> **Response to Reviewer axZ9 (Part 2)**
>
> ### ***Q1: Extend to longer-term weather prediction***
>
> Thank you for your constructive suggestion. We agree that long-range forecasting is an important research direction, as demonstrated by several outstanding works such as Pangu and GraphCast. In contrast, **our work is primarily motivated by the demands of short-term forecasting, which is highly relevant for many practical applications [8][9] such as renewable energy prediction** (e.g., wind and solar power integration) [10], **event scheduling and management** (e.g., outdoor sports or public gatherings), and **aviation planning** (e.g., flight routing and safety) [11]. In this context, **3D modeling and physics-informed continuous-time approaches are particularly critical**, so we aim to address this specific research gap.
>
> While our main paper focuses on short-term results, we have included some medium and long-range forecasting experiments in $\underline{\text{Table 18 and Appendix G.2 on Page 29}}$. For your convenience, we'd like to present the results below.
>
>
> **Table R2:Long lead time prediction results of global weather forecasting based on $5.625°$ ERA5
> dataset.**
>
> |Variables|Lead-Time Δt (h)|IFS|ClimaX*|ClimaX|ClimODE|DeepPrim|DeepPrim*|IFS|ClimaX*|ClimaX|ClimODE|DeepPrim|DeepPrim*|
> |---|---|---|---|---|---|---|---|---|---|---|---|---|---|
> | | | |**RMSE**| | | | | |**ACC**| | | | |
> |z500|36|66.7|455.0|126.4|259.6|154.3|160.7|1.00|0.89|1.00|0.96|0.99|0.99|
> ||72|147.0|687.0|244.1|478.7|319.4|331.9|0.98|0.73|0.97|0.88|0.95|0.94|
> ||144|430.3|801.9|523.5|783.6|558.7|574.9|0.86|0.58|0.86|0.61|0.81|0.80|
> |t850|36|0.83|2.49|1.25|1.75|1.27|1.19|0.97|0.86|0.97|0.94|0.96|0.97|
> ||72|1.19|3.17|1.59|2.58|1.88|1.78|0.94|0.76|0.98|0.85|0.94|0.94|
> ||144|2.3|3.97|2.54|3.62|2.75|2.64|0.77|0.69|0.84|0.77|0.82|0.84|
> |t2m|36|0.95|2.87|1.33|1.70|1.49|1.21|0.93|0.83|0.97|0.94|0.96|0.97|
> ||72|1.15|3.97|1.43|2.75|1.88|1.57|0.90|0.83|0.98|0.85|0.95|0.95|
> ||144|1.83|3.38|2.01|3.30|2.78|2.60|0.75|0.38|0.92|0.79|0.83|0.85|
> |u10|36|1.07|2.98|1.57|2.25|1.70|1.60|0.96|0.69|0.93|0.83|0.93|0.93|
> ||72|1.69|3.70|2.18|3.19|2.55|2.44|0.90|0.30|0.94|0.66|0.81|0.82|
> ||144|3.17|4.24|3.24|4.02|3.65|3.55|0.66|0.30|0.63|0.35|0.55|0.57|
> |v10|36|1.12|2.98|N/A|2.29|1.75|1.64|0.96|0.69|N/A|0.83|0.92|0.92|
> ||72|1.75|3.80|N/A|3.30|2.58|2.43|0.90|0.39|N/A|0.63|0.79|0.80|
> ||144|3.3|4.42|N/A|4.24|3.70|3.56|0.65|0.25|N/A|0.32|0.52|0.54|
>
>  Empowered by pre-training on the CMIP6 dataset, ClimaX shows relatively stable performance for forecasting at longer lead time, as pre-training lays a great foundation for post-training and degrades the difficulty of long-lead time forecasting. Specifically, our DeepPrim, despite not being pre-trained on any external data, presents comparable results in forecasting $t850$ $t2m$, and $u10$ in comparison with pre-trained ClimaX. Moreover, our DeepPrim consistently achieves better performance than the non-pre-trained ClimaX $^{*}$ and ClimODE.
>
> Additionally, in fact, as pointed out in recent work [12][13], accurate long-term forecasting often requires **specialized training techniques, model ensemble strategies (e.g., OneForecast [12]), and autoregressive refinement (e.g., FuXi [13])**
>
> We'd like to note that **these aspects are beyond the primary scope of our current work, but they are orthogonal and complementary directions**. Empowering our approach with these techniques could further enhance medium- and long-range forecasting of our DeepPrim, which we see as a promising avenue for future research.
>
> In light of your feedback, we have also supplemented further discussion on enhancement for long lead-time forecasting in $\underline{\text{Appendix H on Pages 32 and 33}}$ of the revised paper.
>
> *Reference:*
>
>
> [8]. Radar refractivity retrieval: Validation and application to short-term forecasting. Journal of Applied Meteorology, 2005.
>
> [9]. A dynamic convolutional layer for short-range weather prediction. CVPR, 2015.
>
> [10]. Fusionsf: Fuse heterogeneous modalities in a vector quantized framework for robust solar power forecasting. KDD, 2024.
>
> [11]. No more flying blind: Leveraging weather forecasting for clear-cut risk-based decisions. Transportation Research Interdisciplinary Perspectives, 2025.
>
> [12]. OneForecast: A Universal Framework for Global and Regional Weather Forecasting. ICML, 2025.
>
> [13]. Fuxi: A cascade machine learning forecasting system for 15-day global weather forecast. npj Climate and Atmospheric Science, 2023.

---

> ### Author Response · Authors · 2025-11-20
> **Response to Reviewer axZ9 (Part 3)**
>
> ### ***Q2: Reason for Choosing Vision Transformer backbone***
>
> We appreciate your insightful comment regarding our backbone design. Our decision is motivated both by modeling considerations and empirical evidence.
>
> **From a modeling perspective**, patch-based self-attention in a ViT backbone naturally captures global interactions in weather dynamics. Building on vanilla ViT, we further design a **3D Bi-directional ViT (3DBiViT)** tailored to provide separate but coupled attention along the horizontal and vertical (pressure) dimensions, enabling **explicit and customized intra‑ and inter‑level 3D interactions via a dual‑attention mechanism**.
>
> **From an empirical perspective**, we have carried out detailed ablations regarding backbone models in $\underline{\text{Appendix G.5 (Page 31) and Table 21 (Page 32)}}$. To directly resolve your concern, here we summarize the results from Table 20 in Table R3 below, including replacing its 3D-BiViT with a vanilla Vision Transformer (ViT), GNN, or CNN module.
>
> **Table R3: Additional ablation results of ForceNet in DeepPrim based on Latitude-weighted RMSE. The forecasting lead time is $\Delta t = 6h$.**
>
> |Model|Force Net|z500 ($m^{2}/s^{2}$)|t850 ($K$)|t2m ($K$)|u10 ($m/s$)|v10 ($m/s$)|
> |-|-|-|-|-|-|-|
> |DeepPrim|3D-BiViT|50.1|0.76|0.89|0.92|0.94|
> |Variant#1|Vanilia ViT|64.4|0.94|0.99|1.04|1.06|
> |Variant#2|GNN|69.3|0.91|0.98|1.01|1.02|
> |Variant#3|CNN(ResNet)|65.3|0.87|0.96|0.99|1.03|
>
> * **Additional Ablations and Analysis**: Specifically, ForceNet (3D-BiViT) is designed to learn the force terms in the Navier-Stokes equations, thereby effectively modeling 3D atmospheric dynamics by capturing both inter- and intra-pressure-level interactions for upper-air and surface variables in a coordinated way. In our ablation study, replacing 3D-BiViT with vanilla ViT, CNN, or GNN modules resulted in varying degrees of performance degradation. It is because **neither vanilla ViT, GNN, nor CNN explicitly models 3D interactions between pressure levels, which may lead to bias in estimating the velocity term $\dot{\mathbf{v}}^*$ and thus degrading the forecasting performance**. The additional ablations further demonstrate the necessity of our 3D design and its coherence with the overall model architecture.
>
>
> ### ***Q3: The ODE in Eqn. 6***
>
> We appreciate your insightful observation. **Yes**, the ODE in Eq. (6) is indeed solved using a Neural ODE formulation.
>
> Our motivation for explicitly adopting a Neural ODE is that continuous‑time integration is essential for our setting: it allows DeepPrim to (i) produce predictions at arbitrary lead times without retraining or changing the architecture, and (ii) evolve the state under a dynamics that is structurally aligned with the primitive equations. This continuous‑time view directly addresses challenges of physical consistency and temporal resolution in atmospheric modeling, as the model is trained to learn tendencies that are integrated in time rather than discrete frame‑to‑frame mappings.
>
>
> ***Q4&5: Discussions on z500 variable and GraphCast***
>
>
> Thank you for your insightful question regarding the model’s performance on Z500 and GraphCast. We'd like to provide several possible reasons for this:
>
> 1. **Methodological and Physical Factors**:
>
> In meteorology, Z500 is primarily influenced by large-scale global circulation patterns, which require effective modeling of long-range dependencies. GraphCast, for example, leverages a graph-based structure that explicitly encodes both local and global interactions in a hierarchical manner, providing a strong prior for capturing such patterns. In contrast, our current implementation focuses on continuous-time 3D dynamics and short-range processes, which may limit its ability to represent the most global features as effectively.
>
> 2. **Model Capacity and Scaling**:
>
> Due to computational resource constraints, our model currently uses a relatively shallow architecture (e.g., only one Transformer layer and 128 hidden dimensions in the 3D-BiViT module, as discussed and summarized in $\underline{\text{Appendix B.4 and Tables. 6-8 on Pages 17-19}}$ ). According to scaling laws observed in deep learning [16] and deep weather forecasting [14][15], increasing the model’s depth (e.g., layers) and width (e.g., hidden dimensions)would likely improve its ability to capture large-scale atmospheric structures such as Z500. We are happy to explore larger model variants in future work as resources permit.
>
>
> *Reference:*
>
> [14]. Accurate medium-range global weather forecasting with 3d neural networks. Nature, 2023.
>
> [15]. Pangu-weather: A 3d high-resolution model for fast and accurate global weather forecast. arXiv:2211.02556.
>
> [16].  Scaling Laws for Neural Language Models. arXiv:2001.08361,2020.
>
> Thank you again for your careful and professional review. We hope our response can address your concerns.

---

### Author Response · Authors · 2025-11-28
**Summary of Revisions**

We sincerely thank all the reviewers for their insightful reviews and valuable comments, which are instructive for us to improve our paper further.


**Pioneering the exploration of physics-informed 3D deep weather forecasting via primitive equation learning**, this paper presents DeepPrim, which effectively models atmospheric motion by integrating the Navier-Stokes equation with 3D-BiViT and captures weather evolution by leveraging the learned advection and source-sink model. Experimentally, **DeepPrim achieves impressive performance in both global and regional short-term weather forecasting across multiple resolutions ($0.25°\sim 5.625°$)** and exhibits superior capacity in capturing 3D atmospheric dynamics.

We are pleased that the reviewers generally held positive opinions of our work, in that DeepPrim is regarded as **"a novel 3D deep weather forecaster"** and **"highly original"** $\underline{\text{Reviewer Rdm2}}$; it **"pioneers the explicit learning of the Primitive Equations"** with **"physics-motivated inductive bias"** $\underline{\text{Reviewer Rdm2}}$; our study tackle a topic **"of broad interest"** $\underline{\text{Reviewer ZswK}}$; the method is seen as a **"physics-driven 3D deep learning framework"** and shows **"improved physical fidelity in 3D temporal weather modeling"** $\underline{\text{Reviewer k1Xf}}$; and the paper offers a **"nice combination of physical/numerical methods"**, **"strong results"**, and **"nice ablation studies"** $\underline{\text{Reviewer axZ9}}$.

The reviewers also raised insightful and constructive feedback. We made every effort to address all the concerns by providing detailed clarifications, sufficient evidence, requested results, and in-depth analysis. All the revisions are supplemented in our revised paper and marked in `red`. Here is the summary of the major revisions:

* **Add more *"Apples-to-apples"* experimental comparison with ClimODE** ($\underline{\text{Reviewer  ZswK}}$): we have supplemented more experiments regarding comparison with ClimODE with consistent experimental settings (e.g., same input and output variables, training years), and discussions of our novelty beyond ClimODE, further demonstrating DeepPrims' effectiveness and contributions.


* **Provide the discussions of medium&long-range forecasts** ($\underline{\text{Reviewers axZ9, k1Xf and Rdm2}}$): we have emphasized that our current research focus is **physics-informed short-term forecasting** and **techniques for medium&long-range forecasts is orthogonal to our research scope**. We have further discussed the corresponding enhancements, further enriching the prospects for future research.

* **Add Clarifications of our work's computational effort** ($\underline{\text{Reviewer Rdm2)}}$: we have provided the computational effort analysis regarding the training and inference time costs, further showing the practicality, reproducibility, and parameter efficiency of DeepPrim.

* **Clarify some technical details and key concepts** ($\underline{\text{Reviewers ZswK and Rdm2}}$): we have clarified key aspects, such as the effect of physical information, the motivation of the source-sink model, further enhancing the technical quality of DeepPrim.

The valuable suggestions from the four reviewers are very helpful for us to improve our paper. We'd be very happy to answer any further questions in the discussion phase.

---

### Meta-Review · Area_Chair_xkse · 2026-01-13

**Summary:**

I’m leaning towards acceptance. The paper makes a coherent case that “physics-structured continuous-time modeling” can buy tangible accuracy gains for short-term weather forecasting, and the experimental story (multi-resolution results + ablations) is solid. The main reason that drives the spread of the reviewer scores is whether the contribution is genuinely beyond a careful light modification of ClimODE and whether the comparisons were fair (variables, training years, protocol). High-level, I do think the proposed work is indeed a 3D primitive-equation extension of ClimODE, not a brand-new paradigm. However, it pushes the same underlying continuous-time “advection as inductive bias” idea with a non-trivial extension: pressure-coordinate 3D primitive-equation setting (vertical coupling across pressure levels) with an explicit decomposition into advection + learned forcing/parameterization modules.

**Reviewer Concerns:**

The rebuttal largely resolves the process/soundness concerns (crediting ClimODE, pressure-gradient handling, formulation-vs-code explanation) and strengthens the empirical case. What remains outstanding is mostly “next layer” work: no real uncertainty quantification, no precipitation/extreme-event evaluation, and no formal conservation/stability guarantees. Then there's debate about novelty beyond ClimODE, which is left open unfortunately due to the discussion being cut short.

**Reviewer Scores:**

All other than ZswK would likely stay positive while I would expect ZswK to raise their score moderately but unlikely to move into a favoring state.

---

### Decision · Program_Chairs · 2026-01-26

Accept (Poster)